# The Polycomb system sustains promoters in a deep OFF state by limiting pre-initiation complex formation to counteract transcription

Aleksander T. Szczurek ✉, Emilia Dimitrova, Jessica R. Kelley, Neil P. Blackledge & Robert J. Klose ✉

The Polycomb system has fundamental roles in regulating gene expression during mammalian development. However, how it controls transcription to enable gene repression has remained enigmatic. Here, using rapid degron-based depletion coupled with live-cell transcription imaging and single-particle tracking, we show how the Polycomb system controls transcription in single cells. We discover that the Polycomb system is not a constitutive block to transcription but instead sustains a long-lived deep promoter OFF state, which limits the frequency with which the promoter can enter into a transcribing state. We demonstrate that Polycomb sustains this deep promoter OFF state by counteracting the binding of factors that enable early transcription pre-initiation complex formation and show that this is necessary for gene repression. Together, these important discoveries provide a rationale for how the Polycomb system controls transcription and suggests a universal mechanism that could enable the Polycomb system to constrain transcription across diverse cellular contexts.

The capacity to initiate and maintain defined gene expression patterns is fundamental to complex multi-cellular development. At its most basic level, this relies on transcription factors recognizing DNA sequences in gene regulatory elements to control RNA polymerase (Pol) II activity at the core gene promoter[1]. However, in eukaryotes, chromatin states at gene regulatory elements can also profoundly influence transcription and gene expression, and the systems that create these states are essential for normal gene regulation and development[1–4]. While there is an emerging appreciation of the mechanisms through which transcription factors instruct transcription[1], how chromatin-based systems influence transcription remains very poorly understood and a major conceptual gap in our knowledge of gene regulation.

The Polycomb repressive system represents a paradigm for chromatin-based gene regulation and is essential for appropriate gene expression during animal development[5–7]. It comprises two distinct histone modifying complexes, Polycomb repressive complexes 1 and 2 (PRC1 and PRC2, respectively). PRC1 mono-ubiquitylates H2A at lysine 119 (H2AK119ub1) and PRC2 methylates histone H3 at lysine 27 (H3K27me3). In vertebrates, both PRC1 and PRC2 are targeted to promoters of genes that have CpG island elements. Here they can deposit histone modifications and through feedback mechanisms create Polycomb chromatin domains that have high levels of H2AK119ub1, H3K27me3 and occupancy of PRC1 and PRC2 complexes. We refer to target genes where Polycomb domains form as Polycomb genes[6,8]. Polycomb chromatin domains have important roles in counteracting gene expression and help to maintain the inactive state of genes in tissues where they should not be expressed[5–7], with previous work also suggesting a more pervasive role in constraining gene expression[8–12]. However, how the Polycomb controls transcription to repress gene expression remains very poorly understood.

Department of Biochemistry, University of Oxford, Oxford, UK. ✉e-mail: aleksander.szczurek@bioch.ox.ac.uk; rob.klose@bioch.ox.ac.uk

**Fig. 1 | Imaging Polycomb gene transcription in live cells. a**, Top: schematic illustrating the transcription imaging approach. MS2 repeats were inserted into a promoter-proximal intron of the genes of interest. As RNA Pol II passes through the array, nascent RNA presents MS2 stem loops that are bound by MCP–GFP leading to accumulation of fluorescence signal at the active transcription site. Bottom: an example image of a cell with a nascent transcription spot corresponding to the active TSS. The white dashed lines indicate the cell outline.

**b**, Example of a transcription activity trajectory from cells engineered to contain the MS2/MCP–GFP system (*Zic2*). Maximal projections of the focalized MCP–GFP signal are shown above the trajectory to illustrate the pulsatile nature of transcription. **c**, Example transcription activity trajectories for Polycomb genes (*Zic2* and *E2f6*) and a reference gene (*Hspg2*). ON (green), permissive (violet), and OFF periods (black) are illustrated. The *y* axis represents transcriptional activity (in RNA molecules). Source numerical data are available in Source data.

A central experimental constraint that has limited our understanding of how gene regulatory mechanisms function in situ is that the process of transcription is not uniform across cells. Instead, transcription is stochastic within individual cells over time and varies substantially between cells in a population[13,14]. As such, ensemble approaches for analysing transcription do not capture key features of the transcription cycle that are essential for understanding how regulatory mechanisms effect gene expression. To overcome this, single-cell transcription analysis complemented with detailed understanding of the cellular dynamics of the factors that regulate transcription is emerging as an important avenue to uncover how transcription is controlled to regulate gene expression[13,14].

We and others have shown using ensemble approaches in embryonic stem (ES) cells that the Polycomb system, in particular PRC1 and H2AK119ub1 (PRC1/H2AK119ub1)[8,15–20], has a central role in constraining gene expression through limiting the activity of RNA Pol II at Polycomb genes[21]. This has demonstrated that factors necessary to promote transcription of Polycomb genes are present and that the Polycomb system limits some key aspect of transcription to enable repression. Analysis of these effects in single cells suggested that Polycomb could influence the frequency of transcriptional bursts, but this observation relied on inferring kinetic parameters based on modelling RNA transcript levels in fixed cells[21–23]. As such, how the Polycomb system controls transcription remains essentially unknown.

To address this fundamental question, here we use rapid degron approaches, live-cell imaging and genomics to determine how PRC1/H2AK119ub1 regulate transcription. We discover that non-canonical PRC1 and H2AK119ub1 have an important role in sustaining a deep promoter OFF state by limiting transcription pre-initiation complex (PIC) engagement with gene promoters to counteract transcription. As such, we reveal that Polycomb chromatin domains limit the earliest steps of transcription to enable gene repression.

## Results

### Imaging Polycomb gene transcription in live cells

To begin understanding how the Polycomb system influences transcription, we used a highly sensitive MS2 aptamer-based system, which is capable of capturing transcription with single-transcript sensitivity in living cells[24] (Fig. 1a). To implement this, we used CRISPR–Cas9 engineering in mouse ES cells to create lines in which MS2 repeats were inserted into the first intron of two representative Polycomb genes (*Zic2* and *E2f6*) that have their promoters embedded within a typical Polycomb chromatin domain (Extended Data Fig. 1a–c) and are subject to very low levels of transcription in wild-type cells but become de-repressed when PRC1 is depleted (Extended Data Fig. 1c,d). We also engineered MS2 repeats into a moderately expressed reference gene that lacks a discernible Polycomb chromatin domain (*Hspg2*) and is not influenced by PRC1 repression (Extended Data Fig. 1b–d). These cell lines were engineered to express an MS2 RNA-binding protein fused to green fluorescent protein (MCP–GFP), enabling nascent transcription imaging and quantification of transcription in live cells[24] (Fig. 1a and Extended Data Figs. 1b and 3a,b).

When we imaged these cell lines, bright MCP–GFP foci were evident which corresponded to nascent RNA-fluorescence in situ hybridization (FISH) signal for each gene (Extended Data Fig. 1b), and we found that nascent transcription could be quantified in live cells with

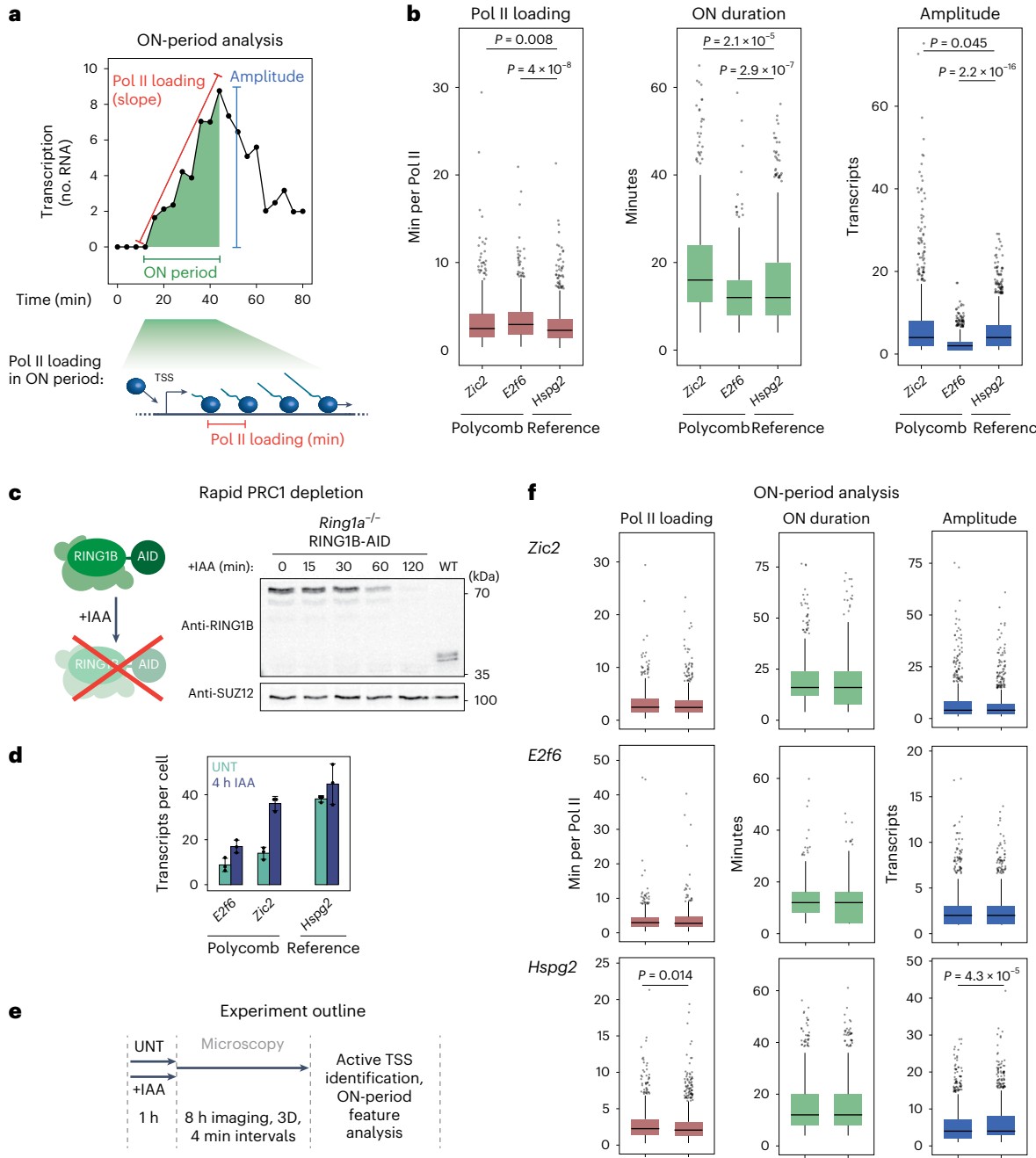

**Fig. 2 | PRC1 does not constrain transcription during ON periods.**
**a**, Schematic illustrating the ON-period features extracted from transcription imaging trajectories. These include the rate of RNA Pol II initiation within the ON period (from linear fit of the slope), the duration of the ON period (min) and the amplitude of the ON period (transcripts). **b**, Box plots centred on the median value comparing the ON-period features, with the interquartile range (IQR) demarcating the minimal and maximal values, whiskers as 1.5× IQR and outliers as dots. Individual data points correspond to individual ON periods (at least 573 measured per box plot). *P* values were estimated using a two-sided Kolmogorov–Smirnov. Box plots represent data from four (*Zic2*), three (*E2f6*) and two (*Hspg2*) biological replicates. **c**, Left: diagram illustrating the auxin-inducible system used to rapidly deplete the catalytic subunit of PRC1 (RING1B) in a *Ring1a*⁻/⁻ background. Right: western blot analysis of RING1B-AID levels over a 2-h period after addition of auxin (IAA) (right) compared with a wild-type (WT)

mouse ES cell line. Shown is a representative example of three independent experiments. **d**, smRNA-FISH analyses of *E2f6*, *Zic2* and *Hspg2* expression 4 h after PRC1 depletion. Dots represent individual biological replicates (*n* = 3, with >400 cells per replicate) and error bars represent the s.d. **e**, Schematic illustrating the approach to image transcription in live cells with (+IAA) or without (untreated, UNT) PRC1 depletion. **f**, Box plots (as in **b**) corresponding to ON-period analysis for *Zic2*, *E2f6* (Polycomb genes) and *Hspg2* (reference) in untreated and PRC1-depleted conditions. Individual data points correspond to individual ON periods (at least 599 measured per box plot). Statistical significance was calculated as in **a** and *P* values < 0.05 are shown. For *Hspg2* Pol II loading, *P* = 0.01376; for *Hspg2* amplitude, *P* = 4.2704 × 10⁻⁵. Box plots represent data from four (*Zic2*), three (*E2f6*) and two (*Hspg2*) biological replicates. Source numerical data and unprocessed blots are available in Source data.

single-transcript sensitivity (Extended Data Fig. 3c,d). Importantly, transcription of Polycomb genes was detected in agreement with these genes being expressed, albeit at low levels (Extended Data Fig. 1c,d). When we measured MCP–GFP fluorescence signal corresponding to nascent transcription over time, we observed that transcription was pulsatile (Fig. 1b), in line with previous live-cell transcription imaging in mammalian cells[13,14]. Furthermore, transcription trajectories for all three genes were characterized by transcriptionally permissive periods, within which there were distinct bursts of transcription initiation that we refer to as ON periods, where multiple RNA polymerases transcribe in close succession (Fig. 1c). Permissive periods were interspersed by long-lived OFF periods where the gene was not transcribed at all. Some OFF periods were highly persistent, extending for the entire duration (8 h) of the imaging movie, and clonal expression analysis revealed instances where OFF periods could extend across cell divisions[24,25] (Extended Data Fig. 2a,b). Therefore, our imaging approach captures the transcriptional behaviour of Polycomb genes and provides us with an opportunity to study how the Polycomb system regulates transcription in live cells.

## PRC1 does not constrain transcription during ON periods

With the capacity to image the transcription of Polycomb genes, we could begin to explore how the Polycomb system might regulate transcription. Initially we focused on ON periods and developed a transcription imaging analysis approach to extract the number of transcripts produced, duration and Pol II loading frequency during ON periods (Fig. 2a and Extended Data Fig. 3e,f). When we compared ON-period features for Polycomb genes (*Zic2* and *E2f6*) and the reference gene (*Hspg2*), we found that they were similar (Fig. 2b) despite Polycomb genes being much more lowly expressed (Fig. 2d).

This suggested that Polycomb-mediated repression may not primarily manifest from limiting transcription during ON periods. To test this, the MS2 reporter system was integrated into a degron cell line in which the addition of the small-molecule auxin (indole-3-acetic acid, IAA) leads to rapid depletion of RING1B, the structural core and catalytic subunit of PRC1, leading to turnover of H2AK119ub1[21,26] (Fig. 2c). Importantly, depletion of PRC1/H2AK119ub1 caused Polycomb gene de-repression and resulted in an approximately 2–2.5-fold increase in transcript levels as assessed by single-molecule RNA-FISH (smRNA-FISH), with *Zic2* reaching transcript levels similar to the reference gene (Fig. 2d). Examining ON-period features, we found they were largely unaffected after PRC1/H2AK119ub1 depletion despite these genes displaying increased transcript levels (Fig. 2d–f). Therefore, we conclude that Polycomb-mediated repression is not achieved by PRC1 constraining transcription during ON periods.

## PRC1 sustains a deep OFF state refractory to transcription

Depletion of PRC1/H2AK119ub1 did not affect transcription during ON periods, suggesting that PRC1/H2AK119ub1 regulates some other feature of transcription. One possibility was that PRC1 could limit the frequency of transcription events (ON periods) during permissive periods or the duration of permissive periods (Fig. 3a). To test this, we imaged transcription in the presence or absence of PRC1 and quantified the time between ON periods within permissive periods (Fig. 3b) and the duration of permissive periods (Fig. 3c). Similarly to ON-period analysis, depletion of PRC1/H2AK119ub1 had only minor effects on transcription during permissive periods, although we did observe a small increase in the duration of permissive periods for the reference gene (Fig. 3c). Therefore, PRC1/H2AK119ub1 does not repress Polycomb genes by regulating either ON-period (Fig. 2) or permissive-period (Fig. 3b,c) features.

Having observed little effect of PRC1/H2AK119ub1 on either ON-period or permissive-period features, we postulated the effects on expression must manifest from an increase in the frequency with which Polycomb genes exit from long-lived OFF periods and enter into

permissive periods where transcription occurs. Consistent with this, when we examined the fraction of time that promoters spend in permissive periods, we discovered PRC1/H2AK119ub1 depletion caused a clear increase, despite permissive-period duration remaining largely unaltered (Fig. 3d). This was also evident in heat maps illustrating single-cell transcription imaging traces for Polycomb genes (Fig. 3e). Although the relative increase in the fraction of time spent in permissive periods and the expression changes after PRC1 depletion do not precisely converge (Figs. 3d and 2d), this is probably due to the non-equilibrium nature of transcript accumulation in our rapid degron system, which relies on the interplay between new transcript production and mRNA half-life. Therefore, we posit that PRC1/H2AK119ub1 counteracts transcription by sustaining promoters in a long-lived deep OFF state and that elevated expression after PRC1 depletion results from an increased fraction of time spent in the permissive period.

## PRC1 decreases the probability of exiting the deep OFF state

If PRC1/H2AK119ub1 represses transcription by sustaining a deep OFF state, an increased frequency of transitioning out of this deep OFF state should account for elevated gene expression observed in smRNA-FISH after PRC1/H2AK119ub1 depletion (Fig. 2d). To investigate this possibility, we built a simple three-state gene expression model that incorporated parameters measured in live-cell imaging for ON periods (Fig. 2b), the number of ON periods and time between them within permissive periods (Extended Data Fig. 4a–d), and transcript half-lives (Extended Data Fig. 4e). Stochastic simulations of gene expression were then carried out with differing probabilities of transitioning from OFF periods to permissive periods ($P_{O>P}$; Fig. 3f) to identify the $P_{O>P}$ value that corresponded to the transcript distributions measured by smRNA-FISH in untreated cells (Extended Data Fig. 4f,g). We then asked whether increasing the $P_{O>P}$ value in these gene expression simulations would reproduce the increased expression and transcript distributions measured in cells when PRC1/H2AK119ub1 was depleted (Figs. 2d and 3f). Importantly, for both *E2f6* and *Zic2* an approximately 2.5-fold increase in $P_{O>P}$ resulted in similar transcript distributions to those observed experimentally after PRC1/H2AK119ub1 depletion, consistent with this being the point of transcriptional control (Fig. 3f). Therefore, by combining live-cell imaging, stochastic simulations and gene expression analysis, we show that the Polycomb system sustains a long-lived deep promoter OFF state that is refractory to transcription to repress gene expression.

## PRC1 counteracts binding of early PIC-forming components

The process of transcription is orchestrated by several distinct regulatory mechanisms that contribute to transcript production[1,27,28]. To understand how PRC1 sustains the deep OFF state, we set out to define what regulatory feature of transcription PRC1/H2AK119ub1 controls. The behaviour of individual factors that regulate the core process of transcription are, like the process of transcription itself, known to be stochastic and highly dynamic. Therefore, capturing the breadth of their dynamic behaviours is not possible using classical ensemble genomic approaches. However, these dynamic behaviours can be measured and quantified in living cells using single-particle tracking (SPT), where the dynamics of individual molecules is directly observed as they interact with chromatin[29–35]. Therefore, we reasoned that a similar approaches could be applied to explore the regulatory stage of transcription affected by PRC1/H2AK119ub1.

To enable SPT, we used CRISPR–Cas9 genome engineering and the HaloTag protein fusion system to label core transcription regulators involved in distinct steps of transcription[27,28] (Fig. 4c and Extended Data Fig. 5a,b). To examine early transcription initiation, we fused a HaloTag to the TATA-box binding protein (TBP) and the TAF1 and TAF11 components of TFIID[36]. TBP function in PIC formation is counteracted by negative cofactor 2 (NC2) through binding to a surface on TBP required for engagement of the general transcription factors TFIIA and B[37].

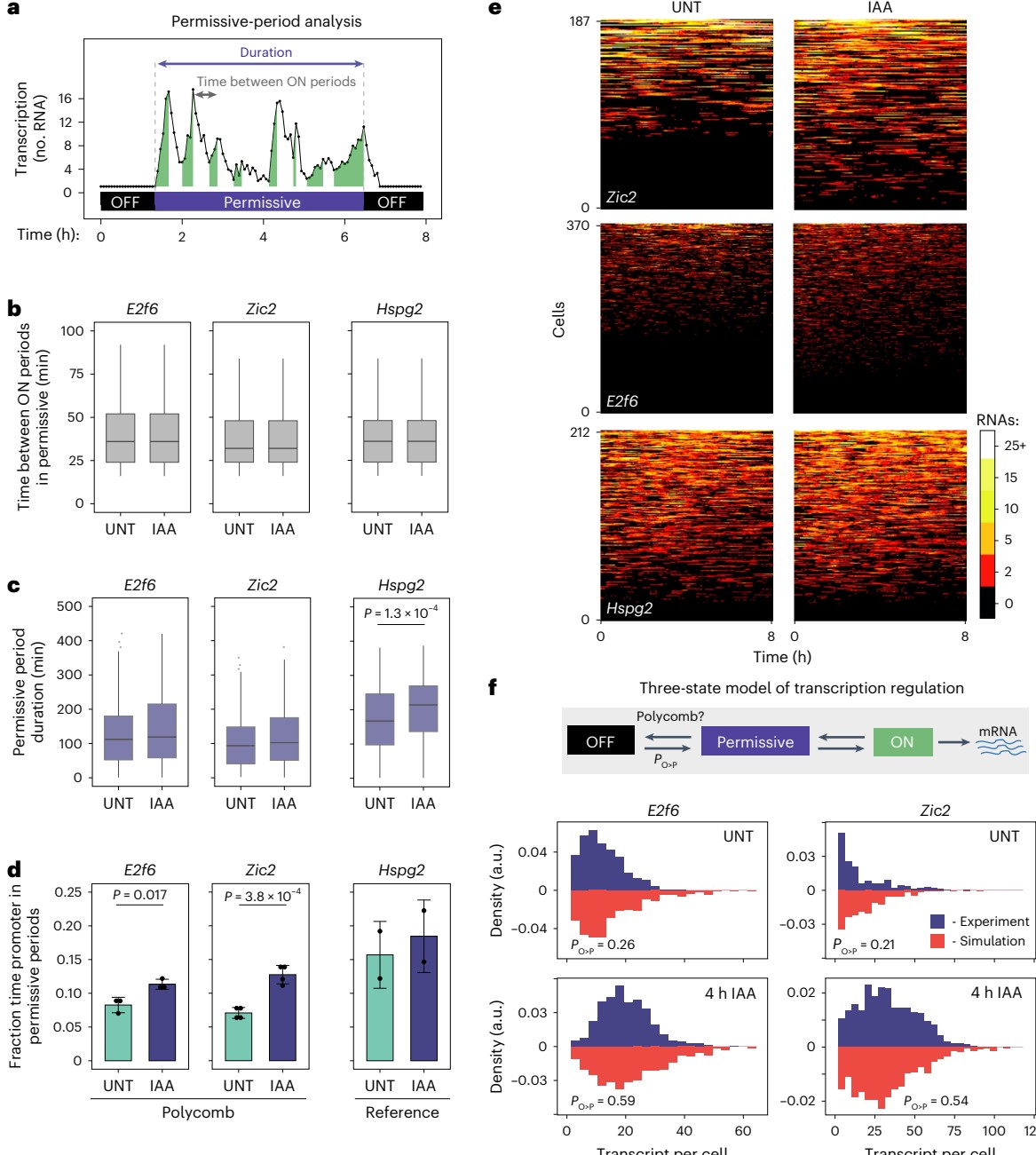

**Fig. 3 | PRC1 sustains a deep OFF state that is refractory to transcription and counteracts gene expression. a**, Schematic illustrating the features extracted from transcription imaging trajectories for permissive-period analysis. These include the time between ON periods within permissive periods (grey arrow) and the duration of permissive periods (purple arrow). **b**, Box plots centred on the median value comparing the time between ON periods for *E2f6*, *Zic2* and *Hspg2* in untreated or IAA-treated (PRC1-depleted) conditions showing the IQR, which is demarcated by the minimal and maximal value, and whiskers as 1.5× IQR. At least 1,010 instances of time intervals between ON periods represent each box plot. *P* values represent two-sided Kolmogorov–Smirnov and *P* values < 0.05 are shown. Box plots represent data from four (*Zic2*), three (*E2f6*) and two (*Hspg2*) biological replicates. **c**, Box plots (as in **b**) comparing the duration of permissive-periods for *E2f6*, *Zic2* and *Hspg2* in untreated or IAA-treated conditions. At least 212 durations of permissive period represent each box plot. Box plots represent data from four (*Zic2*), three (*E2f6*) and two (*Hspg2*) biological replicates. For *Hspg2*, $P = 1.263 \times 10^{-4}$. **d**, Bar graphs showing the fraction of total imaging time

spent in permissive periods for *E2f6*, *Zic2* and *Hspg2*. Data are mean and s.d. from four (*Zic2*), three (*E2f6*) and two (*Hspg2*) biological replicates. For *E2f6*, $P = 0.01756$; for *Zic2*, $P = 3.850 \times 10^{-4}$. **e**, Heat maps illustrating transcription imaging trajectories of individual cells for *E2f6*, *Zic2* and *Hspg2* in untreated or IAA-treated conditions over the 8-h imaging time course (horizontal axis). The amplitude of transcription is illustrated in the scale bar (right) and the number of imaging time courses is indicated on the *y* axis. Heat maps were randomly subsampled to represent equal number of measurements in untreated and IAA-treated conditions to facilitate qualitative comparison. **f**, Top: schematic illustrating the simple three-state model of transcription used to simulate gene expression distributions. Bottom: histograms comparing transcript per cell distributions from smRNA-FISH in experiments (blue bars, experimental) and simulations (red bars) for Polycomb genes in untreated or IAA-treated conditions. The best-fit $P_{O>P}$ value for both untreated and 4 h IAA-treated conditions are indicated. Source numerical data are available in Source data.

Therefore, we fused NC2β to a HaloTag to capture inhibition of early PIC formation, and TFIIB whose interaction with TBP is essential for progression of PIC formation[38]. PIC formation then advances through binding of the mediator coactivator complex, so we fused a HaloTag to the MED14 component of mediator. Once RNA Pol II engages with the PIC, TFIIH is recruited by contacting mediator and RNA Pol II[39,40] and its CDK7 component phosphorylates the C-terminal repeats of RNA Pol II during early transcription elongation. Therefore, we fused CDK7 to a HaloTag to capture this step of transcription. As RNA Pol II enters into early elongation, CDK9 phosphorylates the negative elongation factor (NELF) and RNA Pol II to overcome RNA Pol II pausing and ensure productive elongation. To capture factors related to this stage of transcription we fused a HaloTag to CDK9, NELF-B and the largest subunit of RNA Pol II, RPB1.

To image these transcription regulators in single cells with single-molecule precision, we used a photo-activatable Halo dye coupled with highly inclined and laminated optical sheet microscopy[41]. By imaging at a high frame rate, we quantified the fraction of molecules bound to chromatin (measure of association)[42] (Fig. 4a and Extended Data Fig. 5c) and by imaging at a low frame rate, we estimated the stable binding time of molecules (measure of dissociation)[43] (Fig. 4b and Extended Data Fig. 5d). Interestingly, by focusing on the earliest regulatory steps involving TBP (Fig. 4c), we observed that PRC1/H2AK119ub1 depletion resulted in a nearly 50% increase in the bound fraction of TBP and its binding time also increased (Fig. 4d). This indicates that TBP engages more frequently and remains bound for longer in the absence of PRC1/H2AK119ub1. When we examined the dynamics of other TFIID components, TAF11 showed an increased bound fraction whereas TAF1 was unaffected, but both factors displayed increases in stable binding time. It has been proposed that lobe A of TFIID, which contains TAF11, and lobes B/C of TFIID, which contain TAF1, may exist in distinct pre-assembled subcomplexes[44,45]. This suggests that PRC1/H2AK119ub1 may primarily influence engagement of TBP and TFIID lobe A, with the net result being more stable binding of the TFIID holocomplex. In contrast, the bound fraction of the TBP inhibitory factor NC2β was largely unaffected, but its duration of binding was dramatically reduced, consistent with elevated stable binding of a TBP-containing TFIID complex. The bound fraction and duration of MED14 binding was also elevated upon PRC1 depletion, consistent with mediator engagement depending on TFIID[46]. This suggests that in the absence of PRC1/H2AK119ub1, the association and stable binding of early PIC forming components is increased, whereas the stable binding time of the negative cofactor complex is reduced.

To understand whether these early effects would influence downstream general transcription factors, we examined TFIIB and the TFIIH component CDK7 (Fig. 4d). TFIIB showed only a slight increase in bound fraction but displayed elevated stable binding, whereas CDK7 was largely unaffected. We then examined CDK9 and NELF-B and found that their bound fractions were unaffected, but the stable binding time of CDK9 increased whereas it decreased slightly for NELF-B, in

line with elevated transcription initiation when PRC1/H2AK119ub1 is depleted[47]. Importantly, when we examined RNA Pol II via measuring RPB1 dynamics, we observed little effect, supporting the idea that PRC1 regulates early transcription events and does not considerably affect the amount of elongating RNA Pol II, which is primarily captured in our measurements. Furthermore, this result indicates that the increase in the amount of elongating RNA Pol II that occurs at more lowly expressed Polycomb genes does not contribute enough to the overall amount of elongating RNA Pol II to influence our measurements. On the basis of these detailed kinetic measurements, we propose that PRC1/H2AK119ub1 limits the binding of factors involved in the earliest stages of PIC formation (Fig. 4e).

## cPRC1 does not control stable PIC binding or repression

There are a number of distinct PRC1 complexes which are characterized either as canonical (cPRC1) or non-canonical (ncPRC1) depending on their subunit composition and function (Fig. 5a). cPRC1 complexes contain chromobox (CBX) and polyhomeotic (PHC) proteins, which compact chromatin and can nucleate phase separation of Polycomb chromatin domains[48]. cPRC1 complexes are poor E3 ubiquitin ligases contributing only modestly to H2AK119ub1. ncPRC1 complexes interact with RYBP and YAF2 proteins that stimulate their E3 ubiquitin ligase activity leading to deposition of most H2AK119ub1 in Polycomb chromatin domains[6,8,20] (Extended Data Fig. 6a). To define which PRC1 complexes control the earliest stages of PIC binding to counteract gene expression, we focused on cPRC1 complexes that uniquely form around a single scaffold protein (PCGF2) in ES cells. If the effects on the binding dynamics of the early PIC-forming components and Polycomb gene de-repression were dependent on cPRC1, its depletion should phenocopy complete removal of all PRC1 complexes. Therefore, we engineered bTAG or dTAG degrons into the endogenous *Pcgf2* gene. Addition of the small-molecule compounds AGB1 or dTAG-13 caused a rapid depletion of PCGF2 and a corresponding loss of cPRC1 complex binding to chromatin in Polycomb chromatin domains (Fig. 5b,d and Extended Data Fig. 6b,c).

We then depleted cPRC1 in a HaloTag-labelled TAF11 cell line and carried out SPT to capture the chromatin binding dynamics of TFIID (Fig. 5b). Depletion of cPRC1 increased the bound fraction of TAF11 (Fig. 5c), consistent with the effects observed when all PRC1 complexes were depleted simultaneously (Fig. 4d). However, interestingly, in contrast to the simultaneous depletion of all PRC1 complexes, depletion of cPRC1 did not affect the stable binding time of TAF11 (Fig. 5c). To understand how these cPRC1-dependent effects on TAF11 binding dynamics were related to PRC1-dependent repression, we depleted cPRC1 and examined the expression of *E2f6* and *Zic2* using smRNA-FISH. In stark contrast to depleting all PRC1 complexes simultaneously, rapid depletion of cPRC1 did not result in de-repression of *E2f6* or *Zic2* (Fig. 5d,e and Extended Data Fig. 6d). Together, this demonstrates that cPRC1 can regulate the dynamic interactions TFIID makes with chromatin and its bound fraction, but it does not regulate stable binding of TFIID (Fig. 5c

**Fig. 4 | PRC1 counteracts binding of early PIC-forming components. a**, An example of individually colour-coded single-molecule tracks acquired at high frame rate (left). These tracks are used for kinetic modelling in SPOT-ON[42] to obtain bound fractions. **b**, An example frame from stable binding time measurements acquired at low frame rate with stably bound molecules indicated with arrow heads. Stable binding times for the protein of interest (POI) are extracted from bi-exponential fits (dotted lines) from cumulative distributions (solid lines) and corrected for photobleaching using estimates of stable binding of histone H2B-HT (blue). **c**, A cartoon illustrating stages of PIC assembly and transcription regulation. Protein factors studied by SPT are indicated. **d**, Dot plots illustrating the bound fractions (top) and stable binding time (bottom) for a panel of transcription regulators in untreated or PRC1-depleted (IAA-treated) conditions. Individually colour-coded dots represent values for individual biological replicates and are connected with grey lines, error bars

represent s.d. and horizontal lines show the mean value. A minimum of three biological replicates were measured with approximately 100 cells per replicate for bound fraction analysis and approximately 20 cells for stable binding time measurements per biological replicate. *P* values were determined by one-sided paired *t*-tests and are presented whenever data reach statistical significance ($P < 0.05$). Bound fraction: TBP, $P = 0.012903$; TAF11, $P = 0.01352$; MED14, $P = 0.032109$; stable binding time: TBP, $P = 0.010049$; TAF1, $P = 0.006401$; TAF11, $P = 0.040219$; NC2β, $P = 0.024727$; TFIIB, $P = 0.025326$; MED14, $P = 0.041231$; CDK9, $P = 0.023027$; NELF-B, $P = 0.024577$. **e**, Scatter plot integrating the effects on bound fraction and stable biding times measured in SPT. Dots correspond to the mean fold change (FC) values for individual proteins and the error bars correspond to s.e.m. The data represents at least three biological replicates as indicated in **d**. Solid grey vertical and horizontal lines correspond to 1 (no change). Source numerical data are available in Source data.

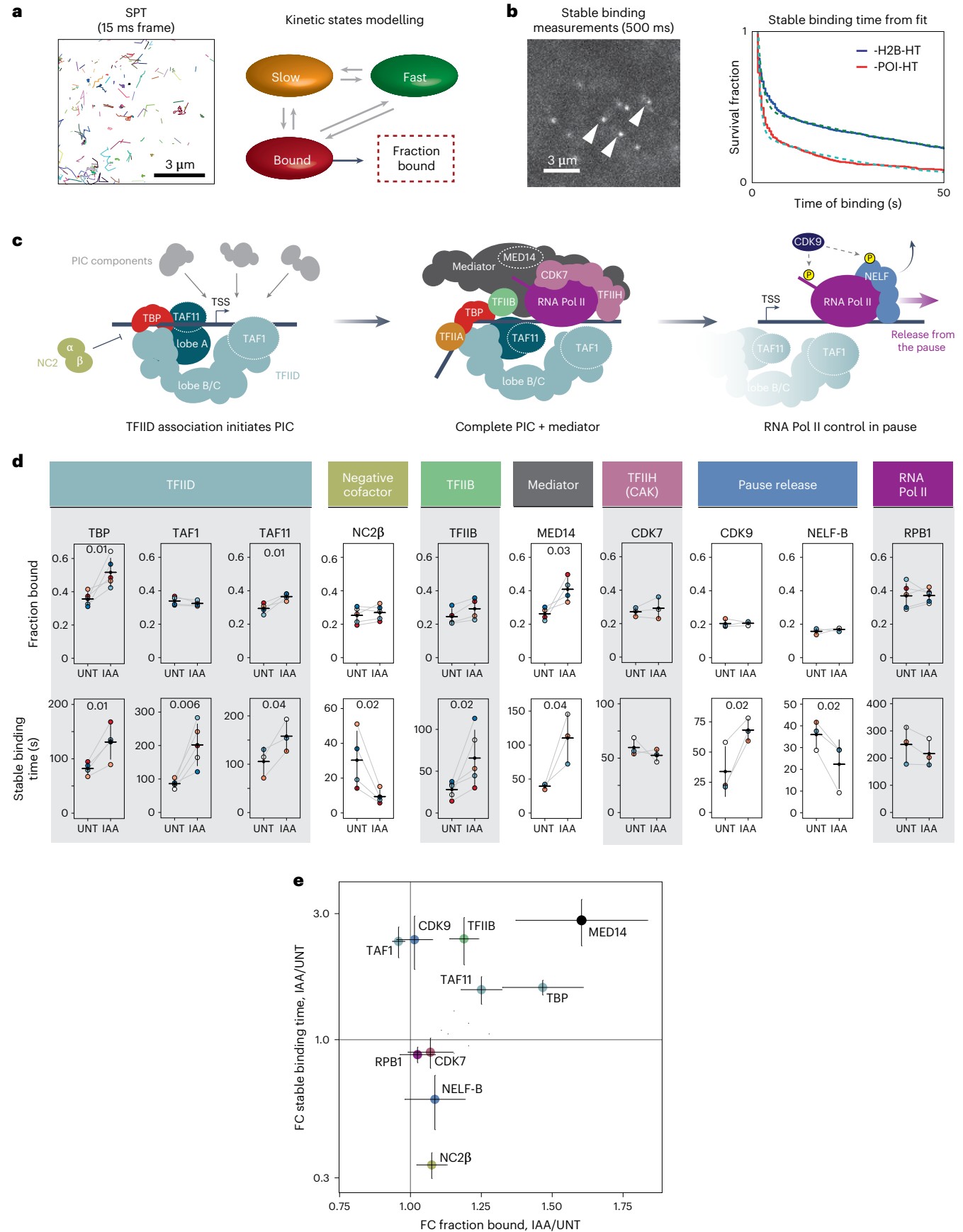

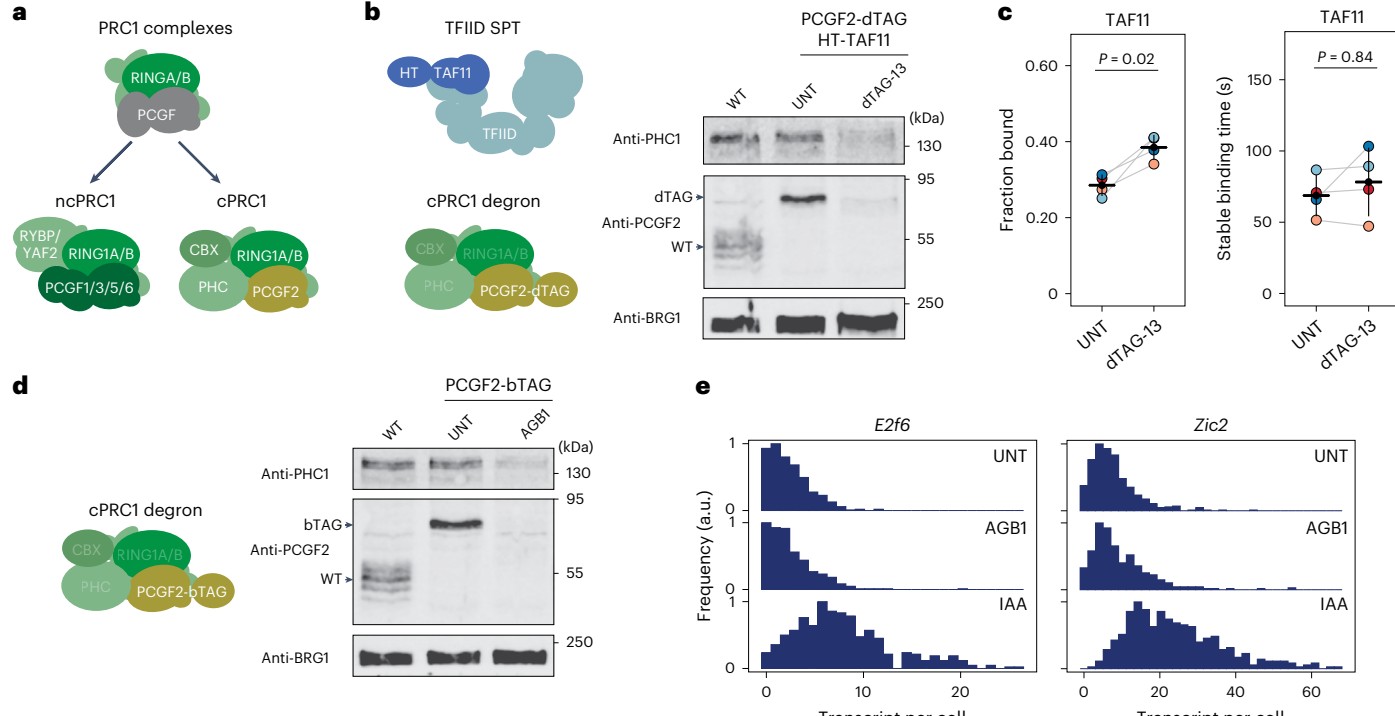

**Fig. 5 | cPRC1 complexes do not regulate stable PIC binding nor contribute centrally to Polycomb repression. a**, A cartoon illustrating the composition of ncPRC1 and cPRC1 complexes. **b**, Cartoon representation of the degron cell line in which cPRC1 complexes can be depleted and TFIID dynamics measured by examining TAF11 by SPT imaging (left). Western blot analysis illustrating depletion of cPRC1 complexes within 2 h of dTAG-13 treatment (right). BRG1 was used as a loading control. The western blot was performed once. **c**, Dot plots illustrating the bound fraction (left) and stable binding time (right) for HaloTag-fused TAF11 (HT-TAF11) in untreated or cPRC1-depleted (dTAG-13-treated) conditions. Individually colour-coded dots represent values for individual biological replicates (*n* = 4) and are connected with grey lines, error bars represent s.d. and horizontal lines show the mean value. *P* values represent

one-sided paired *t*-tests. *P* = 0.020877 (fraction bound); and *P* = 0.846956 (stable binding time). A minimum of approximately 100 cells for bound fraction analysis and approximately 20 cells for stable binding time measurements were measured per biological replicate (indicated as colour-coded dots). **d**, As in **b** except PCGF2 was tagged with bromoTAG (bTAG). Western blot analysis demonstrates PCGF2-bTAG degradation after 2 h of AGB1 treatment. This experiment was performed once. **e**, smRNA-FISH analysis of transcript-per-cell distributions for untreated cells, cells with PCGF2-bTAG depleted (AGB1-treated), and cells with RING1B-AID depleted (IAA-treated). Depletions were performed for 4 h and at least 400 cells were measured for each gene in each condition. Source numerical data and unprocessed blots are available in Source data.

and Extended Data Fig. 6b). This suggests that ncPRC1, as opposed to cPRC1, predominates in counteracting stable TFIID binding and that the absence of ncPRC1 complexes and H2AK119ub1 leads to Polycomb gene de-repression.

## PRC1 constrains TFIID binding to inhibit gene expression

SPT suggested that ncPRC1 or H2AK119ub1 may counteract the stable binding time of TFIID to limit the very earliest regulatory steps of transcription and maintain gene repression. While SPT captures transcription factor binding dynamics with single-molecule precision, it does not provide information about where effects on binding occur in the genome. To understand where TFIID binding was affected, we carried out calibrated chromatin immunoprecipitation coupled to massively parallel sequencing (cChIP–seq) for endogenously tagged TAF1 before and after PRC1 depletion. We chose TAF1 as it is the largest subunit of TFIID and a component of the TFIID holocomplex[36]. When we sorted Polycomb gene and non-Polycomb gene transcription start sites (TSSs) based on PRC1 occupancy, we observed on average the highest levels of TAF1 at non-Polycomb genes (Fig. 6a) in line with these genes being more highly expressed. Importantly, we also observed some TAF1 binding at Polycomb genes, but the levels were much lower, in line with the repressed state of these genes and consistent with the idea that PRC1 could limit TFIID complex binding to sustain a deep promoter OFF state. To test this possibility, we depleted PRC1 and observed a clear increase in TAF1 occupancy at Polycomb genes (Fig. 6a,b), which is

qualitatively consistent with increased stable binding times measured by SPT (Fig. 4d,e). We also validated these effects by ChIP–quantitative PCR (ChIP–qPCR) analysis for TAF1 and other factors identified in our SPT analysis (Extended Data Fig. 7a). Interestingly, using cChIP–seq analysis we also observed a modest yet significant increase in TAF1 binding across non-Polycomb gene TSSs, indicating that PRC1 may also constrain the binding of TFIID more broadly (Fig. 6a,b and Extended Data Fig. 7b). Consistent with this possibility, low levels of PRC1 are detected at non-Polycomb gene promoters, and when we analysed gene expression across these genes, we observed a modest increase in expression after PRC1 depletion (Fig. 6a and Extended Data Fig. 7c). These findings agree with previous observations that PRC1 and H2AK119ub1 may have more subtle yet pervasive effects on gene expression[8,21]. Nevertheless, we find the effects on expression and increases in TAF1 binding correlated best at Polycomb genes (Extended Data Fig. 7e), suggesting that the Polycomb system has a prominent role maintaining these genes in a lowly transcribed or inactive state. Together, these observations indicate that PRC1 limits transcription and gene expression by counteracting TFIID binding to gene promoters, with the largest effects occurring at lowly transcribed Polycomb genes with high levels of PRC1 and H2AK119ub1.

PRC1/H2AK119ub1 depletion caused increased TFIID binding at Polycomb genes and an increased propensity to exit from the deep transcriptional OFF state. Therefore, we wondered whether TFIID was required for the de-repression of Polycomb genes. To test this, we

engineered a degron tag into the endogenous *Taf1* gene in the PRC1 degron cell line (Fig. 6c,d) as TAF1 is integral to the formation of the TFIID holocomplex[45]. We then depleted either PRC1 or PRC1 and TAF1 simultaneously and examined expression of *Zic2* and *E2f6* Polycomb genes using smRNA-FISH (Fig. 6e,f). This revealed that neither Polycomb gene was de-repressed without TAF1, indicating that TFIID binding enables elevated expression in the absence of PRC1/H2AK119ub1 and that Polycomb-dependent transcription control is focused on limiting TFIID-dependent transcription initiation. Therefore, we discover Polycomb-mediated gene repression relies on sustaining a deep OFF state through limiting TFIID binding at gene promoters.

## Discussion

How chromatin states regulate transcription to control gene expression has remained a major conceptual gap in our understanding of gene regulation. Using rapid degron-based protein depletion, transcription imaging and simulations, we discover that the Polycomb system counteracts transcription by sustaining promoters in a long-lived deep OFF state (Figs. 1–3). Using live-cell SPT and genomic approaches, we demonstrate that the Polycomb system sustains this deep OFF state by counteracting binding of factors that enable early PIC formation (Fig. 4) and that this relies on non-canonical as opposed to canonical PRC1 complexes (Fig. 5). Finally, we show Polycomb gene de-repression is caused by increased TFIID association, demonstrating that the Polycomb system limits association of general transcription factors to maintain repression (Fig. 6). These discoveries provide a rationale for how the Polycomb system regulates transcription.

Several distinct models have been proposed to explain how the Polycomb system influences transcription to counteract gene expression[6,21,49–56]. However, these mostly originate from in vitro biochemistry or ensemble fixed-cell analyses that are blind to the dynamic control processes that regulate transcription in living cells. Our transcription imaging now reveals that PRC1/H2AK119ub1 primarily represses transcription and gene expression by limiting transition out of a deep promoter OFF state and into a permissive state where ON periods or bursts of transcription occur. Previously, using static smRNA-FISH analysis and a two-state model of transcription, we concluded that PRC1 might influence gene expression by regulating transcription burst frequency (that is, the frequency of ON periods within permissive periods)[21]. Now, using live-cell imaging in which we directly observe Polycomb gene transcription, we reveal these genes adhere to a three-state model within which PRC1 limits entry into the permissive state. We demonstrate that this is mediated by counteracting association of early PIC components with the promoter, consistent with recent observations

demonstrating that alterations in TATA box sequences that reduce their affinity for TBP and manipulating factors that affect PIC formation also limit entry into permissive periods[22,24,57,58]. Importantly, effects on PIC formation and gene de-repression appear to rely on non-canonical PRC1 complexes that deposit the majority of H2AK119ub1 at Polycomb chromatin domains, consistent with previous work demonstrating the importance of H2AK119ub1 for Polycomb-mediated repression[9,15,16]. Therefore, we identify central role for Polycomb-mediated and chromatin-based repression in regulating the OFF-to-permissive promoter state transition.

Importantly, our findings in live cells differ from previous in vitro biochemical observations suggesting that Polycomb complexes might block recruitment of mediator, but not TBP or TFIID[49]. A possible explanation for this discrepancy is that chromatin templates used in in vitro reconstitution experiments do not contain H2AK119ub1, which we and others have shown is important for repression in vivo[15,16]. Unlike most other histone modifications, ubiquitylation is a bulky 76 amino acid adduct that dramatically alters the nucleosome, suggesting that it could possibly function to repress transcription by influencing how transcription and other regulatory factors interact with promoter chromatin[39,40]. Recent biochemical and structural work has shown that TFIID and other components of the general transcription machinery make key contacts with nucleosomes as part of early transcription initiation mechanisms[40]. With this in mind, an important avenue for future in vitro biochemical and structural work will be to understand whether H2AK119ub1 influences core transcriptional machinery interaction with promoter chromatin to enable gene repression.

Gene expression is dynamic throughout mammalian development. For example, genes may be inactive during early development and their repression maintained by the Polycomb system, but later in development their expression may be required. Consistent with this requirement, we now discover that Polycomb-dependent repression does not act as a constitutive block to transcription, but instead functions by limiting binding of early PIC-forming components to reduce the probability that a promoter enters into a transcriptionally permissive state. Given the breadth of gene types the Polycomb system must regulate in distinct cellular contexts, limiting general transcription factor function may provide a universal means to constrain transcription at genes with diverse regulatory inputs without having to influence highly divergent gene-specific DNA binding factors or other regulatory influences. In the context of developmental transitions when Polycomb genes become activated, we envisage that limiting the frequency of entering into permissive periods could also ensure low-level activation signals are quelled, yet the gene promoter would remain receptive to strong and

**Fig. 6 | PRC1 constrains TFIID binding to inhibit gene expression. a**, Heat map illustrating cChIP–seq signal for RINGB (PRC1) (green, left) or endogenously T7-tagged TAF1 (blue, right) in untreated or IAA-treated ES cells across TSSs. The distance in kilobases from left and right of TSSs is shown below each heat map. To visualize changes in T7-TAF1 signal, the $\log_2$-transformed fold change in IAA-treated versus untreated (that is, $\log_2\mathrm{FC(IAA/UNT)}$) value is shown to the right of the T7-TAF1 cChIP–seq signal. To visualize steady-state gene expression levels and increases in gene expression after RING1B-AID depletion, RPKM (reads per kilobase per million mapped reads) values for untreated cells and $\log_2\mathrm{FC(IAA/UNT)}$ values were calculated for each corresponding gene using calibrated nuclear RNA sequencing (cnRNA-seq) data[21] and plotted as heat maps on the right. TSSs were segregated into non-Polycomb ($n = 9,899$), Polycomb ($n = 4,869$) and non-CpG islands ($n = 5,869$) groupings based on the presence of non-methylated CpG island (CGI) and binding of PRC1 and PRC2 at their promoters as previously described[8]. Heat maps are ranked by RING1B signal. Genes examined in live-cell imaging of transcription (red) as well as some classical Polycomb genes (black) are indicated. **b**, A meta plot (left) illustrating the $\log_2\mathrm{FC(IAA/UNT)}$ of T7-TAF1 cChIP–seq signal at the three classes of TSSs shown in **a** and a box plot (right) showing $\log_2\mathrm{FC(IAA/UNT)}$ of cChIP signal integrated over ±1 kb from TSSs. The boxes centred on median value show the

IQR to represent minimal and maximal values, the centre lines represent the median and whiskers extend by 1.5× IQR or the most extreme point (whichever is closer to the median), whereas notches extend by $1.58 \times \mathrm{IQR}/n^{1/2}$, giving a roughly 95% confidence interval for comparing medians. *P* values were calculated using a two-sided Wilcoxon rank sum test. ***$P = 2.2 \times 10^{-16}$ (non-Polycomb versus non-CGI), ***$P = 6.5 \times 10^{-132}$ (non-Polycomb versus Polycomb) and ***$P = 2.2 \times 10^{-16}$ (Polycomb versus non-CGI). **c**, Schematic illustrating the combinatorial degron strategy used to examine the contribution of TFIID to de-repression of Polycomb target genes after depletion of PRC1. **d**, Western blot analysis of the levels of RING1B-AID and dTAG-TAF1 after simultaneous addition of IAA and dTAG-13 over a 2-h time course. SUZ12 is shown as a loading control. Shown are the representative result of at least three experiments. **e**, A smRNA-FISH image labelling *Zic2* (Polycomb target) transcripts in untreated cells or after 4 h of IAA treatment (RING1B depletion) illustrating increased transcript numbers. White dashed lines indicate cell outlines. Scale bar, 10 μm. **f**, smRNA-FISH analysis of transcript-per-cell distributions for untreated cells, TAF1-depleted (dTAG-13-treated) cells, RING1B-AID-depleted (IAA-treated) cells, and both RING1B- and TAF1-depleted (IAA + dTAG-13-treated) cells. Depletions were performed for 4 h and at least 400 cells were measured for each gene in each condition. Source numerical data and unprocessed blots are available in Source data.

persistent activation signals necessary to initiate gene expression, as we show is the case of the Polycomb gene *Meis1* (Extended Data Fig. 8). Counteracting weak or inappropriate activation signals may be particularly important during development for suppressing noise and maintaining cell identity, as has been proposed previously as a key role

for the Polycomb system[6]. Once genes are activated, persistent transcription leads to Polycomb chromatin domain erosion in part through the transcriptional machinery guiding Trithorax chromatin-modifying systems, which deposit histone modifications that inhibit Polycomb chromatin domain integrity[5,6,59]. This suggests Polycomb and Trithorax

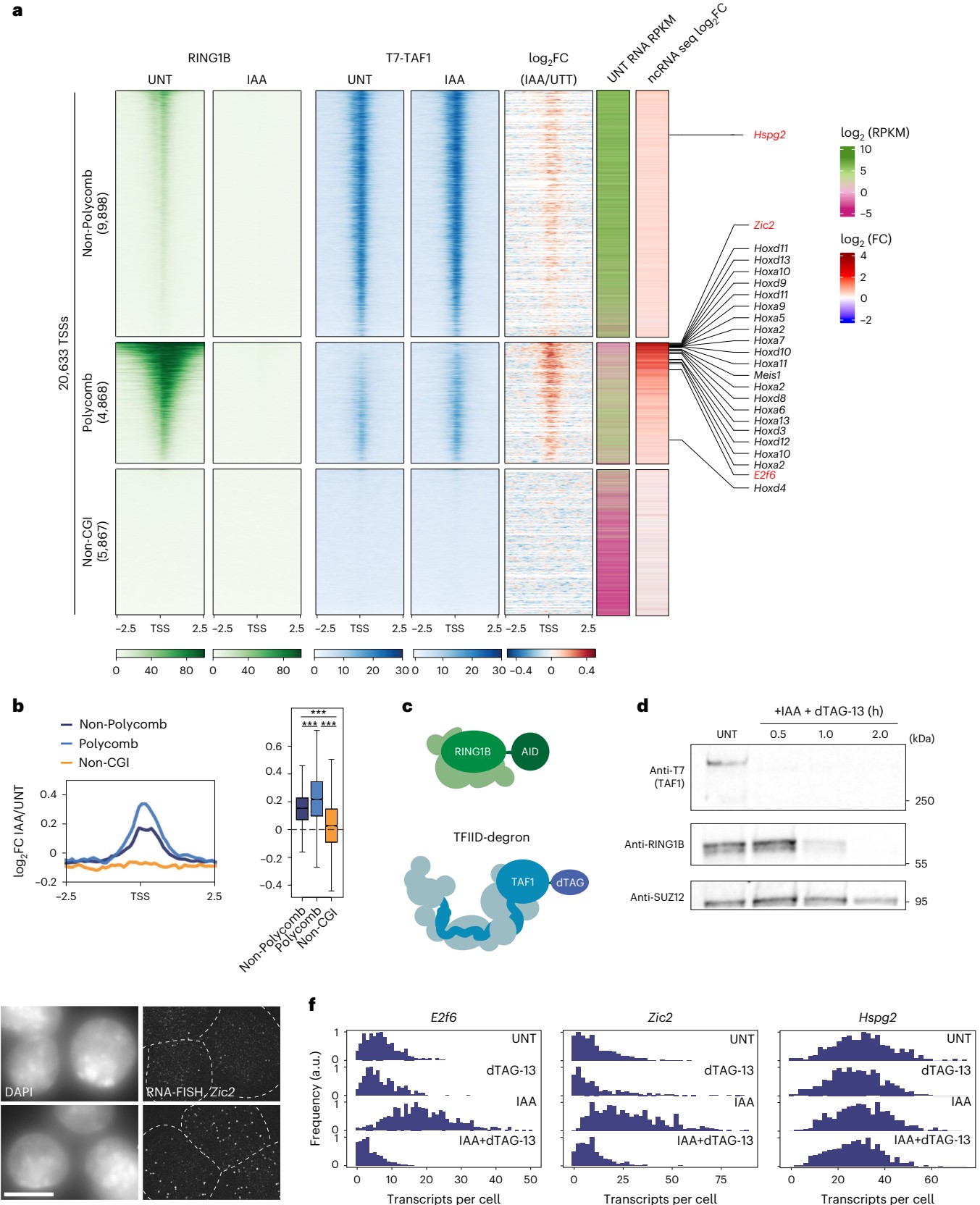

systems may counteract each other by installing chromatin states that decrease or increase the probability that a gene promoter is in a state that is permissive to transcription. In the context of future work, it will be important to uncover whether this control point is the focus of antagonistic Polycomb or Trithorax systems.

In conclusion, we demonstrate that the integration of rapid degron approaches, live-cell imaging of transcription and detailed analysis of transcription regulatory factors by SPT can provide an insight into how chromatin-based gene regulation is controlled in living cells. In doing so, we provide compelling evidence that non-canonical PRC1/H2AK119ub1 represses gene expression by sustaining promoters in a deep OFF state that is refractory to PIC formation and transcription.

## Online content

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

## Methods

### Cell culture

The *Ring1a*[−/−], RING1B-AID mouse embryonic cell line was as previously described and extensively characterized[21,26]. Cells were grown on a gelatinized culture plate at 37 °C and 5% $CO_2$ in DMEM (Gibco) with 10% foetal bovine serum (Sigma), 2 mM L-glutamine (Life Technologies), 1× non-essential amino acids (Life Technologies), supplemented with 0.5 mM β-mercaptoethanol (Life Technologies) and 10 ng ml[−1] leukaemia inhibitory factor (produced in house) and split every other day. To deplete RING1B-AID, cells were treated with IAA (Life Technologies) at 500 μM. To deplete T7-dTAG-TAF1, cells were treated with 20 μM 5,6-dichloro-1-beta-D-ribofuranosylbenzimidazole (DRB) for 1 h, washed three times and treated with 100 nM dTAG-13 (Tocris) for 4 h[60]. To induce degradation of PCGF2-bTAG or PCGF2-dTAG, cells were treated for 4 h with either 500 nM AGB1 or 100 nM dTAG-13, respectively. To induce expression of *Meis1*, the cells were grown in a medium described above for 72 h with 1 μM all-*trans* retinoic acid without leukaemia inhibitory factor.

### Genome engineering

To knock-in HaloTag[61], FKBP12F36V (dTAG, Addgene, 62988), bTAG[62], MS2x128 array[24] or tdMCP-GFP (Addgene, 40649) into specific genomic locations (typically N or C termini of a gene, or the first intron for MS2 array), guide sequences were designed using the CRISPOR tool[63] and cloned into pSptCas9(BB)-2A-Puro(PX459)-V2.0 guide expression plasmid (Addgene, 62988). The complete list of guide sequences can be found in Supplementary Table 1. Targeting constructs used as templates for homology-directed repair were Gibson assembled using Gibson master mix (New England Biolabs) and PCR-amplified homology arms corresponding to the genomic sequence flanking the desired site of insertion. A list of primers used to amplify homology arms are included in Supplementary Table 2. MCP–GFP, dTAG or HaloTag were amplified by PCR from the respective plasmids. The MS2x128 array was cut out of its original plasmid (gift from E. Bertrand)[24] using AleI/NheI restriction enzymes. dTAG was Gibson assembled to include a 3xT7-3xStrepII-tag. Cells were transfected with 2 μg of the targeting construct and 0.5 μg of the guide expressing construct using Lipofectamine 3000 according to the manufacturer's protocol (Thermo Fisher Scientific). One day after transfection, cells were plated sparsely and selected with 1 μg ml[−1] puromycin for 48 h. Puromycin was removed and the cells were grown until distinct colonies formed. Individual clones were picked and propagated in 96-well plates that were then screened for homozygous insertion by PCR. Screening primers are available in Supplementary Table 2. HaloTag and dTAG labelling was validated at the protein level by western blot, and in the case of HaloTag by labelling with tetramethylrhodamine and microscopy (Extended Data Fig. 5a,b). MCP–GFP cells were inspected for expression uniformity (Extended Data Fig. 3a). The integrity of MS2x128-containing lines was further confirmed by PCR using Q5 (New England Biolabs) and Terra (Takara) polymerases as well as by microscopy using RNA-FISH detecting intronic sequences (Extended Data Fig. 1b and Supplementary Table 4) expected to colocalize with nuclear MS2x128/MCP–GFP foci.

### Nuclear extraction and western blot

Nuclear extraction and western blot analysis were performed as described previously[21]. In brief, for nuclear extraction, cells growing on a 10-cm plate were collected, washed once with PBS and resuspended in ten volumes of buffer A (10 mM HEPES pH 7.9, 1.5 mM $MgCl_2$, 10 mM KCl, 0.5 mM dithiothreitol, 0.5 mM phenylmethyl sulfonyl fluoride and protease inhibitor cocktail (Roche)). Subsequently, cells were spun down at 1,500g for 5 min and resuspended in three volumes of buffer A with 0.1% NP-40. Following centrifugation, the pellet was resuspended in one volume of buffer C (5 mM HEPES pH 7.9, 26% glycerol, 1.5 mM $MgCl_2$, 0.2 mM EDTA, protease inhibitor cocktail (Roche) and 0.5 mM dithiothreitol) with 400 mM NaCl and incubated on ice

for 1 h. Nuclei were pelleted by centrifugation at 16,000g for 20 min at 4 °C. The supernatant was retained as nuclear extract. For western blotting, 15–20 μg of nuclear extract was heated in SDS loading buffer at 95 °C for 5 min and loaded on to an acrylamide gel (8–12%) run with Tris–glycine buffer or a 3–8% Tris–acetate NuPAGE gradient gel run with NuPAGE Tris–acetate running buffer (Thermo Fisher Scientific) and separated by electrophoresis. Next, the resolved proteins were transferred onto nitrocellulose membrane using Trans-Blot Turbo Transfer System (Bio-Rad). The membrane was blocked with 5% milk in PBS and 0.1% Tween-20 (PBST-milk) for 1 h. The membrane was transferred to PBST-milk containing primary antibodies and incubated overnight at 4 °C (Supplementary Table 3 contains information on antibodies and dilutions). The next day, membranes were washed three times with PBST-milk and incubated for 1 h with secondary antibody conjugated with IRDye (Li-COR). Following 3 × 5-min washes with PBST and a 5-min wash with PBS, the membrane was visualized with the Odyssey Fc system (Li-COR).

### cCHIP and high-throughput sequencing

cCHIP was performed as previously described[64]. In brief, $5 \times 10^7$ ES cells engineered with T7-dTAG-TAF1 were fixed with 1% formaldehyde (methanol-free, Thermo Fisher Scientific) for 10 min at 25 °C under constant gentle rotation. Fixation was quenched with 150 mM glycine and the cells were washed with ice-cold PBS and snap frozen in LN2. Additionally, $5 \times 10^7$ HEK293T T7-SCC1 cells (a gift from M. Houlard) were fixed with 1% formaldehyde as above and snap frozen in $2 \times 10^6$ aliquots.

For spike-in calibration, $2 \times 10^6$ HEK293T cross-linked cells were resuspended in 100 μl ice-cold lysis buffer (50 mM HEPES pH 7.9, 150 mM NaCl, 2 mM EDTA, 0.5 mM EGTA, 0.5% NP-40, 0.1% sodium deoxycolate and 0.1% SDS) and added to $5 \times 10^7$ fixed ES cells resuspended in 900 μl lysis buffer. The cells were incubated on ice for 10 min and sonicated using Bioruptor Pico sonicator (Diagenode) for 23 cycles (30 s on/30 s off), shearing genomic DNA to produce fragments between 300 bp and 1 kb.

Before immunoprecipitation, chromatin was diluted to 300 μg ml[−1] with lysis buffer and pre-cleared with Protein A agarose beads (Repligen) and blocked with BSA and transfer RNA for 1 h at 4 °C. The pre-cleared chromatin was then incubated with the respective antibody overnight rotating at 4 °C. Antibody-bound chromatin was purified with 20 μl blocked Protein A agarose beads for 3 h at 4 °C. ChIP washes were performed as described previously[64]. ChIP DNA was eluted in 1% SDS and 100 mM $NaHCO_3$ and cross-links were reversed at 65 °C with 200 mM NaCl and RNase A (Sigma) under constant shaking. The samples were then treated with 20 μg ml[−1] proteinase K (Sigma) and purified using a ChIP DNA clean and concentrator kit (Zymo Research). The corresponding input DNA was purified for each sample. The efficiency of each ChIP reaction was confirmed by qPCR. All primers used are listed in Supplementary Table 6.

For cChIP-seq, three reactions were set up for each condition and pooled for library preparation. Before library preparation, 5 ng ChIP DNA was diluted to 50 μl in TLE buffer (10 mM Tris–HCl pH 8.0 and 0.1 mM EDTA) and sonicated with a Bioruptor Pico sonicator for 17 min (30 s on/30 s off). Libraries were prepared using NEBNext Ultra II DNA library prep kit for Illumina (New England Biolabs) and sequenced as 40-bp paired-end reads on Illumina NextSeq 500 platform.

### Massively parallel sequencing, data processing and visualization

For cChIP-seq, paired-end reads were aligned to concatenated mouse (mm10) and spike-in human (hg19) genomes using Bowtie 2 (ref. 65) with the '–no-mixed' and '–no-discordant' options specified. Reads that were mapped more than once were discarded, followed by removal of PCR duplicates using Sambamba[66].

For cChIP–seq visualization and annotation of genomic regions, mouse reads were randomly downsampled based on the spike-in ratio

in each sample[8]. Individual replicates ($n$ = 3) were compared using multiBamSummary and plotCorrelation functions from deepTools (version 3.1.1)[67], confirming a high degree of correlation (Pearson's correlation coefficient >0.9). Normalized replicates were pooled for downstream analysis. Genome-coverage tracks for visualization on the University of California, Santa Cruz (UCSC) genome browser[68] were generated using the pileup function from MACS2[69] for cChIP–seq.

Heat map and meta plot analysis for cChIP–seq was performed using computeMatrix and plotProfile and plotHeatmap functions from deepTools (v.3.1.1)[67], looking at read density at transcription start sites of a custom-built non-redundant mouse gene set ($n$ = 20,633), divided into three categories (non-Polycomb bound, Polycomb bound and non-CGI) based on the presence of a non-methylated CGI and binding of PRC1 + PRC2 at their promoters as defined previously[8]. Intervals of interest were annotated with read counts from merged replicates, using a custom-made Perl script utilizing SAMtools (v1.7)[70]. Box plot analysis of the distribution of $\log_2$FC was performed using a custom R script with boxes showing the IQR and whiskers extending by no more than 1.5× IQR were used. $P$ values were calculated using a Wilcoxon rank sum test. Read counts for all the experiments are included in Supplementary Table 5.

## Gene expression analysis

For gene expression analysis by qPCR with reverse transcription (qRT–PCR), RNA was extracted using a RNeasy extraction kit (Qiagen) and complementary DNA was synthesized using ImProm-II Reverse Transcription system (Promega). qRT–PCR was performed on a Rotor-Gene Q two-plex High Resolution Melt Platform using SYBR Green with primers spanning across exon junctions to prevent the amplification of genomic DNA. All primers used are listed in Supplementary Table 6.

## RNA-FISH protocol and imaging

smRNA-FISH was carried as described previously[21]. In brief, cells were trypsinized and fixed in 3.7% formaldehyde in suspension and then incubated in 70% ethanol at 4 °C for at least 1 h. Cells were then labelled in 2× SSC, 10% formamide and 20% dextran sulfate at 37 °C overnight with a suspension of 48 20–22-nucleotide probes (Stellaris) designed to be evenly distributed across exons or introns of the target transcript. Cells were then spun down and washed multiple times to ensure low non-specific signal. The cells were then incubated with 4,6-diamidino-2-phenylindole (DAPI) to label DNA and agglutinin–Alexa488 to label cell membranes. The cell suspension was mixed 1:1 with Vectashield H-1000 (Vectorlabs), distributed as a monolayer on glass slides and covered with microscopy-grade glass coverslips. Images were acquired using the same microscopy set up as described for live-cell transcription imaging except a 2× magnifying lens was used, resulting in 91.5-nm camera pixel size. To estimate mRNA half-life, transcription initiation was blocked with triptolide (500 nM) for 4 h and the mean numbers of transcripts in cell population were estimated using smRNA-FISH as described above. The experiment was performed in three biological replicates. A mono-exponential decay was assumed to represent the mRNA degradation rates upon transcription block and was used to extract mRNA half-life.

## Live-cell transcription imaging

Transcription was imaged using an Olympus IX83 system fitted with humidified chamber with carbon dioxide atmosphere at 37 °C. The microscope was operated through CellSens software and was equipped with a ×63 1.4-numerical aperture (NA) oil objective lens and a 1,200 × 1,200 px scientific complementary metal-oxide semiconductor (sCMOS) camera (Photometrics). Additional magnifying 1.6× lens was used in front of the camera resulting in final pixel size of 114.4 nm. To image transcription, cells were plated on gelatinized 8-well microscopy μ-slide (IBIDI) 5 h in advance of imaging. At 1 h before imaging, the medium was changed to mouse ES cell medium with fluorobrite

DMEM instead of phenol red DMEM without or with 500 μM auxin in neighbouring wells of the imaging chamber. The imaging conditions were 20 images at 0.7 μm $z$-step interval per frame, 8 h total duration with 4 min time interval. A 20% 490 nm exciting light and 70 ms camera exposure time were used. A minimum of $n$ = 3 biological replicates of untreated and IAA-treated cells were recorded except for *Hspg2*, where two replicates were acquired.

## Identification of active transcription sites in movies

Individual three-dimensional (3D) time-course movies were inspected for cells where there was appearance of transiently accumulating nuclear MCP–GFP signal corresponding to nascent transcription. These cells were cut out and saved as single-cell movies. For foci intensity read out, the following protocol was used: first, the custom-made ImageJ/FiJi script removed the background with rolling ball algorithm (5 px radius) leaving only punctate MCP–GFP signal. Next, 3D Objects Counter[71] was applied to individual 3D time frames to identify active transcription sites in 3D (15 intensity threshold and 10–250 voxel objects). The resulting individual .csv files contained spot volume, intensity and centre of gravity in 3D in individual time frames. The extracted 3D positions were used to confirm correct spot identification in raw movies.

To create time-course fluorescence intensity trajectories for individual active transcription sites (see Fig. 1c for examples) a custom-made R script was used. Overall, the script used previously obtained .csv files with MCP–GFP spot detected in individual time frames to extract the fluorescence intensity of the nascent transcription site and created a combined fluorescence intensity trajectory. In the case of multiple spots detected in a single time frame, for example, when multiple active transcription sites or individual rapidly diffusing pre-mRNAs were identified within the same cell and time frame $t$, the algorithm follows the spot with the shortest 3D Euclidean distance to the spot it already followed in a preceding time frame $t − 1$. If multiple spots were identified in the first time frame of the movie ($t$ = 1), the spot to follow as the transcription site was assigned manually. Every single-cell movie and preliminary trajectory were manually inspected.

These preliminary fluorescence intensity trajectories were then corrected for photobleaching in the following manner: MCP–GFP-expressing cells were imaged with an identical imaging protocol to the one used for live-cell transcription imaging. The constant background intensity value was measured outside the cells and subtracted from every image. The resulting cell images containing only fluorescence signal were thresholded in 3D using 'Huang' settings and total cellular MCP–GFP signal intensity in each time frame was measured. The resulting normalized GFP photobleaching curve representing three biological replicates was approximated with a single exponential fit used next to correct active transcription site fluorescence trajectories through multiplying the extracted transcription site intensity in every time frame $i$ by $1/−\exp(0.05 × i)$, hence accounting for GFP photobleaching during the measurements (Extended Data Fig. 3b). Finally, corrected time course fluorescence trajectories of single active transcription sites were plotted and manually inspected through comparing to raw single-cell movies. A minimum of 250 cells were imaged per biological replicate of which a fraction underwent transcription as judged by MCP–GFP signal accumulation.

## Single pre-mRNA intensity estimation

To capture individual pre-mRNAs reliably, a slightly altered imaging protocol was used. In brief, live cells were imaged in 3D using 20 images at 0.7 μm $z$-intervals with 70 ms camera exposure time (same conditions as used for live-cell transcription imaging); however, a 2× magnifying lens was used (image pixel size 91.5 nm), and resulted in less light arriving at the camera ($0.5723 ± 0.006$ ($n$ = 3 measurements)), and this value was taken into account in single pre-mRNA fluorescence intensity calculation (see below). Exciting light was set at 3× the exciting light intensity used for live-cell transcription imaging

of active transcription sites. For example, 490 nm excitation was set to 83% instead of 20%, which corresponded to 3× higher 490 nm excitation intensity as evident from calibration curve acquired with varying 490 nm excitation intensity and constant camera exposure time (Extended Data Fig. 3c). Candidate single pre-mRNA foci were detected using the 3D Objects Counter[71] after subtracting the background with a rolling ball algorithm twice (radius of 10 px). Foci were identified in (1) two-dimensional maximal projections of 3D images for high-confidence identification and (2) raw 3D images for actual identification. Foci appearing in both approaches were used further. To filter out much brighter spots representing active transcription sites, a maximal volume threshold of $58.6 \times 10^{-3}$ µm³ was applied, the remaining foci were confirmed to be nuclear and were assumed to represent single pre-mRNAs. Their intensity was measured and was further multiplied by $1/0.5723 = 1.747$ (GFP intensity difference originating from using 2× instead of 1.6× magnifying lens, see above) and divided by 3 (to account for 3× the 490 nm excitation intensity used in comparison to actual live-cell transcription imaging protocol). Final single pre-mRNA intensity distributions followed normal distribution with mean (s.d.) of 323 (134), 335 (115) and 330 (116) for *Zic2*, *E2f6* and *Hspg2*, respectively (Extended Data Fig. 3d).

## Analysis of transcription parameters from fluorescence tracks

Transcription ON periods were directly identified in fluorescence trajectories of individual active transcription sites as signal intensity maxima using a custom-made algorithm in R. In brief, the algorithm starts through loading an individual trajectory and uses inflection point identification to attribute individual data points with local maxima or minima with three degrees of strength based on how pronounced they are with respect to surrounding data points. Timepoints where no spot was identified (intensity equal to 0) were automatically set as global minima. The algorithm then plots the trajectories with overlaid candidate preliminary maxima and minima for user inspection. Furthermore, every maximum identified in a fluorescence track was inspected. To identify an ON period, a given maximum is assigned a single nearest preceding minimum because every transcription ON period begins when the fluorescence signal of active transcription site sharply increases and ends when it reaches a maximum. In case no minimum preceding the scrutinized maximum is immediately found while another local maximum is reached, this 'intermediate maximum' is discarded from the analysis and the global minimum search continues until one is found. When a minimum–maximum pair is matched, the fluorescence signal intensity in time frames preceding the maximum is investigated to identify the true end of the ON period. This relies on the fact that the ON period ends when the fluorescence signal ceases to rapidly increase. However, often the global maximum is identified several time frames away due to fluorescence signal fluctuation and the noisy nature of these data. Therefore, to identify the time frame best representing the end of an ON period, the algorithm studies the local relationship of the identified maximum with five preceding frames and resets its position to the time frame where the steep signal increase stops. The final minimum–maximum pair represents an individual ON period. The following parameters are extracted from each ON period: (1) duration time (in minutes), (2) amplitude (in transcripts after converting the arbitrary units of fluorescence into single mRNAs), and (3) RNA Pol II re-initiation rate or time interval between initiating polymerases. To approximate the re-initiation rate, fluorescence signal between respective minimum and maximum within ON period is approximated using a linear fit where its slope represents the speed of transcript production within an ON period. The rate of polymerase re-initiation can only be estimated for ON periods greater than one transcript. Additionally, owing to the 4 min interval used in time course measurements, this analysis could only be reliably carried out for ON periods with amplitudes exceeding 2.5 transcripts (examples are presented in Extended Data Fig. 3e,f).

## Measurements of the fraction of time a promoter spends in the permissive state

Permissive periods were identified from live-cell transcription trajectories as consecutive periods in which ON periods occurred within 60 min of each other. Periods outside of permissive periods were considered OFF periods. To account for the OFF periods that occurred in cells lacking detectable ON periods during the entire 8-h-long trajectory, we assumed that each cell contained on average three alleles, consistent with ES cells spending a large fraction of their cell cycle in S-phase. Assuming alleles are regulated independently of each other (as shown previously[21]) the number of alleles in a permissive period per cell should follow a negative binomial distribution of cells with three, two, one or zero alleles being transcriptionally permissive during the movie. Therefore, the fraction of the cells where no alleles were transcriptionally active was measured (such cells occurred in 8-h-long movies at 36.4(5)%, 40(5)% and 10(3)% for *Zic2*, *E2f6* and *Hspg2*, respectively) and used to simulate a negative binomial distribution of alleles transcriptionally permissive during the movie recapitulating the abundance of the cells with zero alleles that are permissive to transcription (or all three alleles are in OFF state). These distributions (obtained at negative binomial probabilities of 0.284, 0.260 and 0.545 for *Zic2*, *E2f6* and *Hspg2*, respectively) were then used to account for all the alleles in cell population that remained in the OFF state throughout the entire duration of the 8-h-long movie for untreated cells. For the IAA-treated condition, the following values were obtained: cells with zero alleles permissive to transcription comprised 11(2)%, 18(1)% and 9(9)% for *Zic2*, *E2f6* and *Hspg2*, respectively, and the respective probabilities used to simulate negative binomial distributions were 0.65, 0.4355 and 0.555. Lastly, the total duration of permissive-periods for all the alleles was summed and divided by total measurement time (integrated time spent in OFF and permissive periods) to obtain a fraction of time promoter spends in permissive period.

## RNA-FISH in cell colonies

The cells were plated on 8-well IBIDI µ-well chamber (IBIDI) 12, 24 and 48 h before fixation with 3% paraformaldehyde. Then, the cells were permeabilized at 37 °C using 0.5% Triton X-100 for 20 min. RNA-FISH proceeded overnight as described above. Colonies of varying size were manually identified and imaged in 3D using the microscope parameters described above. A custom-made Fiji/ImageJ script was used to manually segment the colonies and cut out maximal projections of individual cells that were then subject to transcript counting using ThunderSTORM[72] as described previously[21].

## Stochastic simulations of transcript-per-cell distributions

The permissive period of the promoter was characterized and the number of ON periods and time between them was measured (Extended Data Fig. 4a,b). First, we simulated permissive periods assuming the number of ON periods follows a Poisson distribution. We further expected that our 8-h-long microscopy measurements may not be able to reliably capture all ON periods within a permissive period and instead can be expected to randomly sample it (Extended Data Fig. 4c). To interpret correctly this experimentally assessed number of ON periods per movie (Extended Data Fig. 4b) and account for the fact that our microscopy measurement may capture only a part of permissive period, we sampled the simulated permissive periods knowing the time interval between ON periods (Extended Data Fig. 4a) using an 8-h-long theoretical measurement sliding window recapitulating our microscopy measurements. The number of ON periods were then counted within that sliding window resulting in the number of ON periods that would be captured experimentally. We then performed this simulation for a range of hypothetical Poisson-distributed numbers of ON periods per theoretical permissive period (Extended Data Fig. 4c) and found a value of ON periods per permissive period (Extended Data Fig. 4d), resulting in a distribution best matching those obtained

experimentally (Extended Data Fig. 4b). This was done through finding a minimum of third-degree polynomial fit (Extended Data Fig. 4c). This strategy allowed us interpret the experimentally measured number of ON periods in 8-h-long microscopy experiments and revealed that number of ON periods per movie measured experimentally for *Zic2* and *E2f6* (Extended Data Fig. 4a) corresponded to Poisson-distributed ON periods per permissive period with means of 8.95 and 9.33, respectively (Extended Data Fig. 4d).

To simulate dynamic transcription of *Zic2* and *E2f6*, we directly measured ON-period amplitudes (Fig. 2b), time intervals between ON periods (Extended Data Fig. 4a) and inferred the number of ON periods per permissive period (Extended Data Fig. 4d). Hence, the simulation of the Polycomb gene was assumed to have three promoter states, that is, an allele may either be in (1) an OFF period (no transcription allowed) or (2) in a permissive period where transcription may take place during (3) ON periods with known amplitudes (Fig. 2b), approximated with a mixed negative binomial and Poisson model, which was then used to randomly draw number of transcripts produced per ON period. Similarly, time intervals between ON periods, were determined by the number of ON periods per permissive period drawn from Poisson distributions (Extended Data Fig. 4d). We simulated individual cells over a period of two 12-h-long cell cycles to allow transcript accumulation. For simplicity, each cell was assumed to have, on average, three alleles (due to relatively short G1 phase in mouse ES cells). Cell cycles were followed by a cell division resulting in random halving the transcript number with 0.5 probability (Extended Data Fig. 4f). Each allele was attributed either OFF or permissive period based on a fixed probability $P_{O>P}$ parameter; each allele drew either of the two and was allowed to repeat the draw once at the onset of the second simulated cell cycle. Then, a third cell cycle of randomly varying duration (0–12 h) was run to desynchronize the cells. At the end, the simulation was stopped and simulated cells containing transcripts accumulated over the full course of simulation were subject to transcript degradation with exponentially distributed survival probability dependent on individual transcript age estimated experimentally (Extended Data Fig. 4e), such that 'old' transcripts were more probable to be degraded. Finally, a transcript-per-cell distribution was obtained having simulated 500 cells.

Simulations were run for a range of $P_{O>P}$ probabilities and the most similar to the experimental mRNA/cell distribution was identified through minimizing the sum-difference between experimental smRNA-FISH and simulated transcript-per-cell distributions (Extended Data Fig. 4g). Using this approach, we identified $P_{O>P}$ values for *Zic2* and *E2f6* in their untreated state. To simulate de-repression following PRC1 depletion, we added an extra step to account for IAA treatment leading to transcript increase: we simulated transcription for an extra 4 h (*Zic2*) and 2.5 h (*E2f6* as we previously noted it de-represses with a delay[21]) where the $P_{O>P}$ probability value was now increased while all the other transcription parameters were fixed and set to the same values for untreated simulations (ON-period amplitude distribution, duration between ON periods and number of ON periods per permissive period). We varied the number of alleles attributed to the cells to account for their different cell cycle stage (cells contained now either two, three or four alleles in OFF or permissive periods). This strategy allowed us to test whether increased $P_{O>P}$ probability can explain the shift in transcript-per-cell distributions following PRC1 depletion (Figs. 2d and 3f). By testing a range of $P_{O>P}$ values, we identified those that recapitulated experimental IAA-treated smRNA-FISH distributions best (Extended Data Fig. 4g, bottom).

## SPT

Cells were plated the day before on gelatinized microscopy dishes with No. 1.5 (MatTek, P35G-1.5-14-C). On the day of measurement, the cells were labelled using 100 nM PA-JF549-Halo (gift from L. Lavis and J. Grimm)[73] for 15 min at 37 °C, followed by washing three times with live-cell imaging medium where regular DMEM was replaced with fluorobrite DMEM (Thermo Fisher Scientific). After 30 min, the cells were washed twice before the live-cell imaging medium was supplemented with 30 mM HEPES.

SPT was performed using the previously described system[61] equipped with an electron multiplying charge-coupled device (EMCCD) camera (Andor, resulting pixel size 96 nm), 100× 1.4 NA objective (Olympus) with objective collar and heated stage maintaining it at 37 °C, laser module (iChrome MLE MultiLaser engine, Toptica Photonics) and translational module (ASI) carrying the fibre optics output used to adjust the beam position between epi and HiLO illumination. For imaging at high camera rate 22 mW of 561 nm laser excitation was used with varied 405 nm excitation to maintain fluorescent signals at low density. A total of 4,000 15 ms frames were acquired per measurement, at least 20 independent measurements containing typically several cells each were acquired per biological replicate. A minimum $n = 3$ biological replicates were acquired for each protein studied.

For stable binding time measurements, after photo-activating sufficient molecules with a 405 nm laser, a long camera exposure time was used (0.5 s) and images were acquired with 0.1 mW 561 nm excitation at different rates for different proteins to adequately address their stable binding: 600 frames at 2 Hz for CDK7-HT, HT-CDK9, NELF-B-HT, T7-HT-TFIIB and HT-NC2β; 300 frames at 1 Hz for T7-HT-Med14 and 200 frames at 0.33 Hz for HT-RPB1, HT-TBP, HT-TAF11 and T7-HT-dTAG-TAF1. Experiments were acquired for a minimum of $n = 3$ biological replicates with a minimum of five movies each and an independent H2B-HT control was measured alongside each replicate to correct for photobleaching (see below).

## SPT analysis

Single-molecule signals were localized with subpixel resolution using stormtracker software[74] running in MATLAB (MathWorks), performing elliptical Gaussian point spread function fit to each single-molecule signal detected based on fixed intensity threshold (the same for all the experiments). Molecule localizations, when appearing in consecutive frames within 8 pixel distance (768 nm) were merged to form tracks (a single frame gap was permitted to account for molecule blinking). The resulting track files were converted to an evalSPT format recognized by the Spot-ON online analysis tool[42] used to determine the molecule-bound fraction through assuming each protein exists in three dynamics states: freely diffusing, slowly diffusing and bound. The following Spot-ON parameters were applied: 0.01 µm length distribution bin width, 10 timepoints, 10 jumps permitted and maximum jump length of 5.05 µm. A localization error of 40 nm was assumed, $z$ correction of 0.7 µm and cumulative density function fitting with three iterations. Diffusion coefficient $D$ was estimated as previously described[74] for tracks that spanned minimum four frames. The resulting $\log_{10}(D)$ distributions were fitted with mix of two Gaussians (mixtools R package) and mobility fractions corresponded to their weights.

## Stable molecule binding time estimation

To estimate stable protein molecule binding times, bound molecules were localized using stormtracker[74]. Subsequently, tracks representing bound molecules were created after identifying signals appearing in consecutive time frames no further away than 192 nm (2 Hz measurements) or 288 nm (0.33 Hz measurements). The distribution of track lengths of stably bound molecules was fit to estimate apparent dwell times $\tau$:

$$y = \frac{A e^{-t/\tau_1}}{e^{-t_1/\tau_1}} + \frac{(1 - A) e^{-t/\tau_2}}{e^{-t_1/\tau_2}}$$

where $y$ denotes the fraction of molecules remaining bound at time $t$, $A$ represents the fraction of the first component of molecules with dwell time $\tau_1$, while $\tau_2$ is usually longer and represents dwell time of the second component extracted to estimate stable binding time (see below).

The first timepoint is represented by $t_1$. Each biological replicate was accompanied by a separate H2B-HT control measurement representing permanently bound molecules. H2B apparent binding time $\tau_{H2B}$ was assumed to be limited solely by dye photobleaching and exceeded that of any measured protein $\tau_{dwell}$. The final corrected protein binding time was defined as follows:

$$\tau_{bound} = \frac{\tau_{H2B} \times \tau_{dwell}}{\tau_{H2B} - \tau_{dwell}}$$

### Statistics and reproducibility

Statistical tests were performed with RStudio 1.2.5019 and Microsoft Excel. Throughout the Article, $P$ values < 0.05 were considered statistically significant. No statistical methods were used to predetermine sample sizes but our sample sizes are similar or greater to those reported in previous publications. No data were excluded from the analyses. The experiments were not randomized. Data collection and analysis were not performed blind to the conditions of the experiments.

### Reporting summary

Further information on research design is available in the Nature Portfolio Reporting Summary linked to this article.

### Data availability

High-throughput sequencing datasets generated in this study are available in the Gene Expression Omnibus (GEO) database under the accession number GSE216636. Published data used in this study include cnRNA-seq for RING1B-mAID (GSE159400)[21]; cChIP–seq for RING1B-mAID, SUZ12, H2AK119ub1, H3K27me3 (GSE159400)[21]; cChIP–seq for PHC1 (GSE119620)[8]; ChIP–seq for RYBP (GSE83135)[18] and annotation for Polycomb domains (GSE119620)[8]. All image datasets or numeric files containing single-molecule localization will be made available upon request. Source data are provided with this paper.

### Code availability

Codes used for the analysis of live-cell transcription data are available via GitHub at https://github.com/aleks-szczure/Szczurek-et-al.-NCB-2024. Scripts used to analyse RNA-FISH data are available via GitHub at https://github.com/aleks-szczure/ThunderFISH. All other codes will be made available upon request.

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

### Acknowledgements

We thank the Klose lab, K. Kus, W. Siwek, S. Uphoff, J. Chubb and L. Tora for input and scientific discussion. We thank L. Lavis and J. Grimm for the gift of the PA-JF$_{549}$-HaloTag ligand, A. Williams for sequencing support, R. Grand for suggesting the bTAG degron, E. Bertrand for the MS2x128 construct, M. Houlard for T7-SCC1 cells and the Micron Advanced Bioimaging Facility for microscopy support (Wellcome Strategic Awards 091911/B/10/Z and 107457/Z/15/Z). The Klose lab is supported by the Wellcome Trust (209400/Z/17/Z) and the European Research Council (681440) and J.R.K by the Oxford-Wolfson Marriott Graduate Scholarship.

### Author contributions

A.T.S. and R.J.K. conceived the project and wrote the Article with contributions from all co-authors. A.T.S. performed most of the experiments, data analysis and visualization. E.D. performed genomics experiments with analyses. N.P.B. performed CHiP–qPCR experiments and analyses. J.R.K. carried out biochemical experiments and contributed to refining the course of the project.

### Competing interests

The authors declare no competing interests.

### Additional information

**Extended data** is available for this paper at https://doi.org/10.1038/s41556-024-01493-w.

**Correspondence and requests for materials** should be addressed to Aleksander T. Szczurek or Robert J. Klose.

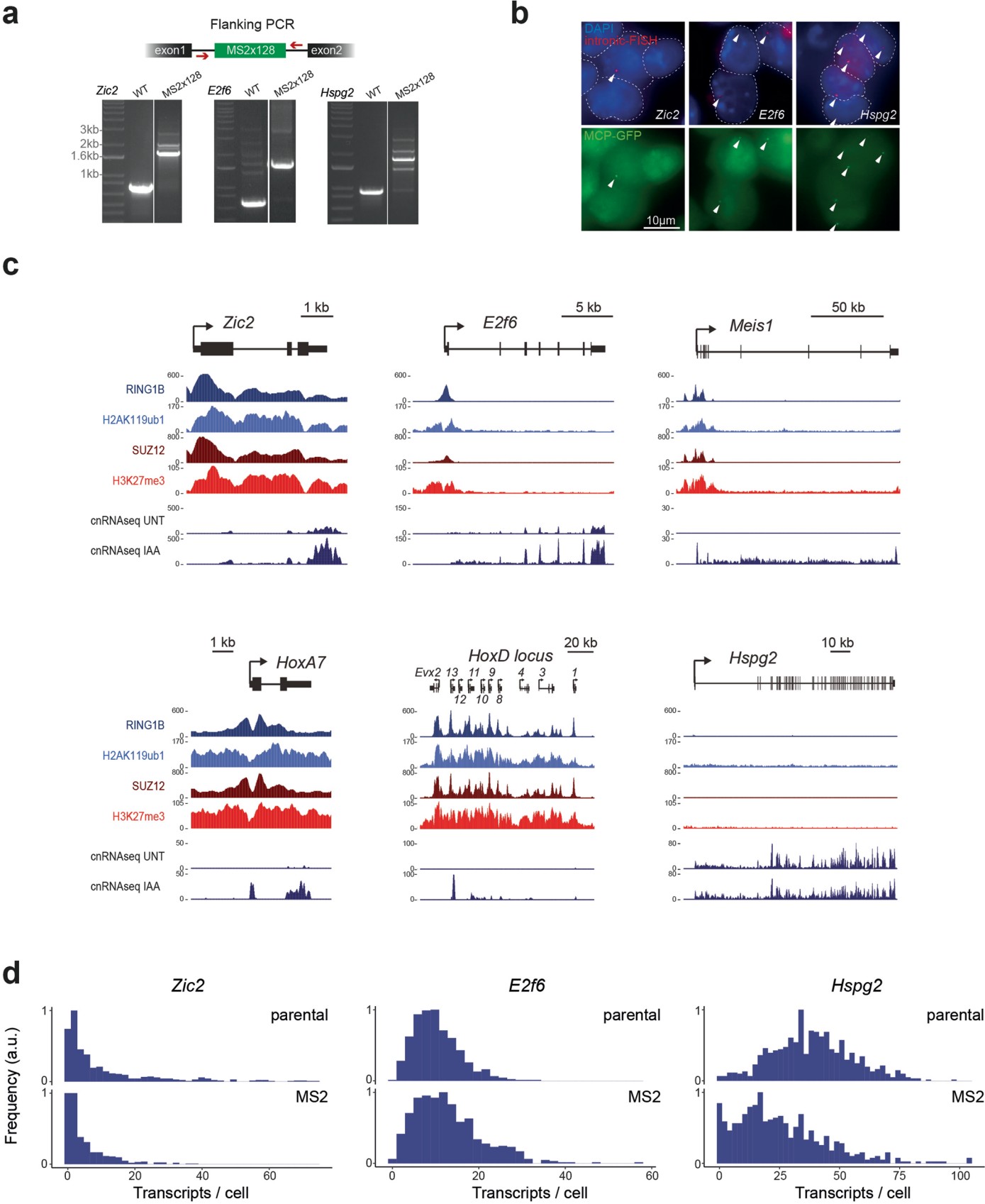

**Extended Data Fig. 1 | See next page for caption.**

**Extended Data Fig. 1 | Characterisation of live-cell transcription imaging in ESCs.** (**a**) Validation that the MS2x128 array is appropriately inserted into the first intron of the corresponding gene. Top: a schematic illustrating the PCR screening strategy. Bottom: PCR results for *Zic2*, *E2f6*, and *Hspg2*. The experiment was repeated twice. (**b**) Images of intronic RNA-FISH (red) and focalized MCP-GFP signal (green) indicating that MCP-GFP accumulates at sites where intronic RNA sequences for *Zic2*, *E2f6*, and *Hspg2* are identified. Nuclei are labelled with 4′,6-diamidino-2-phenylindole (DAPI, blue) and outlined with a dashed line.

Representative of at least 10 images each. (**c**) Genomic cChIP-seq snapshots for *Zic2*, *E2f6*, *Meis1*, *HoxA7*, *HoxD* locus (Polycomb genes) and *Hspg2* (non-Polycomb gene) illustrating signal for RING1B-AID, H2AK119ub1, SUZ12 and H3K27me3. cnRNA-seq signal before and after 4h RING1B-AID depletion is also shown. The data used is from Dobrinic et al.[21]. (**d**) smRNA-FISH analysis of transcript-per-cell distributions for parental (MCP-GFP expressing) and MS2x128 array-containing cell lines. Source numerical data are available in source data.

**a**

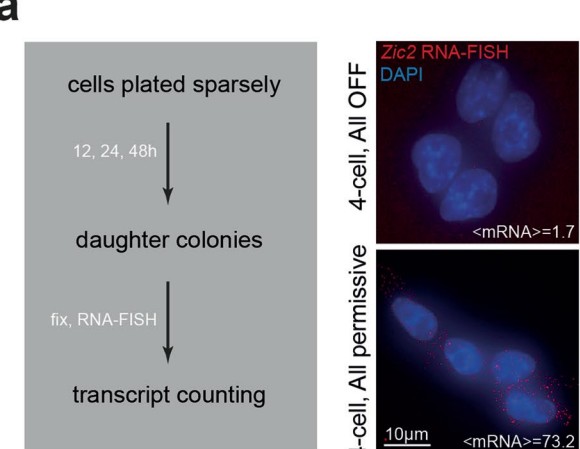

**b**

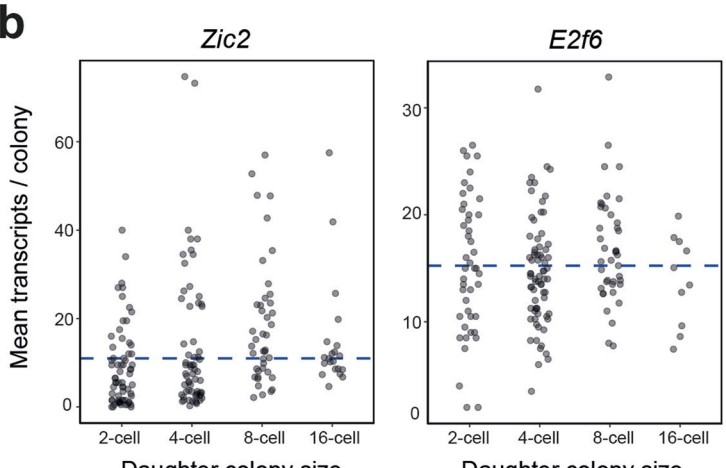

**Extended Data Fig. 2 | Testing heritability of transcription activity of Polycomb-targets across cell divisions.** (**a**) A strategy to assess the number of transcripts-per-cell for Polycomb genes between monoclonal daughter cells (grey box, left). Right: examples of smRNA-FISH images of 4-cell colonies with all cells having or all cells lacking *Zic2* transcripts. This shows that the expression state of Polycomb target genes can be heritably retained across cell divisions. (**b**) Mean number of Polycomb gene transcripts per colony vs. colony size.

Individual dots represent measurements for single monoclonal colonies. The blue dashed line represents the mean number of transcripts-per-cell in all colonies measured. Note, highly- or non-expressing colonies are still found in 4-cell colonies (2 cell divisions) indicating the respective state has been maintained across cell divisions. The data was acquired in two and three biological replicates for *E2f6* and *Zic2*, respectively. Source numerical data and unprocessed blots are available in source data.

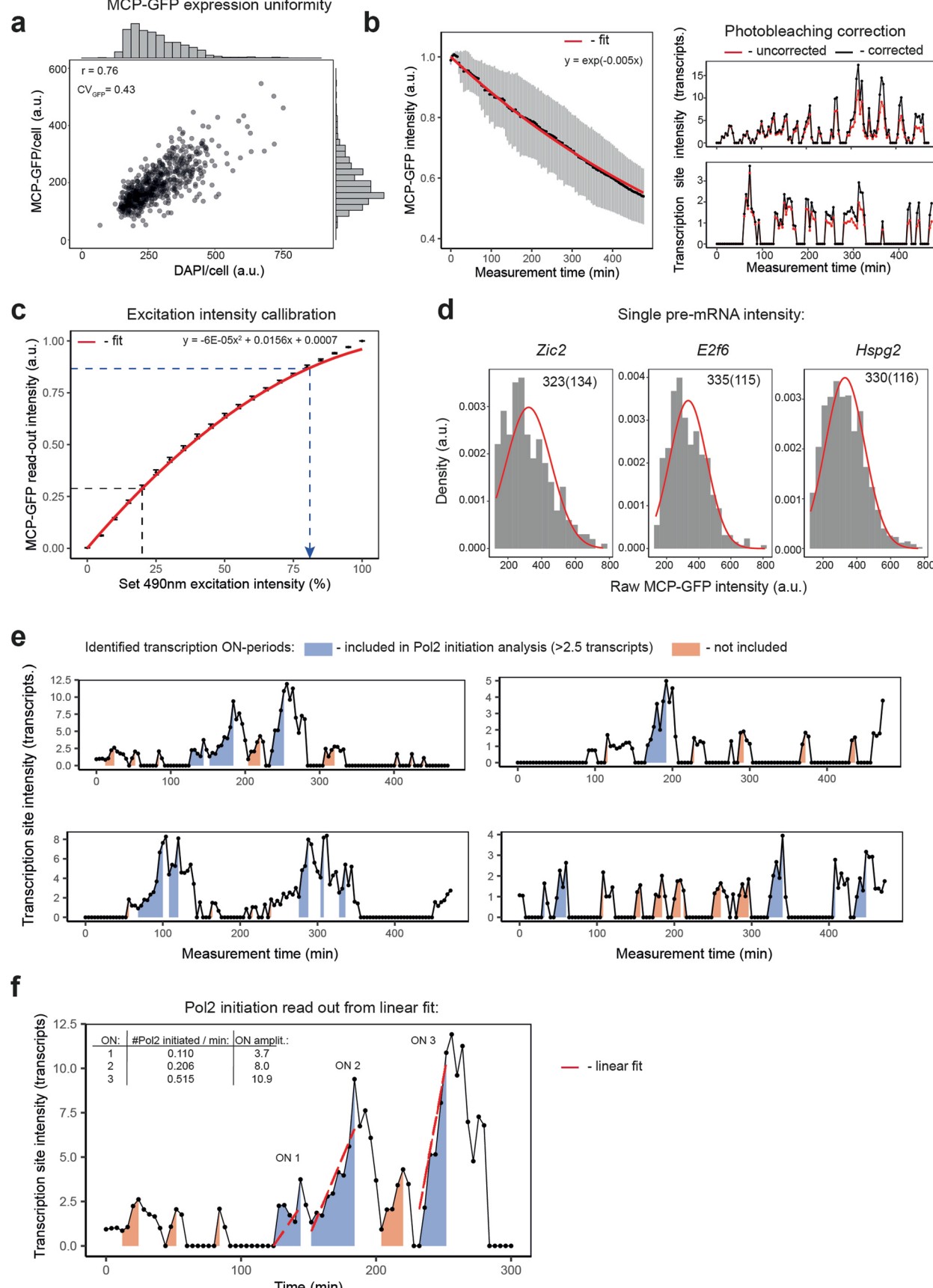

**Extended Data Fig. 3 | See next page for caption.**

**Extended Data Fig. 3 | Characterisation of live-cell transcription imaging with single-transcript sensitivity and ON-period analysis.** (**a**) MCP-GFP expression is uniform across the cell population correlated with DNA content (DAPI signal) (n = 1). (**b**) (Left panel) Measurements of GFP photobleaching (grey datapoints) over a full time-course of live-cell-imaging approximated with an exponential decay (red line) that was used to correct fluorescence intensity in time-course transcription trajectories. (Right panel) Examples of the effect of this correction are presented on the right. (**c**) To measure the intensity of single pre-mRNAs containing 128 MS2 aptamers, imaging was performed using a higher 490nm excitation intensity. The curve quantifies MCP-GFP intensity (y-axis) in response to varying 490nm excitation levels. The blue dashed lines represent values used for live-cell transcription imaging and for single pre-mRNA intensity quantification (dashed line with arrow-head). This curve informed us of the 490nm intensity that excites GFP at 3x the value used in our live-cell transcription measurements. (**d**) Histograms of single pre-mRNA intensities recalculated in values corresponding to live-cell transcription measurements for *Zic2*, *E2f6*, and *Hspg2*. The red line represents a Gaussian fit with mean and standard deviation values indicated above. These values allowed us to recalculate fluorescence intensity units in order to attribute transcript numbers based on fluorescence intensity at the transcription site. Data represents single biological replicate. (**e**) Examples of live-cell transcription trajectories with identified ON-periods indicated in blue or orange depending on whether they were taken into account during RNA Pol II reinitiation rate estimations or not. All ON-periods were taken into account in amplitude and duration analysis. (**f**) An example of a live-cell transcription trajectory with three ON-periods (in blue) with their amplitudes and RNA PolII reinitiation rates (from linear fits, red dashed lines) indicated. Source numerical data are available in source data.

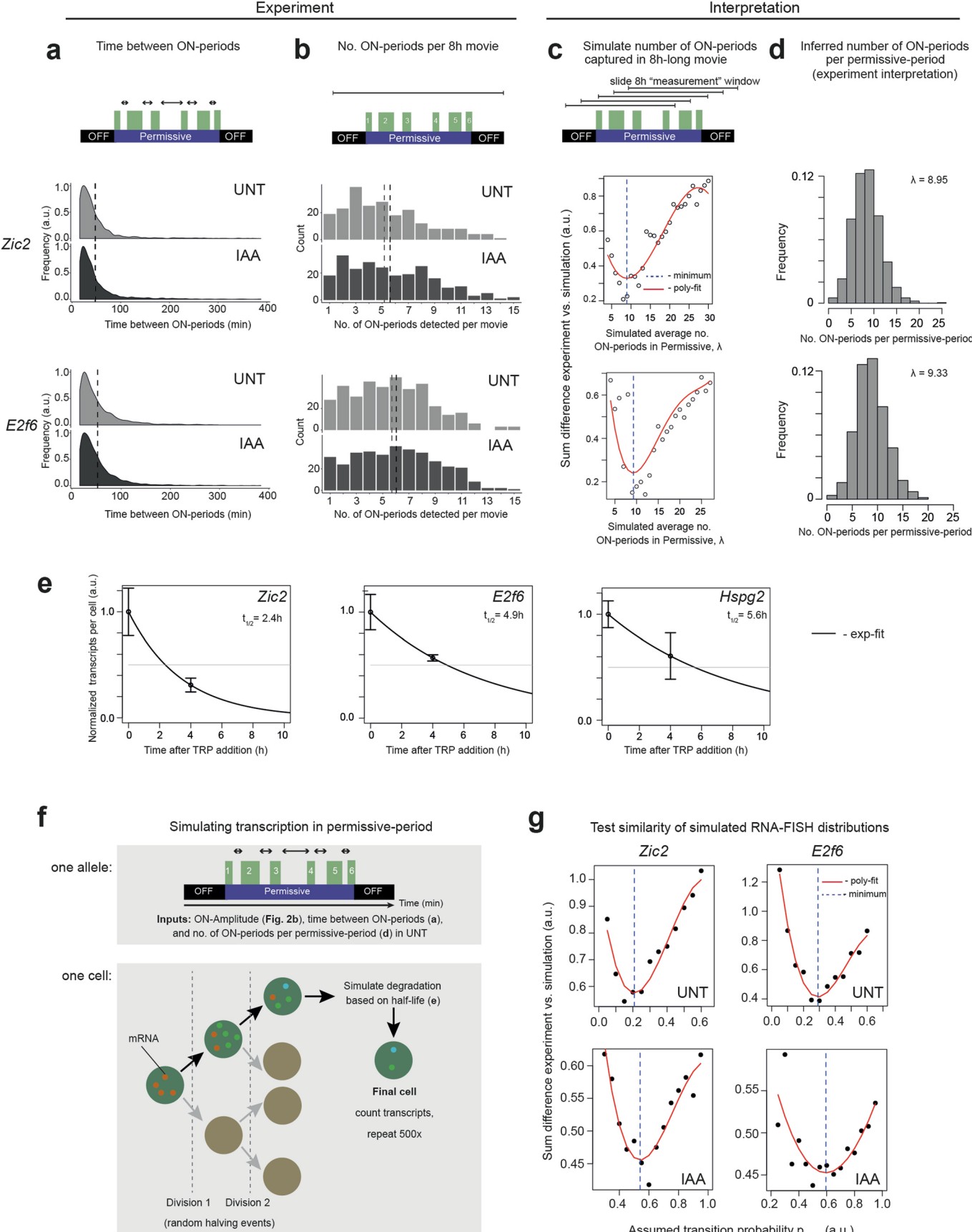

**Extended Data Fig. 4 | See next page for caption.**

**Extended Data Fig. 4 | Stochastic simulations of transcription to obtain transcript-per-cell distributions and estimate transition probability from OFF- to Permissive-states for Polycomb genes.** (a) Density plots of time intervals between ON-periods (indicated as arrows in the cartoon) directly measured from live-cell transcription imaging trajectories for both Polycomb genes *Zic2* (top) and *E2f6* (bottom) for untreated (UNT) and PRC1-depleted (IAA) conditions. Dashed vertical lines represent mean values. ON-, permissive-, and OFF-periods are indicated in the cartoon in green, purple, and black, respectively. (**b**) Histograms of number of ON-periods detected per 8h live-cell transcription movie (indicated in the cartoon as blunt-end horizontal line). Dashed vertical lines represent mean values. (**c**) In order to interpret the detected number of ON-periods per 8h movie and infer the number of ON-periods in a permissive-period, the permissive-periods were simulated with varying mean Poisson-distributed number of ON-periods (λ, x-axis) and 'sampled' using a 'sliding' 8h window to represent the experimental measurement (blunt-end horizontal line in the cartoon). The sum difference between the resulting distribution and experimental distribution (presented in **b**) was calculated (y-axis). The red line represents 3$^{rd}$-degree polynomial fit and its minimum (vertical dashed line) represented the mean number of ON-periods expected to produce most similar distribution of captured ON-periods per 8h measurement window. Plots for *Zic2* (top) and *E2f6* (bottom) are shown. (**d**) Histograms of inferred mean number of ON-periods per permissive-period for *Zic2* (top) and *E2f6* (bottom). (**e**) Estimates of transcript half-lives for *Zic2*, *E2f6*, and *Hspg2*. Data-points represent normalized mean number of transcripts in untreated (t=0) and after 4h of triptolide (TRP) treatment obtained by smRNA-FISH in three biological replicates. Solid black lines represent exponential fits. Horizontal grey lines represent half of the mean transcript number detected in untreated sample while error bars represent standard deviation. The intersection between black and grey lines indicates transcript half-life. (**f**) A cartoon illustrating the strategy to simulate transcription of Polycomb genes. (top) At an individual allele level every parameter of transcription necessary to simulate the permissive-state is quantified or inferred: ON-period amplitude (in transcripts), time between ON-periods, and number of ON-periods in a permissive state. (bottom) Cells were assumed to have on average 3 alleles, and were allowed two full cell cycles followed by cell divisions leading to random halving of the transcript numbers. Single cells were simulated leading to transcript accumulation. Once produced, transcripts were attributed a date-of-birth which was used at the end of the simulation to degrade transcripts based on mRNA half-life. This procedure was repeated 500 times to produce simulated single-cell distribution of transcripts-per-cell. (**g**) The procedure described in (**f**) was repeated using a range of probabilities of transitioning between OFF- and permissive- states ($p_{O>P}$) to produce simulated transcript-per-cell distributions that were then compared to smRNA-FISH experimental data and the most similar were identified by the minimum in 3$^{rd}$ degree polynomial fit (red line) indicated as vertical blue line for *Zic2* (left) and *E2f6* (right) in untreated (UNT) or PRC1-depleted (IAA) conditions. Source numerical data are available in source data.

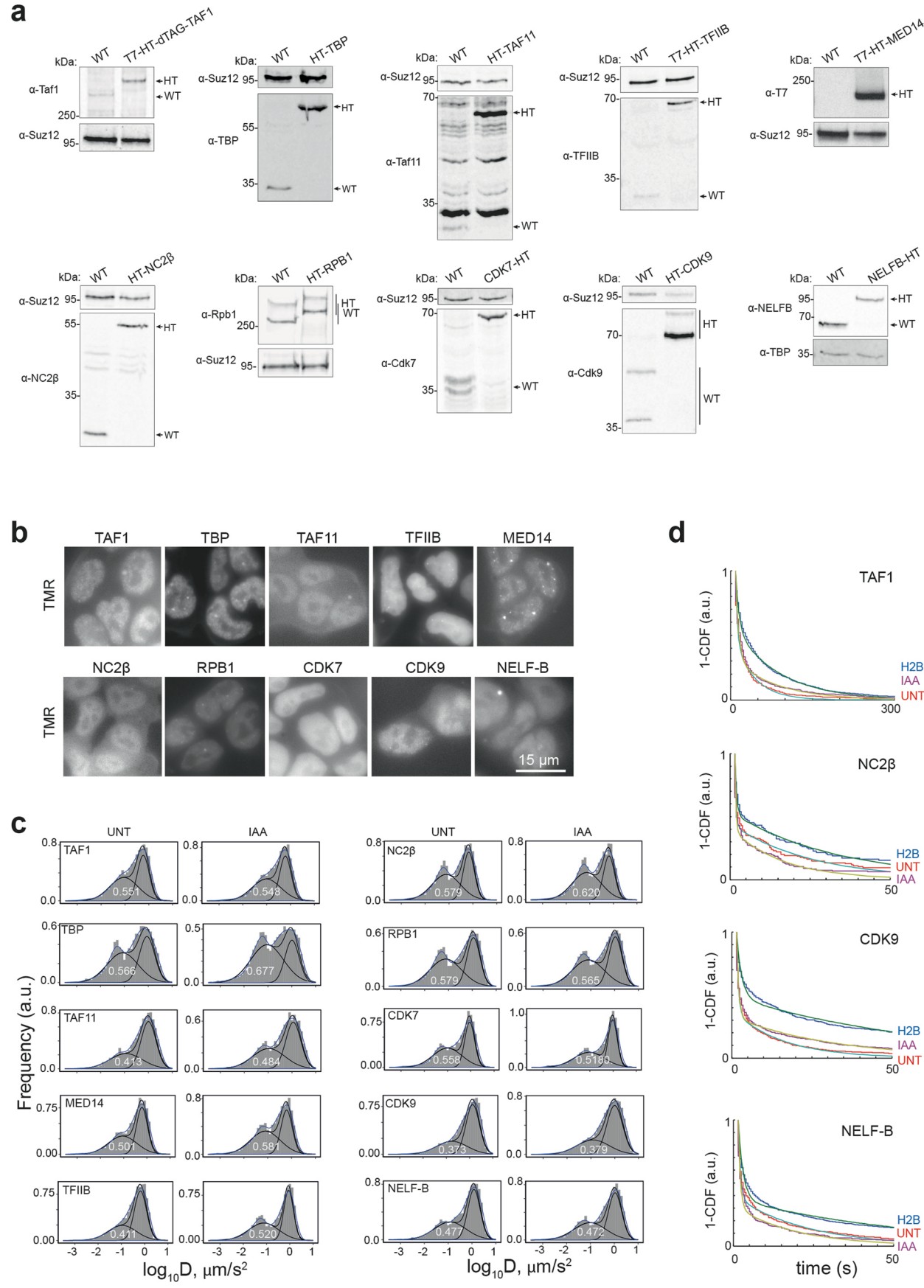

**Extended Data Fig. 5 | See next page for caption.**

**Extended Data Fig. 5 | Extended data to single-particle tracking of transcription regulators.** (**a**) Western blot analysis of endogenously HALO-tagged factors comparing the signals in wild type and tagged lines. Antibodies and molecular weight markers (in kilodaltons (kDa)) are indicated on the left, wild type (WT) and HALO-Tag (HT) protein bands are indicated on the right with arrows. Micrographs are representative results repeated one to three times each. (**b**) Microscopy validation of the HALO-Tag expression in lines with endogenously tagged proteins. HALO-Tag-proteins were visualized using TMR-HALO ligand. All proteins localized to the nucleus. Representatives of at least 3 fields of view. Scale bar represents 15 μm and applies to all the images in the panel. (**c**) Examples of representative biological replicates of histograms of log10(D) calculated from single-particle tracking data acquired at high camera frame rate, obtained for the panel of transcription regulators with (UNT) and without PRC1 (IAA). Black solid lines represent a mixed two-Gaussian fit (to account for immobile and mobile fractions) with indicated value representing immobile portion of molecules. Blue solid line represents histogram density. (**d**) Examples of 1-CDF plots representing single molecule binding times acquired at low camera frame rate. Average stable binding time is extracted from bi-exponential fits indicated in the plots. Examples of data acquired with (UNT, red line) and without PRC1 (IAA, purple line) together with respective H2B-HT (blue). The latter represents a stable binding control used to correct photobleaching.

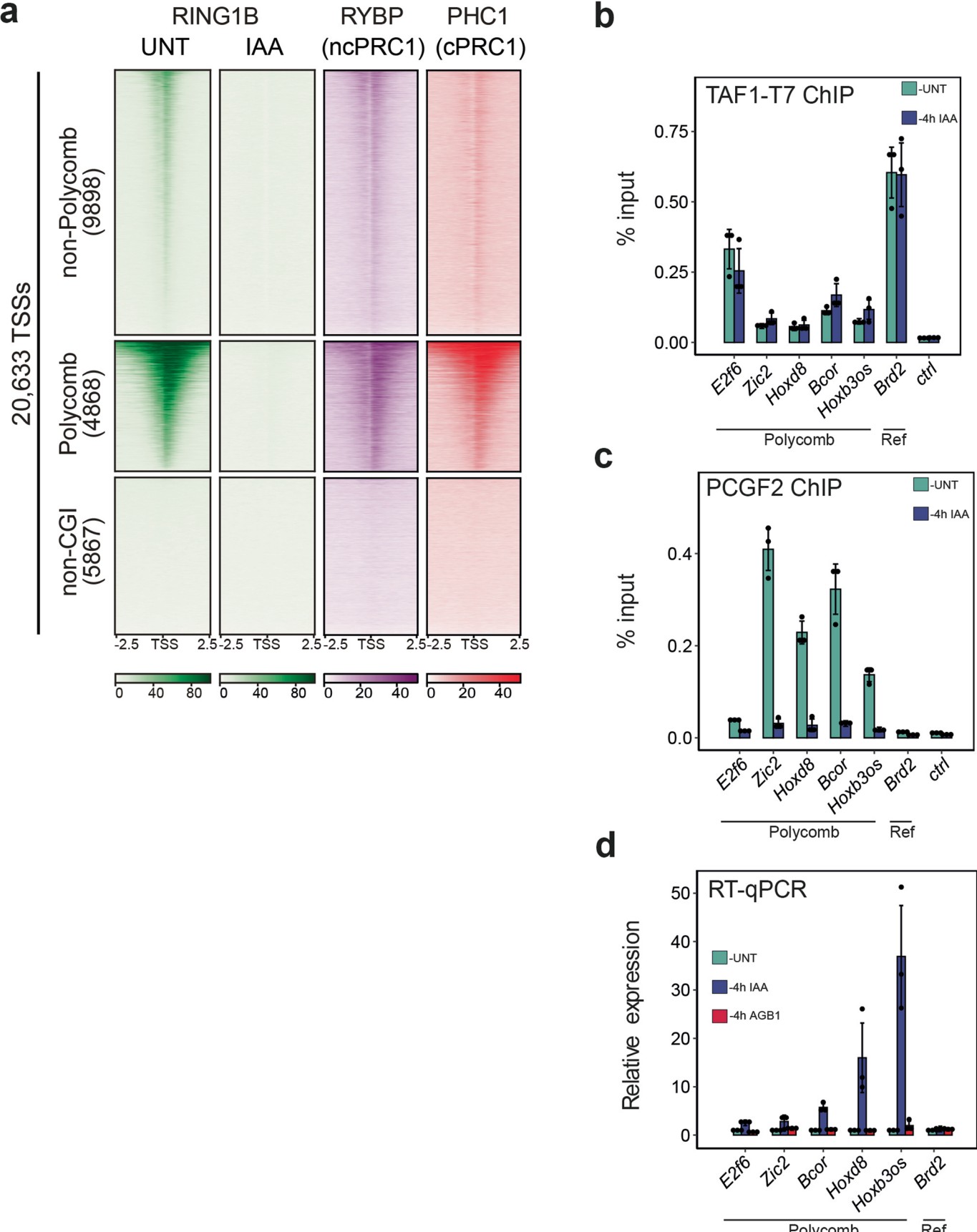

**Extended Data Fig. 6 | See next page for caption.**

**Extended Data Fig. 6 | Genome-wide occupancy of canonical PRC1 complexes and their role in TFIID binding.** (**a**) Heat maps illustrating cChIP-seq signal for RINGB (all PRC1 complexes) (green, left), RYBP (ncPRC1, purple, middle), and PHC1 (cPRC1, red, right). TSSs were segregated into non-Polycomb (n = 9899), Polycomb (n = 4869), and non-CpG islands (n = 5869) groupings as indicated and ranked by decreasing RING1B signal. (**b**) ChIP-qPCR analysis of TAF1 chromatin occupancy at promoters of *E2f6*, *Zic2*, *HoxD8*, *Bcor*, *Hoxb3os* (Polycomb genes), as well as *Brd2* (non-Polycomb gene, 'Ref') prior (UNT, dark blue) and after 4h depletion of PCGF2, a core component of cPRC1 (AGB1, light blue). Ctrl represents ChIP signal at a gene desert region. Error bars represent standard deviation from n = 3 biological replicates throughout the figure. (**c**) ChIP-qPCR analysis of PCGF2 as in (**a**), demonstrating its complete depletion from chromatin after 4h of treatment with AGB1. (**d**) Gene expression analysis of a panel of Polycomb genes using qRT-PCR after 4h depletion of RING1B (all PRC1 complexes, IAA) or PCGF2 (cPRC1 complexes, AGB1). *Brd2* was used as a non-Polycomb gene ('Ref'). Error bars represent standard deviation from n = 3 biological replicates. Source numerical data are available in source data.

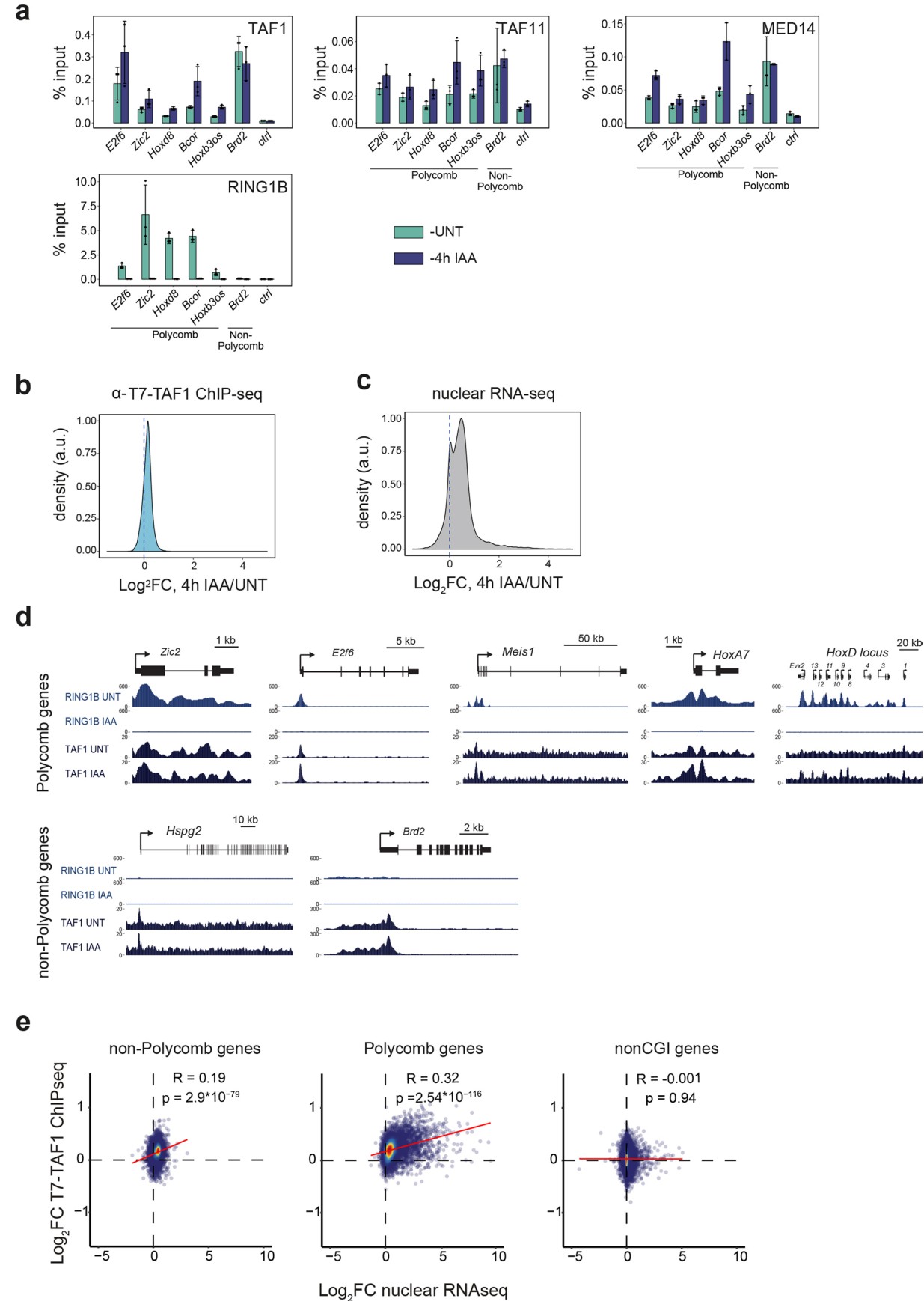

**Extended Data Fig. 7 | See next page for caption.**

**Extended Data Fig. 7 | Effects of PRC1 on the binding of the components of transcription machinery.** (**a**) ChIP-qPCR analysis of TAF1, TAF11, and MED14 chromatin occupancy at promoters of *E2f6*, *Zic2*, *HoxD8*, *Bcor*, *Hoxb3os* (Polycomb genes), as well as *Brd2* (non-Polycomb genes) prior (UNT, blue) and after PRC1 depletion (IAA, orange). Ctrl represents ChIP signal at a gene desert region. RINGB ChIP-qPCR at the target sites demonstrates complete depletion after 4h IAA treatment. Error bars represent standard deviation from n = 3 biological replicates around average values. (**b**) Density plot representing Log2 fold change (4h IAA/UNT) in T7-TAF1 ChIP signal for all the genes (n = 20,633) within the TSSs (+/− 1kb). (**c**) Density plot representing Log2 fold change (4h IAA/UNT) in cnRNA-seq signal for all the genes (n = 20,633). Data from Dobrinic et al.[21]. (**d**) Genomic snapshots for *Zic2, E2f6*, *Meis1*, *HoxA7*, *HoxD* locus (Polycomb genes), as well as *Hspg2* and *Brd2* (non-Polycomb genes) shown RING1B and T7-TAF1 before and after 4h of RING1B depletion (IAA). (**e**) Correlation between changes in expression (Log2 fold change in cnRNA-seq) and changes in T7-TAF1 binding for non-Polycomb genes, Polycomb genes, and genes with no CpG islands (nonCGI genes). R represents two-sided Pearson correlation with exact p-values presented. Source numerical data are available in source data.

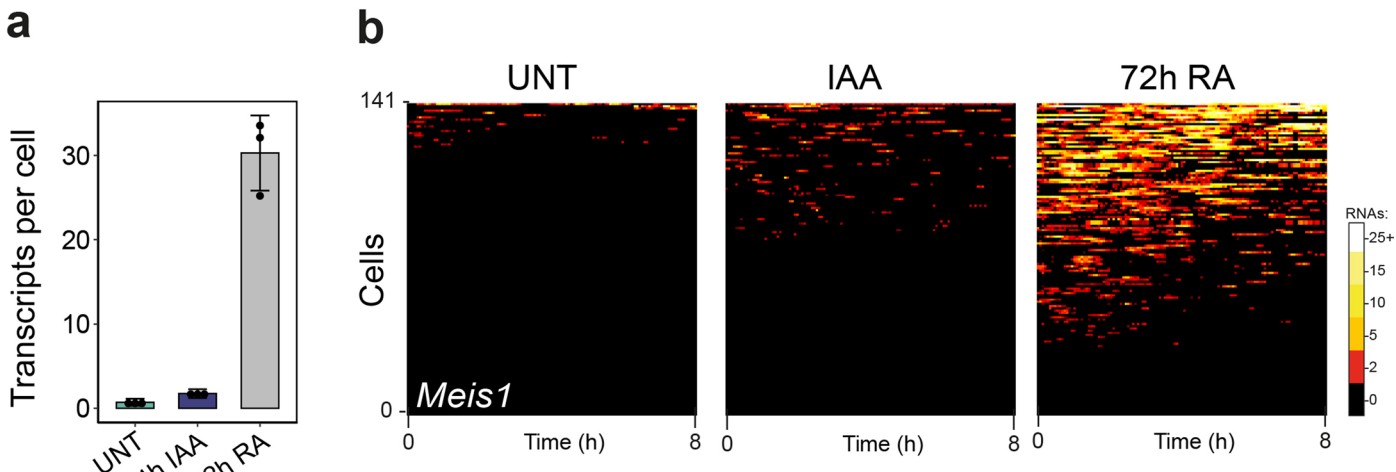

**Extended Data Fig. 8 | Effects of PRC1 depletion on transcription of lowly expressed Polycomb targets are distinct from its activation.** (**a**) Gene expression analysis of *Meis1* after RING1B depletion (4h IAA) and after 72h of retinoic acid treatment (72h RA). Data represents average transcript per cell numbers from single molecule RNA-FISH. Error bars represent standard deviation from n = 3 biological replicates (dots) around the average values.

(**b**) Heatmaps representing live-cell transcription imaging of *Meis1* in untreated (UNT), after RING1B depletion (IAA), and upon retinoic acid treatment (72h RA). Rows represent transcription activity trajectories of individual cells (141 in total). Data represent three biological replicates. Source numerical data are available in source data.

|  | |
|---|---|
| | Aleksander T. Szczurek |

# Reporting Summary

## Statistics

For all statistical analyses, confirm that the following items are present in the figure legend, table legend, main text, or Methods section.

| n/a | Confirmed | |
|---|---|---|
| ☐ | ☒ | The exact sample size (*n*) for each experimental group/condition, given as a discrete number and unit of measurement |
| ☐ | ☒ | A statement on whether measurements were taken from distinct samples or whether the same sample was measured repeatedly |
| ☐ | ☒ | The statistical test(s) used AND whether they are one- or two-sided<br>*Only common tests should be described solely by name; describe more complex techniques in the Methods section.* |
| ☒ | ☐ | A description of all covariates tested |
| ☒ | ☐ | A description of any assumptions or corrections, such as tests of normality and adjustment for multiple comparisons |
| ☐ | ☒ | A full description of the statistical parameters including central tendency (e.g. means) or other basic estimates (e.g. regression coefficient) AND variation (e.g. standard deviation) or associated estimates of uncertainty (e.g. confidence intervals) |
| ☐ | ☒ | For null hypothesis testing, the test statistic (e.g. *F*, *t*, *r*) with confidence intervals, effect sizes, degrees of freedom and *P* value noted<br>*Give P values as exact values whenever suitable.* |
| ☒ | ☐ | For Bayesian analysis, information on the choice of priors and Markov chain Monte Carlo settings |
| ☒ | ☐ | For hierarchical and complex designs, identification of the appropriate level for tests and full reporting of outcomes |
| ☐ | ☒ | Estimates of effect sizes (e.g. Cohen's *d*, Pearson's *r*), indicating how they were calculated |

*Our web collection on statistics for biologists contains articles on many of the points above.*

## Software and code

Policy information about availability of computer code

| Data collection | Image Studio v5.2 (LI-COR), CellSens v.1.17 (Olympus) |
|---|---|
| Data analysis | R version 4.0.2 (R Core Team, 2020), Bowtie 2 (Langmead and Salzberg, 2012), STAR v2.5.4(Dobin et al., 2013), MACS2 v2.1.1 (Zhang et al., 2008), Sambamba 0.6.7 (Tarasov et al., 2015), SAMtools v1.7 (Li et al., 2009), deepTools v3.1.1 (Ramirez et al., 2016), BEDtools v2.17.0 (Quinlan and Hall, 2010), DESeq2 (Love et al., 2014), ImageJ 1.52p through Fiji (Schindelin et al., 2012), Matlab 2013b (Mathworks), Spot-ON v.1 (Hansen et al., 2018) |

For manuscripts utilizing custom algorithms or software that are central to the research but not yet described in published literature, software must be made available to editors and reviewers. We strongly encourage code deposition in a community repository (e.g. GitHub). See the Nature Portfolio guidelines for submitting code & software for further information.

## Data

Policy information about availability of data

All manuscripts must include a data availability statement. This statement should provide the following information, where applicable:
- Accession codes, unique identifiers, or web links for publicly available datasets
- A description of any restrictions on data availability
- For clinical datasets or third party data, please ensure that the statement adheres to our policy

The high-throughput sequencing datasets generated for this study are publically available in the GEO database under GSE216636 accession number (https://www.ncbi.nlm.nih.gov/geo/query/acc.cgi?acc=GSE216636). Localised single molecule positions obtained in particle tracking experiments can be provided upon request. Numerical files for transcription activity trajectories are available upon request. For cnRNA-seq processing we used mm10 (GenBank: BK000964.3, https://www.ncbi.nlm.nih.gov/nuccore/bk000964) and dm6 (GenBank: M21017.1, https://www.ncbi.nlm.nih.gov/nuccore/M21017.1) rDNA genomic datasets.

## Research involving human participants, their data, or biological material

Policy information about studies with human participants or human data. See also policy information about sex, gender (identity/presentation), and sexual orientation and race, ethnicity and racism.

| | |
|---|---|
| Reporting on sex and gender | n.a. |
| Reporting on race, ethnicity, or other socially relevant groupings | n.a. |
| Population characteristics | n.a. |
| Recruitment | n.a. |
| Ethics oversight | n.a. |

Note that full information on the approval of the study protocol must also be provided in the manuscript.

# Field-specific reporting

Please select the one below that is the best fit for your research. If you are not sure, read the appropriate sections before making your selection.

☒ Life sciences ☐ Behavioural & social sciences ☐ Ecological, evolutionary & environmental sciences

For a reference copy of the document with all sections, see nature.com/documents/nr-reporting-summary-flat.pdf

# Life sciences study design

All studies must disclose on these points even when the disclosure is negative.

| | |
|---|---|
| Sample size | Sample sizes were determined based on previous studies using similar techniques to enable reasonable statistical analysis and reproducibility (https://doi.org/10.1038/s41467-021-21130-6). Cell counts for RNA-FISH were previously determined by us (https://doi.org/10.1038/s41594-021-00661-y). Numbers of cells measured in live-cell transcription measurements were large enough to ensure consistency and reproducibility between biological replicates. Number of biological replicates for the ChIP-seq experiments was previously determined by us (https://doi.org/10.1038/s41594-022-00840-5). |
| Data exclusions | No data were excluded |
| Replication | Reported experimental findings were reproducible in multiple biological replicates. The numbers of biological replicates for each experiment are stated. Vast majority of the biological replicates have been performed within a few days apart. |
| Randomization | To plot heatmaps of transcriptional activity and to allow direct comparison between conditions the datasets required to be of the same size. In order to acheive that the data-sets was randomly subsampled to present the same number of cells per condition and hence allow direct qualitative comparison of different conditions as heatmaps (Figure. 3e and Extended Data Figure 8b). cChIP-data were randomly subsampled as mentioned y us previously (https://doi.org:10.1016/j.molcel.2019.03.024). The number of ON-periods per Permissive period in extended Data Figure 4d was assumed to follow Poisson distribution and was randomly drawn from this distribution for the sake of transcription simulations. |
| Blinding | All the imaging experiments have been performed and analysed by a single person and were therefore not blinded. |

# Behavioural & social sciences study design

All studies must disclose on these points even when the disclosure is negative.

| | |
|---|---|
| Study description | *Briefly describe the study type including whether data are quantitative, qualitative, or mixed-methods (e.g. qualitative cross-sectional, quantitative experimental, mixed-methods case study).* |
| Research sample | *State the research sample (e.g. Harvard university undergraduates, villagers in rural India) and provide relevant demographic information (e.g. age, sex) and indicate whether the sample is representative. Provide a rationale for the study sample chosen. For studies involving existing datasets, please describe the dataset and source.* |
| Sampling strategy | *Describe the sampling procedure (e.g. random, snowball, stratified, convenience). Describe the statistical methods that were used to predetermine sample size OR if no sample-size calculation was performed, describe how sample sizes were chosen and provide a rationale for why these sample sizes are sufficient. For qualitative data, please indicate whether data saturation was considered, and what criteria were used to decide that no further sampling was needed.* |
| Data collection | *Provide details about the data collection procedure, including the instruments or devices used to record the data (e.g. pen and paper, computer, eye tracker, video or audio equipment) whether anyone was present besides the participant(s) and the researcher, and whether the researcher was blind to experimental condition and/or the study hypothesis during data collection.* |
| Timing | *Indicate the start and stop dates of data collection. If there is a gap between collection periods, state the dates for each sample cohort.* |
| Data exclusions | *If no data were excluded from the analyses, state so OR if data were excluded, provide the exact number of exclusions and the rationale behind them, indicating whether exclusion criteria were pre-established.* |
| Non-participation | *State how many participants dropped out/declined participation and the reason(s) given OR provide response rate OR state that no participants dropped out/declined participation.* |
| Randomization | *If participants were not allocated into experimental groups, state so OR describe how participants were allocated to groups, and if allocation was not random, describe how covariates were controlled.* |

# Ecological, evolutionary & environmental sciences study design

All studies must disclose on these points even when the disclosure is negative.

| | |
|---|---|
| Study description | *Briefly describe the study. For quantitative data include treatment factors and interactions, design structure (e.g. factorial, nested, hierarchical), nature and number of experimental units and replicates.* |
| Research sample | *Describe the research sample (e.g. a group of tagged Passer domesticus, all Stenocereus thurberi within Organ Pipe Cactus National Monument), and provide a rationale for the sample choice. When relevant, describe the organism taxa, source, sex, age range and any manipulations. State what population the sample is meant to represent when applicable. For studies involving existing datasets, describe the data and its source.* |
| Sampling strategy | *Note the sampling procedure. Describe the statistical methods that were used to predetermine sample size OR if no sample-size calculation was performed, describe how sample sizes were chosen and provide a rationale for why these sample sizes are sufficient.* |
| Data collection | *Describe the data collection procedure, including who recorded the data and how.* |
| Timing and spatial scale | *Indicate the start and stop dates of data collection, noting the frequency and periodicity of sampling and providing a rationale for these choices. If there is a gap between collection periods, state the dates for each sample cohort. Specify the spatial scale from which the data are taken* |
| Data exclusions | *If no data were excluded from the analyses, state so OR if data were excluded, describe the exclusions and the rationale behind them, indicating whether exclusion criteria were pre-established.* |
| Reproducibility | *Describe the measures taken to verify the reproducibility of experimental findings. For each experiment, note whether any attempts to repeat the experiment failed OR state that all attempts to repeat the experiment were successful.* |
| Randomization | *Describe how samples/organisms/participants were allocated into groups. If allocation was not random, describe how covariates were controlled. If this is not relevant to your study, explain why.* |
| Blinding | *Describe the extent of blinding used during data acquisition and analysis. If blinding was not possible, describe why OR explain why blinding was not relevant to your study.* |

Did the study involve field work?   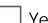 Yes   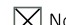 No

# Reporting for specific materials, systems and methods

We require information from authors about some types of materials, experimental systems and methods used in many studies. Here, indicate whether each material, system or method listed is relevant to your study. If you are not sure if a list item applies to your research, read the appropriate section before selecting a response.

## Materials & experimental systems

| n/a | Involved in the study |
|---|---|
| ☐ | ☒ Antibodies |
| ☐ | ☒ Eukaryotic cell lines |
| ☒ | ☐ Palaeontology and archaeology |
| ☒ | ☐ Animals and other organisms |
| ☒ | ☐ Clinical data |
| ☒ | ☐ Dual use research of concern |
| ☒ | ☐ Plants |

## Methods

| n/a | Involved in the study |
|---|---|
| ☐ | ☒ ChIP-seq |
| ☒ | ☐ Flow cytometry |
| ☒ | ☐ MRI-based neuroimaging |

## Antibodies

| | |
|---|---|
| Antibodies used | anti-RING1B Cell Signaling Cat# 5694, Western<br>anti-TBP Abcam Cat# ab818, Western<br>anti-T7 Cell Signalling Cat# 13246, D9E1X, Western, ChIP<br>anti-SUZ12 Cell Signalling Cat# 3737, Western<br>anti-Cdk7 Thermofisher Cat# 31TF2-1F8, Western<br>anti-TFIIB Abcam Cat# ab109106, Western<br>anti-TAF1 Thermofisher Cat# PA5-104490, Western<br>anti-TAF11 Proteintech Cat# 16114-1-AP, Western<br>anti-NC2b Santa Cruz Biotechnology Cat# SC-515024, Western<br>anti-Rpb1 Cell Signalling Cat# 14958, Western<br>anti-Cdk9 Abcam Cat# ab239364, Western<br>anti- NELF-B Cell Signalling Cat#14894S, Western<br>anti-Brg1, Abcam, Cat# ab110641, Western<br>anti-PCGF2, Santa Cruz Biotechnology Cat# sc-10744, Western<br>anti-PCGF2, produced in house, Western<br>anti-PHC1, Cell Signalling, Cat# 13768, Western<br>IRDye 800CW Goat anti-Mouse IgG, LI-COR, Cat# 926-32210, Western<br>IRDye 800CW Goat anti-Rabbit IgG, LI-COR, Cat# 926-32211, Western<br>IRDye 680RD Goat anti-Mouse IgG, LI-COR, Cat# 926-68070, Western<br>IRDye 680RD Goat anti-Rabbit IgG, LI-COR, Cat# 926-68071, Western |
| Validation | anti-RING1B; verified by us previously (Dobrinic et al., 2021) and in this work in Ring1B degron line<br>anti-TBP; verified by us by endogenous tagging and molecular weight shift in Western, manufacturer validated in various cell types by cellular fractionation, 276 citations: https://www.citeab.com/antibodies/753557-ab818-anti-tata-binding-protein-tbp-antibody-1tbp18?des=1ee5e4f398055d5b<br>anti-T7; verified by the manufacturer and by us through Western analysis of nuclear extracts with and withoutT7-tagged protein (Ext. Data Fig. 5a), 19 citations https://www.citeab.com/antibodies/2043128-13246-t7-tag-d9e1x-xp-rabbit-mab?des=770a8fbd62dd70ee<br>anti-SUZ12; manufacturer-validated against various cell lines by Western blot, verified by us previously in Suz12-degron line (Dobrinic et al., 2021)<br>anti-Cdk7; verified by us by endogenous tagging and molecular weight shift in Western, verified by manufactured by knockdown<br>anti-TFIIB; verified by us by endogenous degron tagging, band molecular weight shift and disappearing in Western analysis<br>anti-TAF1; validated here in TAF1 degron line tagged with T7-tag and by molecular weight shift in Western (Ext. Data Fig. 5a)<br>anti-TAF11; verified by us by endogenous degron tagging, band molecular weight shift and disappearing in Western analysis<br>anti-NC2b; verified by us by endogenous tagging and molecular weight shift in Western and degron tagging.<br>anti-Rpb1; verified by us by endogenous tagging and molecular weight shift in Western and by ChIP-seq previously (Dorbinic et al., 2021)<br>anti-Cdk9; verified by us by endogenous tagging and molecular weight shift and presence of second band corresponding to phosphorylation<br>anti-NELF-B; verified by us by endogenous degron tagging, band molecular weight shift and disappearing in Western analysis<br>anti-Brg1; verified by the manufacturer in knockdown and knockout line and by us by endogenous tagging and Western analysis, 213 citations, https://www.citeab.com/antibodies/715090-ab110641-anti-brg1-antibody-epncir111a?des=cf05e3cd167c6000<br>anti-PCGF2 (sc); verified by the manufacturer and by WB in a degron line<br>anti-PCGF2 (custom); verified by WB and ChIPseq in a conditional knock-out line and in two degron lines (this work, Fig. 5)<br>anti-PHC1; verified by the manufacturer and by us in Pcgf2-degron lines where its levels decrease upon Pcgf2 depletion (Fig. 5b, d) |

# Eukaryotic cell lines

Policy information about cell lines and Sex and Gender in Research

| Cell line source(s) | All mouse embryonic stem cell lines used in this study were generated in the Klose lab and were in Ring1B-AID/Ring1A-/- background extensively characterised by us in Dobrinic et al., 2021:<br>MCP-GFP/Rosa26<br>MCP-GFP/Rosa26 MS2x128-Zic2<br>MCP-GFP/Rosa26 MS2x128-E2f6<br>MCP-GFP/Rosa26 MS2x128-Hspg2<br>MCP-GFP/Rosa26 MS2x128-Meis1<br>T7-dTAG-TAF1<br>T7-HaloTag-dTAG-TAF1<br>HaloTag-NC2b<br>HaloTag-TBP<br>HaloTag-Rpb1<br>HaloTag-TAF11<br>Cdk7-HaloTag<br>T7-HaloTag-TFIIB<br>HaloTag-Cdk9<br>NELF-B-HaloTag<br>T7-HaloTag-MED14<br>T7-dTAG-TAF1/PCGF2-bromoTAG<br>HaloTag-TAF11/PCGF2-dTAG<br>Human HEK293T or drosophila SG4 cells (sourced from ATCC) were used as material for calibration but not as an experimental system. |
|---|---|
| Authentication | Each cell line was validated at genetic level by PCR flanking the desired insertion site, and on protein level by Western blot analysis, and by microscopy to test nuclear localisation HaloTag protein tagging. MS2 cell lines were validated for proper array insertion by microscopy and RNA-FISH against intronic sequences meant to colocalize with MCP-GFP signal accumulation. |
| Mycoplasma contamination | All cell lines were regularly tested for mycoplasma contamination and confirmed to be negative. |
| Commonly misidentified lines<br>(See ICLAC register) | No commonly misidentified lines were used in this work |

# Plants

| Seed stocks | Not applicable. |
|---|---|
| Novel plant genotypes | Not applicable. |
| Authentication | Not applicable. |

# ChIP-seq

## Data deposition

☒ Confirm that both raw and final processed data have been deposited in a public database such as GEO.

☒ Confirm that you have deposited or provided access to graph files (e.g. BED files) for the called peaks.

| Data access links<br>*May remain private before publication.* | https://www.ncbi.nlm.nih.gov/geo/query/acc.cgi?acc=GSE216636 |
|---|---|
| Files in database submission | GSM6685040  RING1BmAID-TAF1dTAG-T7-UNT-rep1<br>GSM6685041  RING1BmAID-TAF1dTAG-T7-UNT-rep2<br>GSM6685042  RING1BmAID-TAF1dTAG-T7-UNT-rep3<br>GSM6685046  RING1BmAID-TAF1dTAG-T7-AUX-rep1<br>GSM6685047  RING1BmAID-TAF1dTAG-T7-AUX-rep2<br>GSM6685048  RING1BmAID-TAF1dTAG-T7-AUX-rep3<br>GSM6685049  RING1BmAID-TAF1dTAG-T7-Input-UNT-rep1<br>GSM6685050  RING1BmAID-TAF1dTAG-T7-Input-UNT-rep2<br>GSM6685051  RING1BmAID-TAF1dTAG-T7-Input-UNT-rep3<br>GSM6685055  RING1BmAID-TAF1dTAG-T7-Input-AUX-rep1<br>GSM6685056  RING1BmAID-TAF1dTAG-T7-Input-AUX-rep2<br>GSM6685057  RING1BmAID-TAF1dTAG-T7-Input-AUX-rep3 |
| Genome browser session<br>(e.g. UCSC) | No longer applicable |

# Methodology

| | |
|---|---|
| Replicates | All ChIP-seq experiments were performed in biological triplicates |
| Sequencing depth | All libraries were sequenced as 40bp paired-end reads. |
| Antibodies | anti-T7 Cell Signalling D9E1X, Cat# 13246 |
| Peak calling parameters | All peaks used in this study were published previously (Dobrinic et al, 2021) |
| Data quality | Quality of ChIP-seq data was assessed by visual inspection of individual replicate bigWig files and comparison with other published data sets, as well as by metaplot, heatmap and correlation analysis using deepTools. |
| Software | Paired-end reads were aligned to the concatenated mouse (mm10) and spike-in (dm6 for native, hg19 for cross-linked cChIP-seq) genome sequences using Bowtie 2 ("−no-mixed" and "−no-discordant" options). Only uniquely mapped reads were kept for downstream analysis, after removal of PCR duplicates with Sambamba. Genome coverage tracks were generated using the pileup function from MACS2. Metaplot and heatmap analysis of ChIP-seq read density at regions of interest was performed with computeMatrix and plotProfile/plotHeatmap from deepTools. |

