## [Peer Review File · Nature Cell Biology]

Peer Review Information

Journal: Nature Cell Biology

Manuscript Title: The Polycomb system sustains promoters in a deep OFF-state by limiting pre-initiation complex formation to counteract transcription

Corresponding author name(s): Professor Rob Klose

Editorial Notes:

Reviewer Comments & Decisions:

Decision Letter, initial version:
--

Please delete the link to your author homepage if you wish to forward this email to co-authors.

Dear Professor Klose,

Thank you for submitting your manuscript, "Polycomb sustains promoters in a deep OFF-state by limiting PIC formation to counteract transcription", to Nature Cell Biology and I am very sorry for the delay in communicating our decision to you. The manuscript has now been seen by 3 referees, who are experts in Polycomb (Referee #1); transcription, bursting (Referee #2); and transcription, bursting (Referee #3). As you will see from their comments (attached below), they found this work of potential interest but have raised substantial concerns, which in our view would need to be addressed with considerable revisions before we can consider publication in Nature Cell Biology.

Nature Cell Biology editors discuss the referee reports in detail within the editorial team, including our Chief Editor, to identify key referee points that should be addressed with priority, as opposed to requests that are beyond the scope of the current study. To guide the scope of the revisions, I have listed these points below. Our standard revision process is six months and we are committed to providing a fair and constructive peer-review process, so please feel free to contact us if you would like to discuss any of the referee comments further or if you anticipate any issues or delays addressing the reviews.

I should stress that the referees' concerns about the mechanism are significant and would need to be addressed with experiments and data; reconsideration of the study for this journal and re-engagement of referees will depend on the strength of these revisions. In particular, it would be essential to

dedicate efforts in revision to address the following reviewers' comments:

A. The reviewers requested further analyses and clarifications as to the generality of the proposed mechanism. As Ring1B may bind to both non-Polycomb and Polycomb target genes, it will be important to further probe the model to determine whether the conclusions hold true at non-Polycomb bound genes or if they apply to all or a subset of Polycomb target genes. Additional mechanistic insight linking PRC1 to TFIID binding at gene promoters is also requested:

Rev#1 main introductory comments and points #1, #2, #4, #5, #6

Rev#2 point #1

Rev#3 "Related to the previous point. Assuming the bursting metrics..." paragraph; "Based on these detailed kinetic measurements, we find that PRC1/H2AK119ub1..." last paragraphs

B. The reviewers found the data on the role of TAF1 challenging to interpret. Further investigations are needed to address their points and provide better controlled studies:

Rev#1 point #7

Rev#2 point #2

Rev#3 "When we examined the dynamics of other TFIID components, ..." paragraphs and "We then depleted either PRC1 or PRC1 and TAF1 ..." paragraphs and "Interestingly, we also observed a modest yet significant increase in TAF1 binding..." paragraphs

C. The reviewers raised important concerns about the consistency between changes in transcription bursting dynamics and mRNA levels/copy numbers that need to be addressed thoroughly:

Rev#3 "There is an apparent discrepancy in the effect sizes upon PRC1 depletion ..." paragraph

D. Please also address the reviewers' minor points, requests for clarifications and discussion or text edits, requests for controls or strengthening of existing analyses or data.

E. Finally, please pay close attention to our guidelines on statistical and methodological reporting (listed below) as failure to do so may delay the reconsideration of the revised manuscript. In particular, please provide:

We would be happy to consider a revised manuscript that would satisfactorily address these points,

unless a similar paper is published elsewhere or is accepted for publication in Nature Cell Biology in the meantime.

- ensure that it conforms to our format instructions and publication policies (see below and <https://www.nature.com/nature/for-authors>).
- provide a point-by-point rebuttal to the full referee reports verbatim, as provided at the end of this letter.
- provide the completed Reporting Summary (found here <https://www.nature.com/documents/nr-reporting-summary.pdf>). This is essential for reconsideration of the manuscript will be available to editors and referees in the event of peer review. For more information see <http://www.nature.com/authors/policies/availability.html> or contact me.

When submitting the revised version of your manuscript, please pay close attention to our [href="https://www.nature.com/nature-portfolio/editorial-policies/image-integrity">Digital Image Integrity Guidelines](https://www.nature.com/nature-portfolio/editorial-policies/image-integrity). and to the following points below:

Nature Cell Biology is committed to improving transparency in authorship. As part of our efforts in this direction, we are now requesting that all authors identified as 'corresponding author' on published papers create and link their Open Researcher and Contributor Identifier (ORCID) with their account on the Manuscript Tracking System (MTS), prior to acceptance. ORCID helps the scientific community achieve unambiguous attribution of all scholarly contributions. You can create and link your ORCID from the home page of the MTS by clicking on 'Modify my Springer Nature account'. For more information please visit www.springernature.com/orcid.

This journal strongly supports public availability of data. Please place the data used in your paper into a public data repository, or alternatively, present the data as Supplementary Information. If data can only be shared on request, please explain why in your Data Availability Statement, and also in the correspondence with your editor. Please note that for some data types, deposition in a public repository is mandatory - more information on our data deposition policies and available repositories appears below.

[Redacted]

We hope that you will find our referees' comments and editorial guidance helpful. Please do not hesitate to contact us if there is anything you would like to discuss. Thank you again for considering NCB for your work,

Best wishes,

Melina

Melina Casadio, PhD
Senior Editor, Nature Cell Biology
ORCID ID: <https://orcid.org/0000-0003-2389-2243>

Reviewers' Comments:

Reviewer #1:

Remarks to the Author:

In this manuscript, Szczurek et al aim to identify the mechanism by which Ring1A/B containing PRC1s control transcription in embryonic stem cells (ESCs). Previous approaches to address this question have used measurements taken from bulk populations or fixed single-cells. However, averaged measurements from ensemble approaches may not accurately reflect the true state within any individual cell, and the kinetics of the dynamic and stochastic process of transcription cannot be captured in fixed cells. The authors impressively overcome these previous limitations by engineering a system to study nascent transcription in live cells, endogenously tagging a plethora of proteins, and by utilising live single-cell imaging and single particle tracking techniques. This allows them to interrogate several aspects of transcription in fine detail upon rapid removal of Ring1b in Ring1a KO ESCs

Overall, the manuscript is very well written and the figures are clear. The approaches and techniques used in this manuscript provide the resolution necessary to address the central question and make important advances on previous work. However, the authors should consider the following suggestions. Overall, because in Figure 5a, they show that Ring1b binds both to 'non-Polycomb' bound genes ('ncPRC1' bound genes?) and 'Polycomb' bound genes (Canonical PRC1 bound genes?), a major conceptual concern is that it is unclear if their new results and interpretations are relevant also to non-Polycomb bound genes? It's also not clear if their results and interpretations are relevant to some or all cPRC1 ("Polycomb bound") bound target genes. Some of the suggestions below should hopefully help towards addressing these concerns.

1. The authors have chosen two designated Polycomb target genes to study, namely Zic2 and E2f6.

Could they please comment on this choice, and in particular on how representative (or not) they are of all Polycomb target genes in mouse ESCs? They have some, albeit low, levels of transcription in ESCs, which might suggest they are not as strongly bound/repressed by Polycomb as more classic target genes such as the Hox loci. How would a HoxA/D gene look like in their live cell transcription analyses? Do Hox genes have comparable mRNA levels to Zic2 and E2F6 in ESCs?

2. Related to the first point, Extended Data Figure 1C shows genomic ChIP-seq tracks and RNA-seq signal. Could the authors provide equivalent snapshots for additional Polycomb target genes that are not typically even lowly transcribed in mouse ESCs – for example, the HoxD locus?

3. Figure 2C provides a helpful illustration of the Ring1b-AID degenon system. It would be helpful to make it clear on the figure and in the legend that these cells are also Ring1a-null.

4. Figures 4 and 5 show correlations between PRC1 and TFIID/PIC binding. To move beyond correlations, a key requirement would be that the authors provide more mechanistic insight for how canonical PRC1 or non-canonical PRC1/H2AK119Ub1, or both, might impair TFIID binding at gene promoters.

5. The authors show in Figure 5a that Ring1b binds at both 'non-Polycomb' and 'Polycomb' target TSSs. They write in the legend that "TSSs were segregated into non-Polycomb (n=9899), Polycomb (n=4869), and non-CpG islands (n=5869) groupings as indicated and ranked by RING1B signal". However, they should explain in the legend how this classification between "Polycomb" and "non-Polycomb" was made. Are the non-Polycomb target promoters ncPRC1 bound, but not cPRC1 bound? To help here, they should provide PHC1 and RYBP ChIP-seq data to delineate between the binding of discrete ncPRC1 and cPRC1 complexes in Figure 5a. They should also indicate on the Figure 5a plot the rank/position of their chosen E2f6 and Zic2 genes, as well as representative HoxA or HoxD genes, for comparison. Finally, it would also be helpful in Panel 5a to show the RNA-seq (mRNA level changes) in the UNT, dTAG-13, IAA and IAA and dTAG-13 treated ESCs (as heat maps, but separately as average plots of the 3 groupings; non-Polycomb, Polycomb and non-CpG island) to appreciate fully the mRNA levels of which genes change and which don't upon loss of Ring1b, loss of TAF1, and the combined loss (see more comments on this below).

6. Accompanying Figure 5, they should provide genomic cChIP-seq snapshots for TAF1 at Hspg2, Zic2, E2f6 and perhaps the HoxD locus, as well as including this data in Figure 5a with and without the various treatments for comparison.

7. It's not clear what is the importance or significance of the findings in Figure 5F. The experiment shows that degradation of TAF1 prevents derepression of Polycomb target genes upon removal Ring1b in Ring1a KO ESCs. However, the equivalent data for the control reference gene Hspg2 should also be shown. Given that, as is stated in the manuscript, "the TAF1 protein is integral to the formation of the TFIID holocomplex" and that TFIID is essential for PIC formation and transcription initiation, it seems likely that all or most transcription will be shut down upon TAF1 degradation. The suggestions in point 5 above should hopefully clarify this.

Reviewer #2:

Remarks to the Author:

In their study, Szczurek and colleagues investigate the mechanism by which the repressive PRC1 complex represses gene expression in single mouse Embryonic Stem Cells. Using a rapid degenon-mediated depletion of the RING1B subunit of the PRC1 complex and MS2-aptamer live-cell imaging of the transcriptional activity of 2 PRC1-target genes, E2f6 and Zic2, the authors demonstrate a functional role of the PRC1/H2AK119ub1 axis in the maintenance of deeply repressed transcription state at the promoters of those two genes. PRC1 complex appears to regulate the probability of

promoters to enter a permissive transcriptional state by inhibiting the formation of the Pre-Initiation Complex (PIC) in early transcription events, as demonstrated by single molecule tracking of multiple initiation and elongation factors endogenously labeled by HALO-tag. Finally, the authors extend their demonstration genome-wide by showing that the depletion of PRC1 in mESC leads to a decreased recruitment of the PIC factor TFIID onto PRC1-target genes, mostly, but also to a lesser extent to non-PRC1-target genes.

Overall this is an elegant study that provides mechanistic insight into how and why alleles transition into permissive states which can then initiate transcriptional bursts. Before publication, some comments should be addressed:

Major comments:

1. The authors nicely demonstrate that PRC1 degradation increases the probability of PRC1-target genes to enter permissive state. They also demonstrate that PRC1 degradation leads to an increased and/or more stable binding of some of the PIC components using single-molecule tracking. However, the authors do not directly link the binding of PIC components to the target genes. It is important to demonstrate that some of the most regulated factors (e.g. TBP, TAF11, MED14) indeed bind and are affected by PRC1 degradation at the *E2f6* and *Zic2* loci using CUT&TAG or ChIP-qPCR experiments.

2. In Figure 5a, the authors show that PRC1 degradation induces an increased binding of TAF1 genome-wide, and more specifically onto PRC1-target genes. However, the Log2FC is at maximum 0.4 (or 1.3-fold increase). How do the authors explain such a low enrichment when they see a change of 2-fold in the binding time of TAF1? What happens at *E2f6* and *Zic2* loci?

3. In their previous study (Dobrinic et al., 2021. Figure 7), the authors demonstrate that PRC1 regulates gene transcription by limiting transcription burst frequency, while in their current study, the authors prove that PRC1 does not alter the duration of the "time between ON-periods" or burst frequency (line 127 to line 137 and Figure 3a). Moreover, in their previous study the authors used a 2-state model while current study is based on a 3-state model. The authors should thus discuss the discrepancy with their previous study.

Minor comments:

1. Line 110 — Fig. 2A Its not clear how the bursting attributes are quantified. Please elaborate.
2. Its not clear to me why its surprising that Polycomb regulated genes when transitioned into activate states would behave any different than other genes. Several studies employing live cell imaging methods of transcription such as Wan et al (33979654) have shown that bursts sizes remain mainly consistent across several pathways and model organisms and primarily differ in burst frequency modulation. Perhaps this could be clarified in the text
3. Line 114, "polycomb-target genes" vs "polycomb genes" is confusing to the reader. Please clarify.
4. Lines 118-119, please mention that RING1B is a subunit of PRC1 that is targeted by the deproton system.
5. Fig 3D. Its interesting that only a small percentage of alleles become active compared upon PRC1 degradation to the control. Why aren't there more alleles active? Do all alleles become active eventually?
6. Figure 3E result is fantastic. Very clear that more alleles are active. However the fraction active doesn't seem to be similar to that quantified in Fig 3D. How is Fig. 3D quantified?
7. It would be good to show the rates of the model in 3F.
8. What is the fold change for 5b? Presumably this is statistically significant.

9. Extended Figure 1a, the PCR screening experiment reveals multiple bands for the MS2x128 condition for the 3 labeled genes. Please, explain the presence of multiple amplicons. Moreover, there is no amplification corresponding to a "WT"/endogenous band, does it mean that the cell lines generated are homozygous?
10. Extended Figure 1b, in the MS2-GFP channel, why are there multiple spots corresponding to Hspg2 transcription? Are there random integrants of MS2-containing construct? In that same experiment, please clarify the location of the intronic probes (not overlapping the LHA/RHA sequences)?
11. Extended Figure 5a, please, discuss the HALO-tagging of endogenous proteins. Are all the proteins homozygously tagged? Moreover, why does not MED14 WB contain a band for the endogenous proteins (contrarily to the other tagged protein-associated WB)?

Reviewer #3:

Remarks to the Author:

Summary of the key results: The polycomb repressive system consists of two highly conserved complexes, PRC1 and PRC2, which are essential for mammalian development. They repress genes by depositing chromatin marks at promoters containing CpG islands. However, how these marks affect downstream transcription kinetics is unknown. Previous work has focused on ensemble measurements, making it difficult to infer the precise steps PRC1/2 act on, because genes transcribe out of sync across cells.

The paper focuses on the action of PRC1, which mono-ubiquitylates H2A. The authors measure the changes in transcription upon acute depletion of the catalytic subunit of PRC1, RING1B using smFISH and knock-in MS2 repeats into endogenous PRC1-responsive genes, enabling quantification of transcription bursting kinetics. These experiments show that PRC depletion does not affect bursting dynamics while the genes are in a permissive state, but rather increases the time spent in the transcriptional permissive states. A simple stochastic model recapitulates the difference in total transcripts produced, predicting that PRC1 represses the entry into the permissive state. The authors support their findings using live-cell single-molecule tracking of a suite of general transcription factors in the presence or absence of PRC1, and find that PRC1 primarily influences the "early" PIC components by decreasing the bound fraction and stable binding time of these proteins. This result agrees well with the findings from MS2 experiments in which PRC1 influences the time these genes exist in a transcriptionally permissive state.

Originality and significance: The experiments are well-designed to specifically address the gap in our understanding of Polycomb repression, clearly outlining which step of the transcription cycle is impacted by PRC1 (de-repression, via inhibiting PIC assembly). These results nicely lay out what PRC1 can, and cannot do quantitatively to transcription regulation, which will inform models of expression dynamics during development. This work also identifies an inhibition of the early PIC, which will likely inspire follow up work to articulate the precise molecular intermediates of this competition.

Data & methodology: I appreciate the authors using endogenous genes to record transcriptional activity, and acute depletions that rapidly change the transcriptional state of the cell, because it provides access to direct effects of PRC depletion on relevant target genes. The suite of PIC factors measured by SMT is impressive.

Appropriate use of statistics and treatment of uncertainties: the data is rigorously acquired and analyzed. One note is that the authors chose to only analyze bursts of >2.5 transcripts. Small bursts

(under the 2.5 threshold transcript) seem equally frequent, if not more frequent than larger bursts in the time traces displayed. How sensitive are the conclusions of the modeling to the choice of that threshold? E.g. what are the conclusions when a threshold of 1 transcript or more is chosen?

Conclusions: Conclusions are well supported, based on a gold standard model system, the analysis of two PRC1-target and one control gene, and the analysis of several PIC components.

Clarity and context/references The premise of the paper is well supported by the literature, and the manuscript is well written and easy to understand for a broad audience.

Suggested improvements:

There is an apparent discrepancy in the effect sizes upon PRC1 depletion of the RNA copy number compared to that of the changes in bursting dynamics. For instance, *Zic2* jumps ~ 2.5 fold (from 15 to 38 RNAs per cell, Fig. 2d), which should in principle be explained by a doubling of the bursting output. Yet the sole bursting parameter that changes upon PRC depletion (fraction of time in permissive period) increases by 2 fold only. More so for *E2f6*, which RNA levels increase by 2 fold, but for which the only significant change, the increase in permissive fraction is only $\sim 30\%$. At equilibrium, the transcription output is expected to be proportional to the fraction of time spent in the active state (see eg. formula 2.9 in <https://www.ncbi.nlm.nih.gov/pmc/articles/PMC10073913/>), so this discrepancy is surprising. The first factor that comes to mind is the non-equilibrium nature of the experiment: an acute depletion followed by recording of dynamics over a time window shorter than some lifetimes in the system (the repressive state lasts several cell cycles). This is not a critique of this experimental design which is absolutely necessary to avoid secondary effects. But since there are conflicting factors at play in the dynamics, could the authors explain the reason for the apparent effect size mismatch, and provide an intuitive reason to the reader? For instance, RNA levels will lag at a high level due to RNA lifetimes in the hr range; on the other hand, incomplete re-entry into the permissive state during the experimental window might lead to lower RNA counts than expected at equilibrium.

Related to the previous point. Assuming the bursting metrics reflect the 'true' effect size of PRC1 repression in the unperturbed state (rather than the RNA levels which reflect the dynamic response of the system upon acute perturbation), the impact of PRC1 on expression seems modest (30% to 2-fold), which came a bit as a surprise, because PRC repression is often presented in the literature as an on/off switch, but these results would suggest that it is a much more subtle regulatory lever. How do the genes studied here compare to the rest of PCR-regulated genes? Are they "representative examples" of PRC repression, or weak responders? If they are representative, the present findings would suggest that PRC repression alone is unlikely to generate the kind of on/off switches observed during development, and maybe just provides a robustness against weak/off target activation, as the authors discuss. It would be interesting for the authors to comment on the relevance of PRC1 'strength' for its developmental role.

The value of $P_{o>p}$ for *Zic2* IAA is 0.54 in Fig 3f, which does not seem to coincide with the position of the minimum in Extended data Fig. 4g (bottom left panel). Can the authors clarify or correct?

"When we examined the dynamics of other TFIID components, TAF11 showed an increased bound fraction whereas TAF1 was unaffected but both factors displayed increases in stable binding time." This is curious, how might this be possible? If the stable binding time is increased by a factor of 2, surely that would influence the bound fraction of TAF1? Unless the TAF1 on-rate onto chromatin is reduced by the same factor?

"We then depleted either PRC1 or PRC1 and TAF1 simultaneously and examined the expression of the Zic2 and E2f6 Polycomb target genes using smRNA-FISH (Fig. 5E, F). Importantly, this revealed that neither Polycomb target gene was derepressed in the absence of TAF1, suggesting that TFIID binding enables elevated expression in the absence of PRC1/H2AK119ub1."

Depletion of TAF1 does not completely abolish transcription, and does not seem to have any substantial effect on E2f6 and Zic2 when PRC1 is present (Fig 5F). This seems strange as it is a basal transcription factor. Could the authors comment on why they think this is the case?

"Interestingly, we also observed a modest yet significant increase in TAF1 binding across non-Polycomb enriched transcription start sites, indicating that PRC1 may constrain the binding of TFIID more broadly (Fig. 5B and Extended Data Fig. 6B) Consistent with this possibility, low levels of PRC1 are detected at non-Polycomb gene promoters, and when we analysed gene expression across these genes we also observed a modest increase in expression after PRC1 depletion (Extended Data Fig. 6A)"

Is there a positive correlation between the increase of TAF1 at promoters and an increase in logFC of expression for these genes? The Extended Data Fig. 6 shows the distribution of these parameters in bulk, but did the authors consider plotting these against each other in a scatter plot?

"Based on these detailed kinetic measurements, we find that PRC1/H2AK119ub1 limits the binding of factors involved in the earliest stages of PIC 229 formation (Fig. 4E)."

The interpretation is consistent with the data, but this statement seems too strong without any corresponding in vitro data to support this claim, though such studies are beyond the scope of this study. The ChIP data is certainly helpful towards making this conclusion, but secondary effects of the removal of PRC1 perturbation cannot be conclusively ruled out in the context of a live cell. Perhaps the authors could slightly soften some of the language surrounding this claim, and make clear these potential caveats?

Methods should be written concisely, but should contain all elements necessary to allow interpretation

and replication of the results. As a guideline, Methods sections typically do not exceed 3,000 words. The Methods should be divided into subsections listing reagents and techniques. When citing previous methods, accurate references should be provided and any alterations should be noted. Information must be provided about: antibody dilutions, company names, catalogue numbers and clone numbers for monoclonal antibodies; sequences of RNAi and cDNA probes/primers or company names and catalogue numbers if reagents are commercial; cell line names, sources and information on cell line identity and authentication. Animal studies and experiments involving human subjects must be reported in detail, identifying the committees approving the protocols. For studies involving human subjects/samples, a statement must be included confirming that informed consent was obtained. Statistical analyses and information on the reproducibility of experimental results should be provided in a section titled "Statistics and Reproducibility".

All Nature Cell Biology manuscripts submitted on or after March 21 2016 must include a Data availability statement as a separate section after Methods but before references, under the heading "Data Availability". For Springer Nature policies on data availability see <http://www.nature.com/authors/policies/availability.html>; for more information on this particular policy see <http://www.nature.com/authors/policies/data/data-availability-statements-data-citations.pdf>. The Data availability statement should include:

- Accession codes for primary datasets (generated during the study under consideration and designated as "primary accessions") and secondary datasets (published datasets reanalysed during the study under consideration, designated as "referenced accessions"). For primary accessions data should be made public to coincide with publication of the manuscript. A list of data types for which submission to community-endorsed public repositories is mandated (including sequence, structure, microarray, deep sequencing data) can be found here <http://www.nature.com/authors/policies/availability.html#data>.
- Unique identifiers (accession codes, DOIs or other unique persistent identifier) and hyperlinks for datasets deposited in an approved repository, but for which data deposition is not mandated (see here for details <http://www.nature.com/sdata/data-policies/repositories>).
- At a minimum, please include a statement confirming that all relevant data are available from the authors, and/or are included with the manuscript (e.g. as source data or supplementary information), listing which data are included (e.g. by figure panels and data types) and mentioning any restrictions on availability.
- If a dataset has a Digital Object Identifier (DOI) as its unique identifier, we strongly encourage including this in the Reference list and citing the dataset in the Methods.

We recommend that you upload the step-by-step protocols used in this manuscript to the Protocol Exchange. More details can be found at www.nature.com/protocolexchange/about.

FIGURES – Colour figure publication costs \$600 for the first, and \$300 for each subsequent colour

figure. All panels of a multi-panel figure must be logically connected and arranged as they would appear in the final version. Unnecessary figures and figure panels should be avoided (e.g. data presented in small tables could be stated briefly in the text instead).

All imaging data should be accompanied by scale bars, which should be defined in the legend. Cropped images of gels/blots are acceptable, but need to be accompanied by size markers, and to retain visible background signal within the linear range (i.e. should not be saturated). The boundaries of panels with low background have to be demarked with black lines. Splicing of panels should only be considered if unavoidable, and must be clearly marked on the figure, and noted in the legend with a statement on whether the samples were obtained and processed simultaneously. Quantitative comparisons between samples on different gels/blots are discouraged; if this is unavoidable, it should only be performed for samples derived from the same experiment with gels/blots were processed in parallel, which needs to be stated in the legend.

Regardless of format, all figures must be vector graphic compatible files, not supplied in a flattened raster/bitmap graphics format, but should be fully editable, allowing us to highlight/copy/paste all text and move individual parts of the figures (i.e. arrows, lines, x and y axes, graphs, tick marks, scale

bars etc.). The only parts of the figure that should be in pixel raster/bitmap format are photographic images or 3D rendered graphics/complex technical illustrations.

The total number of Supplementary Figures (not including the “unprocessed scans” Supplementary Figure) should not exceed the number of main display items (figures and/or tables (see our Guide to Authors and March 2012 editorial <http://www.nature.com/ncb/authors/submit/index.html#suppinfo>; <http://www.nature.com/ncb/journal/v14/n3/index.html#ed>). No restrictions apply to Supplementary Tables or Videos, but we advise authors to be selective in including supplemental data.

Each Supplementary Figure should be provided as a single page and as an individual file in one of our accepted figure formats and should be presented according to our figure guidelines (see above). Supplementary Tables should be provided as individual Excel files. Supplementary Videos should be

provided as .avi or .mov files up to 50 MB in size. Supplementary Figures, Tables and Videos must be accompanied by a separate Word document including titles and legends.

GUIDELINES FOR EXPERIMENTAL AND STATISTICAL REPORTING

REPORTING REQUIREMENTS – We are trying to improve the quality of methods and statistics reporting in our papers. To that end, we are now asking authors to complete a reporting summary that collects information on experimental design and reagents. The Reporting Summary can be found here <https://www.nature.com/documents/nr-reporting-summary.pdf> If you would like to reference the guidance text as you complete the template, please access these flattened versions at <http://www.nature.com/authors/policies/availability.html>.

Author Rebuttal to Initial comments

We would like to thank the reviewers for their detailed consideration of our manuscript and very helpful comments. We were pleased that the reviewers found that our study has *'impressively overcome ... previous limitations'* and provided *'the resolution necessary to address the central question and make important advances on previous work'* (Reviewer 1), represents *'an elegant study that provides mechanistic insight into how and why alleles transition into permissive states which can then initiate transcriptional bursts'* (Reviewer 2), and finally that our *'conclusions are well supported, based on a gold standard model system'* and *'the manuscript is well written and easy to understand for a broad audience'* (Reviewer 3).

Below we have provided a point-by-point response to the reviewers' constructive comments and described how we have generated new cell lines and carried out extensive new experimentation and analysis to address these comments. We have also detailed how we have integrated these interesting new findings into the improved and revised manuscript. In all cases the reviewer comments are in black text and our responses in blue text.

Reviewers' Comments:

Reviewer #1:

Remarks to the Author:

In this manuscript, Szczurek et al aim to identify the mechanism by which Ring1A/B containing PRC1s control transcription in embryonic stem cells (ESCs). Previous approaches to address this question have used measurements taken from bulk populations or fixed single-cells. However, averaged measurements from ensemble approaches may not accurately reflect the true state within any individual cell, and the kinetics of the dynamic and stochastic process of transcription cannot be captured in fixed cells. The authors impressively overcome these previous limitations by engineering a system to study nascent transcription in live cells, endogenously tagging a plethora of proteins, and by utilising live single-cell imaging and single particle tracking techniques. This allows them to interrogate several aspects of transcription in fine detail upon rapid removal of Ring1b in Ring1a KO ESCs

Overall, the manuscript is very well written and the figures are clear. The approaches and techniques used in this manuscript provide the resolution necessary to address the central question and make important advances on previous work.

We thank the reviewer for their careful consideration of our manuscript and for highlighting how 'the new approaches and techniques used in this manuscript provide the resolution necessary to address the central question and make important advances on previous work.'

However, the authors should consider the following suggestions. Overall, because in Figure 5a, they show that Ring1b binds both to 'non-Polycomb' bound genes ('ncPRC1' bound genes?) and 'Polycomb' bound genes (Canonical PRC1 bound genes?), a major conceptual concern is that it is unclear if their new results and interpretations are relevant also to non-Polycomb bound genes? It's also not clear if their results and interpretations are relevant to some or all cPRC1 ("Polycomb bound") bound target genes. Some of the suggestions below should hopefully help towards addressing these concerns.

We have provided a detailed response to these reviewer suggestions throughout our point-by-point responses to the reviewer comments below. However, we considered it poignant to also discuss at a

very top level these two important suggestions here to clarify how our original findings and new experiments now address these points:

Point A - 'Overall, because in Figure 5a, they show that Ring1b binds both to 'non-Polycomb' bound genes ('ncPRC1' bound genes?) and 'Polycomb' bound genes (Canonical PRC1 bound genes?), a major conceptual concern is that it is unclear if their new results and interpretations are relevant also to non-Polycomb bound genes?'

We thank the reviewer for bringing up this important point. By way of clarification, we and others have previously demonstrated that low-level PRC1 (RING1b) occupancy is detected by ChIP-seq across virtually all CpG island-associated gene promoters¹ and that this relies on non-canonical PRC1 (ncPRC1) complexes sampling non-methylated CpG DNA (e.g. via ncPRC1.1) or other DNA motifs (e.g. via ncPRC1.6) within promoter regions. These sampling activities appear to be important for identifying lowly transcribed or inactive CpG island-associated gene promoters enabling the formation of Polycomb chromatin domains which, in contrast to non-Polycomb gene promoters, are characterised by high-level occupancy of canonical PRC1 (cPRC1), non-canonical PRC1 (ncPRC1), and H2AK119ub1 (referred to as 'Polycomb' genes -see response to point 5 below and illustrated in new Extended Data Fig. 6a)²⁻⁹. Depletion of PRC1 causes a pronounced derepression of Polycomb genes (see new Fig. 6a, Extended Data Fig. 7c, and^{5,6}) and the largest effects on TAF1 binding (Fig. 6a), consistent with Polycomb chromatin domains limiting TFIID binding to maintain the repression of Polycomb genes. Interestingly, we have also previously observed very subtle effects on the expression of non-Polycomb genes when PRC1 is depleted (also see new Fig. 6a, Ext. Data Fig. 6a, 7c)¹⁰, presumably due to a minor influence of the sampling form of ncPRC1 complexes and we now also observe a smaller but widespread effect on binding of TAF1 (TFIID) across non-Polycomb gene promoters (Fig. 6a, Ext. Data. Fig. 7b,e). Therefore, we conclude that PRC1 primarily influences transcription by counteracting TFIID engagement, with the effects being most pronounced at Polycomb genes where Polycomb chromatin domains play an important role in maintaining gene repression. To ensure this important point is clear to the reader we have drawn attention to this on lines 293-296 of the revised manuscript.

Point B - 'It's also not clear if their results and interpretations are relevant to some or all cPRC1 ("Polycomb bound") bound target genes'

Polycomb chromatin domain features and the effects we observe on derepression and TAF1 (TFIID) binding after PRC1 depletion are very similar across Polycomb genes (see detailed response to point 1 and point 5). The only clear difference we observe across Polycomb genes is that the absolute level of transcript resulting from derepression following PRC1 depletion can vary depending on what appears to be the underlying level of activation signal that individual Polycomb genes are subject to (see response to point 2 and reviewer Fig. 1). Furthermore, in new experiments we now show that cPRC1 has little effect on stable PIC binding or derepression of Polycomb genes, demonstrating that ncPRC1 is the key determinant in Polycomb chromatin domain-mediated repression (new Fig. 5, Extended Data Fig. 6). This extends our mechanistic understanding of Polycomb repression and illustrates that this mechanism of repression appears to work across Polycomb genes to maintain gene inactivity.

We thank the reviewer for bringing up these interesting and important suggestions as they have been very helpful in guiding our new work that provides further mechanistic insight and has clarified several of our results regarding transcription control by PRC1 across different gene types (Polycomb vs non-Polycomb genes). We have elaborated on how we have addressed these and other suggestions in our

point-by-point response to the reviewers' comments and how we have integrated our new revisions and findings into the extended and improved manuscript.

1. The authors have chosen two designated Polycomb target genes to study, namely *Zic2* and *E2f6*. Could they please comment on this choice, and in particular on how representative (or not) they are of all Polycomb target genes in mouse ESCs?

In retrospect we realise we could have more clearly explained our logic for selecting Polycomb genes for live-cell transcriptional imaging. To select Polycomb genes we drew on our previously published genomics-based chromatin analysis and single-molecule RNA-FISH (smRNA-FISH) measurements of gene expression in the PRC1-degron cell line (Dobrinic et al, NSMB, 2021)¹⁰. We used this information to select Polycomb genes for live-cell transcription imaging that (1) had typical Polycomb chromatin domain features (high levels of PRC1, PRC2, H2AK119ub1, and H3K27me3 based on ChIP-seq) and (2) that displayed increased expression after PRC1 depletion (RNA-seq and smRNA-FISH) (Fig. 2d, Extended Data Fig. 1c). In order to characterise how PRC1 regulates transcription we needed to be able to compare transcription before and after PRC1 depletion. This necessitated we focus on genes that had a level of pre-existing activation signal sufficient to drive at least some low-level transcription which is constrained by PRC1 in the unperturbed state, so that we could quantitate and compare features of transcription before and after PRC1 depletion. We have now edited the main text of the revised manuscript to clearly state the logic that guided our selection of Polycomb target genes as follows on lines 85-92:

*'To implement this, we used CRISPR-Cas9 engineering in mouse ESCs to create lines where MS2 repeats were inserted into the first intron of two representative Polycomb genes (*Zic2* and *E2f6*) that have their promoters embedded within a typical Polycomb chromatin domain (Extended Data Fig. 1a-d) and which are subject to a very low level of transcription in wild type cells but become derepressed when the Polycomb system is depleted (Extended Data Fig. 1c). We also engineered MS2 repeats into a moderately expressed reference gene that lacks a discernible Polycomb chromatin domain (*Hspg2*) and is not influenced by Polycomb repression (Extended Data Fig. 1b-d).'*

They have some, albeit low, levels of transcription in ESCs, which might suggest they are not as strongly bound/repressed by Polycomb as more classic target genes such as the *Hox* loci. Do *Hox* genes have comparable mRNA levels to *Zic2* and *E2f6* in ESCs?

The reviewer raises two important queries as to whether *Zic2* and *E2f6* differ from what the reviewer refers to as 'classic' Polycomb target genes, including the *Hox* genes. We have now explored these two points in detail and respond to each in turn:

(1) 'not as strongly bound'- As evident from ChIP-seq snapshots in new Extended Data Fig. 1c, *E2f6* and *Zic2* have levels of PRC1/2 and H2AK119ub1/H3K27me3 comparable to the levels at other Polycomb target genes including the *HoxD* locus, *HoxA7* (a representative *Hox* gene), and *Meis1*, which are often considered 'classic' Polycomb target genes based on their identification in genetic assays. As indicated by the reviewer, *E2f6* and *Zic2* have some, albeit low, levels of transcription in the unperturbed state, and as described above this was a key requirement for dissecting how PRC1 regulates transcription using live-cell transcription imaging.

(2) ‘not as strongly repressed’- We do not believe that *E2f6* and *Zic2* are less ‘strongly repressed’ by their Polycomb chromatin domain. Instead, compared to *Meis1* and *HoxA7*, they simply appear to have higher levels of pre-existing activation signal. This is evident when we quantify the absolute number of transcripts (smRNA-FISH) in the unperturbed and PRC1-depleted state for *E2f6* and *Zic2* and compare this to *Meis1* and *HoxA7* (Reviewer Fig. 1a below). This shows that *Meis1* and *HoxA7* have lower transcript numbers in the unperturbed state, but also yield significantly lower transcript numbers when PRC1 is depleted compared to *E2f6* and *Zic2* (Reviewer Fig. 1a). However, the fold change in transcript number (Reviewer Fig. 1b) is similar across *Meis1*, *HoxA7*, *E2f6*, and *Zic2*, consistent with the level of activation signal, as opposed to repressive capacity of the Polycomb chromatin domain, dictating expression in each context.

Reviewer Figure 1 – A comparison of transcript number with fold changes in transcript numbers after PRC1 depletion.

(a) Bar plots indicating the average transcripts per cell as measured by smRNA-FISH for the indicated genes in untreated (UNT) and 4 hours after PRC1 depletion (IAA). The measurements represent the average of 3 biological replicates with standard error.
 (b) As per (a) but illustrating the fold change in average transcripts per cell.

These analyses further demonstrate that *E2f6* and *Zic2* are ‘representative’ Polycomb genes and importantly are subject to sufficient pre-existing transcription signals that allow us to study PRC1-mediated transcription regulation using live-cell imaging.

How would a *HoxA/D* gene look like in their live cell transcription analyses?

We purposely did not attempt to carry out live-cell transcription imaging analysis of genes in the *Hox* loci. The reason for this is two-fold. Firstly, *Hox* genes are situated in complex atypical multi-gene clusters which include a number of non-coding RNAs. This makes engineering these loci in a manner where we can ensure transcript-specific analysis challenging. Secondly, and more importantly, as explained in detail above, in order to study transcriptional control by PRC1 we must be able to examine, quantify, and compare Polycomb gene transcription before and after PRC1 depletion. Based on smRNA-FISH we have shown that the expression levels of *Hox* genes and other ‘classical’ Polycomb genes like *Meis1* can be extremely low in the unperturbed state and although they display elevated transcript levels after PRC1 depletion, their expression still remains extremely low, consistent with them being subject to extremely low levels of pre-existing activation signal (as illustrated in Reviewer Fig. 1). As described above, genes with low levels of pre-existing activation signal are not suitable for

detailed quantification, analysis, and comparison of live cell transcription imaging data. To further demonstrate this point, we have now engineered a new transcriptional imaging cell line in response to the reviewer’s query, where we inserted the MS2 array into the first intron of the *Meis1* gene (which has a similar expression level and Polycomb domain features to the *Hox* gene, Reviewer Fig. 1, Extended Data Fig. 1c), and carried out live-cell transcription imaging. Despite imaging cells for long periods of time (8 hours) in many cells, we very rarely observed transcriptional events in the unperturbed state, meaning any analysis of these events would be underpowered and extremely difficult to compare the untreated to the PRC1-depleted state. This limitation is not because we can’t technically capture the transcription of *Meis1*, as treatment with retinoic acid, which causes *Meis1* activation, leads to frequent transcriptional events (Reviewer Fig. 2). These results are now included in the new Extended Data Fig. 8.

Reviewer Figure 2. Gene expression and transcription analysis of *Meis1* upon PRC1 depletion or activation.

(a) mRNA-FISH analysis of *Meis1*. Average transcript per cell numbers for *Meis1* (UNT), after PRC1 depletion (4h IAA), and after activation using 72h retinoic acid (RA) in absence of leukemia inhibitory factor (72h RA). Individual dots represent biological replicates (n=3).

(b) Live-cell transcription imaging of *Meis1*. In untreated (UNT), PRC1 depleted (IAA) and activated state (after 72 RA treatment). 141 cells per each condition are presented in the heat-map.

In summary, we conclude that the Polycomb target genes *Zic2* and *E2f6* are ‘representative’ of typical Polycomb genes and suitable for live-cell transcription imaging as they have sufficient pre-existing levels of activation signal to dissect and quantitate PRC1 mediated transcription control.

2. Related to the first point, Extended Data Figure 1C shows genomic ChIP-seq tracks and RNA-seq signal. Could the authors provide equivalent snapshots for additional Polycomb target genes that are not typically even lowly transcribed in mouse ESCs – for example, the *HoxD* locus?

As requested by the reviewer, we have now provided ChIP-seq and RNA-seq tracks for the *HoxD* locus, *HoxA7*, and *Meis1* for comparison to *E2f6* and *Zic2* (new Extended Data Fig. 1c) This illustrates that *E2f6* and *Zic2* share Polycomb chromatin domain features with ‘classical’ Polycomb target genes, but also have sufficient underlying transcription signals to observe and quantify transcription in live-cell imaging (as described in detail in response to point 1).

3. Figure 2C provides a helpful illustration of the Ring1b-AID degron system. It would be helpful to make it clear on the figure and in the legend that these cells are also Ring1a-null.

As suggested by the reviewer we have now updated this in both the main Fig. 2c and the figure legend.

4. Figures 4 and 5 show correlations between PRC1 and TFIID/PIC binding. To move beyond correlations, a key requirement would be that the authors provide more mechanistic insight for how canonical PRC1 or non-canonical PRC1/H2AK119Ub1, or both, might impair TFIID binding at gene promoters.

As requested by the reviewer, and to provide more mechanistic insight, we have carried out additional experiments to understand the contribution of canonical PRC1 (cPRC1) and non-canonical PRC1 (ncPRC1) to TFIID binding and gene expression. To achieve this, we have generated new degron cell lines where we can rapidly and specifically deplete cPRC1 by degrading its core structural component, PCGF2, that subsequently leads to instantaneous decreases in levels of other cPRC1 component PHC1 (new Fig. 5). After depleting cPRC1 and analysing TFIID chromatin-association by single particle tracking, we observed an increase in the bound fraction of TFIID, suggesting that cPRC1 can regulate the dynamic interactions that TFIID makes with chromatin, thus influencing the total bound fraction. However, more interestingly, and in contrast to removing all PRC1 complexes, depletion of cPRC1 did not affect the stable binding time of TFIID as measured by single particle tracking, nor did it lead to Polycomb target gene derepression as quantified by smRNA-FISH and qRT-PCR (new Extended Data Fig. 6). Therefore, ncPRC1 appears to be the key determinant in regulating stable TFIID binding events and is key to maintaining gene repression. We have now illustrated these results in new Fig. 5 of the revised manuscript and discuss these important findings in a new paragraph of the main text on lines 238-269 of the revised manuscript and in lines 339-342 of the discussion. We thank that reviewer for suggesting these important new experiments. They have significantly extended our mechanistic insight and very nicely reveal that it is ncPRC1, as opposed to cPRC1, is the primary determinant in regulating stable TFIID binding and transcriptional repression.

5. The authors show in Figure 5a that Ring1b binds at both 'non-Polycomb' and 'Polycomb' target TSSs. They write in the legend that "TSSs were segregated into non-Polycomb (n=9899), Polycomb (n=4869), and non-CpG islands (n=5869) groupings as indicated and ranked by RING1B signal". However, they should explain in the legend how this classification between "Polycomb" and "non-Polycomb" was made.

We thank the reviewer for highlighting this oversight. These TSSs and associated genes were unbiasedly classified based on our previous detailed genomic dissection of Polycomb system occupancy in ESCs⁶. However, we did not appropriately cite this in the figure legend or materials and methods.

For the benefit of the reviewer, in Fursova et al, mouse TSSs/genes from a custom-build non-redundant gene set (n = 20,633) were classified into three distinct gene categories:

(1) Polycomb- These are TSS that overlap with a non-methylated CpG islands (NMIs/CGIs defined in Long et al, 2013)¹¹ and are also bound by high levels of both PRC1 (RING1B) and PRC2 (SUZ12) based on ChIP-seq analysis and peak calling⁶.

(2) non-Polycomb- These are TSSs that overlap with a non-methylated CpG island but are not bound by high levels of PRC1 and PRC2.

(3) non-CpG island- These are TSSs that did not overlap with a non-methylated CpG island.

To ensure the origin of these classifications is clear to the reader, we now drawn attention to Fursova et al by citing this in the figure legend for Figure 6a and the materials and methods section of the revised manuscript. We have further included the exact gene annotation for each group in the Source Data File for Figure 6a.

Are the non-Polycomb target promoters ncPRC1 bound, but not cPRC1 bound? To help here, they should provide PHC1 and RYBP ChIP-seq data to delineate between the binding of discrete ncPRC1 and cPRC1 complexes in Figure 5a. They should also indicate on the Figure 5a plot the rank/position of their chosen *E2f6* and *Zic2* genes, as well as representative *HoxA* or *HoxD* genes, for comparison.

As suggested by the reviewer, we have now created a heatmap to show PHC1 (cPRC1) and RYBP (ncPRC1) binding profiles across our gene TSS classifications (Extended Data Fig. 6a) and also shown the position of *E2f6*, *Zic2*, *Meis1*, *Hoxa/d genes*, and *Hspg2* in Fig. 6a (formerly Fig. 5). As is evident from this new analysis (and previous work from our group and others) RYBP is most highly enriched at Polycomb gene TSSs, but also binds at lower levels across non-Polycomb gene TSSs. We have previously shown that this broad enrichment of RYBP across non-Polycomb gene TSSs is primarily accounted for by the ncPRC1.1 and ncPRC1.6 complexes that contain DNA binding domains that allow them dynamically sample CGIs and gene promoters^{1,3,6,12}. These profiles are in line with the unbiased classification based on genomic analysis⁶ of genes into Polycomb, non-Polycomb, and non-CpG island as described in detail above.

Finally, it would also be helpful in Panel 5a to show the RNA-seq (mRNA level changes) in the UNT, dTAG-13, IAA and IAA and dTAG-13 treated ESCs (as heat maps, but separately as average plots of the 3 groupings; non-Polycomb, Polycomb and non-CpG island) to appreciate fully the mRNA levels of which genes change and which don't upon loss of Ring1b, loss of TAF1, and the combined loss (see more comments on this below).

We thank the reviewer for this suggestion, but we have purposely not attempted to carry out the suggested RNA-seq analyses following TAF1 depletion due to an important technical limitation inherent to such experiments. The limitation is that for most expressed genes the half-life of their respective mRNAs is longer than the experimental time-frame of our rapid perturbation experiments¹³. Therefore, we would not be able to accurately capture the effects on the mRNA levels that result from the depletion of TAF1 as this would be conflated with the decay of existing mRNA produced before the depletion. Therefore, we chose instead to focus specifically on the Polycomb genes (*E2f6* and *Zic2*) for which we have detailed transcription information and used our highly sensitive smRNA-FISH approach to understand the requirements for their derepression. These more targeted analyses are possible because Polycomb genes have very low levels of mRNA in the unperturbed context and smRNA-FISH allows us to accurately measure the accumulation of new mRNAs following our rapid perturbations. Importantly, these findings show that derepression of Polycomb genes correspond to the elevated binding of TFIID (Extended Data Fig. 7e) and that TFIID is required for this derepression (Figure 6f).

6. Accompanying Figure 5, they should provide genomic cChIP-seq snapshots for TAF1 at *Hspg2*, *Zic2*, *E2f6* and perhaps the *HoxD* locus, as well as including this data in Figure 5a with and without the various treatments for comparison.

As suggested by the reviewer, we have now included TAF1 ChIP-seq snapshots for *Zic2*, *E2f6*, *Hspg2* and also *Meis1*, *HoxD* locus, *Hoxa7*, and *Brd2* in untreated (UNT) and Ring1B depleted (IAA) conditions in Extended Data. Fig. 7d.

7. It's not clear what is the importance or significance of the findings in Figure 5F. The experiment shows that degradation of TAF1 prevents derepression of Polycomb target genes upon removal Ring1b in Ring1a KO ESCs. However, the equivalent data for the control reference gene *Hspg2* should also be shown. Given that, as is stated in the manuscript, "the TAF1 protein is integral to the formation of the TFIID holocomplex" and that TFIID is essential for PIC formation and transcription initiation, it seems likely that all or most transcription will be shut down upon TAF1 degradation. The suggestions in point 5 above should hopefully clarify this.

The experiment in Figure 6f (former Fig. 5f) is important and significant because it answers (1) whether Polycomb gene derepression requires new initiation and (2) whether this relies on TFIID. We have elaborated on this importance and significance of these two points as follows:

(1) A number of distinct models for Polycomb-mediated repression have been proposed ranging from counteracting transcription initiation¹⁴ to controlling the release of an initiated but stably paused polymerase¹⁵ (reviewed in²). Our new SPT and ChIP-seq discoveries demonstrate that depletion of PRC1 causes a rapid and elevated binding of TFIID and entry into a transcriptionally permissive state. To test whether TFIID-dependent initiation events are required for Polycomb gene derepression, we simultaneously depleted TFIID and PRC1 and then examined gene expression. This revealed that TFIID and new initiation events, as opposed to release of an initiated and stably paused polymerase, is absolutely required for Polycomb gene derepression. This is important and significant as it supports the conclusion that PRC1 limits the earliest steps of transcription initiation to maintain gene repression.

(2) Although the textbook view is that TFIID is central to transcription initiation, there is an ongoing debate as to whether all initiation events rely on TFIID¹⁶⁻²⁴. Our SPT and ChIP-seq demonstrated increased TFIID binding was coincident with Polycomb gene derepression, suggesting that TFIID binding may be required for the observed derepression after PRC1 depletion. Therefore, to definitively test whether Polycomb repression relies on counteracting a canonical TFIID-dependent initiation mechanism, we simultaneously depleted TFIID and PRC1 and then examined gene expression. This revealed that TFIID is indeed essential for Polycomb gene derepression, consistent with PRC1 counteracting canonical TFIID-dependent transcription initiation (lines 306-309).

Finally, we have now carried out new smRNA-FISH analysis for *Hspg2* to complement *E2f6* and *Zic2* in Fig. 6f as requested by the reviewer.

We thank the reviewer for raising this point as it has allowed us to ensure the significance and importance of this experiment is clear to the reader in the main text of the revised manuscript.

Reviewer #2:

Remarks to the Author:

In their study, Szczurek and colleagues investigate the mechanism by which the repressive PRC1 complex represses gene expression in single mouse Embryonic Stem Cells. Using a rapid degran-mediated depletion of the RING1B subunit of the PRC1 complex and MS2-aptamer live-cell imaging of the transcriptional activity of 2 PRC1-target genes, *E2f6* and *Zic2*, the authors demonstrate a functional role of the PRC1/H2AK119ub1 axis in the maintenance of deeply repressed transcription state at the promoters of those two genes. PRC1 complex appears to regulate the probability of promoters to enter a permissive transcriptional state by inhibiting the formation of the Pre-Initiation Complex (PIC)

in early transcription events, as demonstrated by single molecule tracking of multiple initiation and elongation factors endogenously labeled by HALO-tag. Finally, the authors extend their demonstration genome-wide by showing that the depletion of PRC1 in mESC leads to a decreased recruitment of the PIC factor TFIID onto PRC1-target genes, mostly, but also to a lesser extent to non-PRC1-target genes.

Overall this is an elegant study that provides mechanistic insight into how and why alleles transition into permissive states which can then initiate transcriptional bursts. Before publication, some comments should be addressed:

We thank the reviewer for their careful consideration of our study and supportive comments about our approach to studying this important problem and the new mechanistic insight we glean. We have responded point-by-point to the reviewer's helpful comments below and note that addressing these has significantly strengthened the revised manuscript.

Major comments:

1. The authors nicely demonstrate that PRC1 degradation increases the probability of PRC1-target genes to enter permissive state. They also demonstrate that PRC1 degradation leads to an increased and/or more stable binding of some of the PIC components using single-molecule tracking. However, the authors do not directly link the binding of PIC components to the target genes. It is important to demonstrate that some of the most regulated factors (e.g. TBP, TAF11, MED14) indeed bind and are affected by PRC1 degradation at the *E2f6* and *Zic2* loci using CUT&TAG or ChIP-qPCR experiments.

We agree that it would be useful to examine how the binding of other factors affected in our SPT analysis are influenced at *E2f6* and *Zic2* by ChIP-qPCR. As suggested by the reviewer, we have now carried new ChIP-qPCR analysis for the suggested set of regulated factors including TAF1, TAF11, and MED14 at several Polycomb genes (including *E2f6* and *Zic2*) and a control gene. Unfortunately, in our hands ChIP for TBP using several antibodies failed to yield reliable ChIP signal at lowly expressed Polycomb genes. Nevertheless, in each case, we observe elevated binding of the examined factors specifically at Polycomb genes, including *E2f6* and *Zic2*, consistent with our single particle tracking observations. We have now integrated these new findings into Extended Data Fig. 7a of revised manuscript and drawn attention to the main text in lines 283-287.

'we depleted PRC1 and observed a clear increase in TAF1 occupancy at Polycomb enriched genes (Fig. 6a,b) which is qualitatively consistent with increased stable binding times measured in SPT (Fig. 4d,e). We also validated these effects by ChIP quantitative PCR analysis for TAF1 and other factors identified in our SPT analysis (Extended Data Fig.7a).'

2. In Figure 5a, the authors show that PRC1 degradation induces in increased binding of TAF1 genome-wide, and more specifically onto PRC1-target genes. However, the Log2FC is at maximum 0.4 (or 1.3-fold increase). How do the authors explain such a low enrichment when they see a change of 2-fold in the binding time of TAF1? What happens at *E2f6* and *Zic2* loci?

We thank the reviewer for highlighting this important point. We would like to stress that we do not believe it is possible to directly compare fold changes in binding effects measured using SPT and ChIP because each assay has its own inherent limitations and biases. For example, SPT captures the behaviour of individual molecules in live cells over a wide-range of binding times depending on the imaging modality, but is an ensemble of binding effects across distinct loci. In contrast, ChIP experiments provide locus-specific binding information, but rely on crosslinking of proteins to chromatin for which the ChIP signal is known to be highly protein-dependent and likely dictated by a combination of their residency time at specific loci and the availability of suitable residues in proximity

to chromatin to support crosslinking (see²⁵⁻²⁷). Given that SPT and CHIP rely on very different measurement parameters and limitations, we view them as complimentary or orthogonal approaches from which qualitative, as opposed to direct quantitative, comparisons should be applied.

As suggested by the reviewer, we have now used CHIP-qPCR to quantitate the increased binding of TAF1 (and other factors identified in SPT) at the *E2f6* and *Zic2* promoter before and after PRC1 depletion (Extended Data Fig. 7a). Importantly, we observe TAF1 binding increases consistent with increased binding times in SPT. To make these important points clear to the reader, we have added the following sentence to the main text of the revised manuscript on lines 283-287 as follows:

'we depleted PRC1 and observed a clear increase in TAF1 occupancy at Polycomb enriched genes (Fig. 6a,b) which is qualitatively consistent with increased stable binding times measured in SPT (Fig. 4d,e). We also validated these effects by CHIP quantitative PCR analysis for TAF1 and other factors identified in our SPT analysis (Extended Data Fig. 7a).'

3. In their previous study (Dobrinic et al., 2021. Figure 7), the authors demonstrate that PRC1 regulates gene transcription by limiting transcription burst frequency, while in their current study, the authors prove that PRC1 does not alter the duration of the “time between ON-periods” or burst frequency (line 127 to line 137 and Figure 3a). Moreover, in their previous study the authors used a 2-state model while current study is based on a 3-state model. The authors should thus discuss the discrepancy with their previous study.

As indicated by the reviewer, in Dobrinic et al 2021 we used fixed-cell single-molecule RNA-FISH (smRNA-FISH) measurements in conjunction with a simple 2-state model to infer how Polycomb target gene transcription might be controlled. The limitation of this approach is that one must ‘infer’ transcriptional kinetics from fixed-cell measurements with the aid of modelling. Using this static fixed-cell approach, in Dobrinic et al we concluded that the Polycomb system may limit burst frequency to constrain transcription. However, in contrast to smRNA-FISH based-approaches, live-cell transcription imaging now allows us to directly visualise and measure the kinetics of transcription with single-transcript sensitivity in live cells. This has allowed us to discover that Polycomb genes adhere to what appears to be a 3-state transcription behaviour where deep OFF-periods are interspersed by permissive-periods where stochastic ON-periods (or bursts) of transcription occur. By depleting PRC1 and directly observing how transcription is affected, we discover that Polycomb constrains entry into permissive-periods as opposed to limiting the time between ON-periods or burst frequency during Permissive-periods, as indicated by the reviewer.

While at face value this may appear to be discrepant with our previous findings, we believe the use of a 2-state model to infer transcription kinetics from fixed cell smRNA-FISH¹⁰ effectively compressed the features of permissive- and ON-periods into one feature (previously referred to as “inferred burst size”). In light of this limitation, our previous smRNA-FISH and modelling approach meant we could not distinguish between an effect on entry into permissive-periods as opposed to an effect on time between ON-periods (burst frequency within permissive-periods), something that now is uniquely revealed through direct live-cell imaging of transcription. Importantly, to further validate that this control mechanism explains the changes in gene expression we observe after PRC1 depletion, we develop a simple 3-state model that integrates all the transcriptional parameters measured in live cell imaging for *E2f6* and *Zic2* and show that altering the probability of entering into the permissive-state can recapitulate the expression effects observed in smRNA-FISH analysis after PRC1 depletion. We thank the reviewer for raising this point, and we have now added text to revised manuscript on lines 331-335 in the discussion to highlight how our state-of-the-art live cell imaging approaches allow us

to move beyond more coarse-grained observation/conclusions from smRNA-FISH and uncover the mechanism of transcription control by Polycomb as follows:

‘Our live-cell transcription imaging now reveals that PRC1/H2AK119ub1 primarily functions to repress transcription and gene expression by limiting transition out of a deep promoter OFF-state and into the permissive-state where ON-periods or bursts of transcription occur. Previously, using static smRNA-FISH analysis and two-state model of transcription, we concluded that PRC1 might influence gene expression by regulating transcription burst frequency (i.e. the frequency of ON-periods within permissive-periods)²¹. Now, using live-cell imaging where we are able to directly observe Polycomb gene transcription, we reveal these genes adhere to a three-state model within which PRC1 limits entry into the permissive state’

Minor comments:

1. Line 110 — Fig. 2A Its not clear how the bursting attributes are quantified. Please elaborate.

To capture ON-period features from transcription trajectories we searched local minima and maxima of three different degrees of strength to identify peaks of promoter activity (where ON-period ends) with respective, preceding in time local minima (ON period start). This strategy allows us to identify sections of the trajectories during which the signal increases as this represents loading new PolII molecules onto promoter possible only during ON-period. We have provided several examples of effectiveness of this strategy for identifying ON-periods in raw time-course transcriptional trajectories in Extended Data Fig. 3. Having identified the ON-periods we extract information about their i) duration (as we know when they begin and end), ii) amplitude (as we know the maximum transcriptional output in them), and iii) RNA PolII reinitiation from a linear fit to the data points within the ON-period (presented in Extended Data Fig. 3f). This information can be found in the “Analysis of transcription parameters from fluorescence tracks” section of materials and methods.

2. Its not clear to me why its surprising that Polycomb regulated genes when transitioned into activate states would behave any different than other genes. Several studies employing live cell imaging methods of transcription such as Wan et al (33979654) have shown that bursts sizes remain mainly consistent across several pathways and model organisms and primarily differ in burst frequency modulation. Perhaps this could be clarified in the text.

We agree with the reviewer that the use of the term ‘surprised’ (lines 114 of the original submission) in describing the absence of major differences in the ON-period features of the Polycomb genes and the reference gene was perhaps not fully warranted given that burst frequency modulation is a mechanism for transcription control in different contexts. We have now edited this sentence to replace ‘we were surprised to find’ with ‘we found’ as follows in the revised manuscript on lines 115-118 to address this:

‘When we compared ON-period features for Polycomb genes (Zic2 and E2f6) and the reference gene (Hspg2) we found that they were similar (Fig. 2b) despite Polycomb genes being much more lowly expressed (Fig. 2d)’

3. Line 114, “polycomb-target genes” vs “polycomb genes” is confusing to the reader. Please clarify.

Polycomb genes are defined based on having transcription start sites that overlap with a non-methylated CpG island (NMIs/CGIs defined in¹¹) and that are also bound by high levels of both PRC1 (RING1B) and PRC2 (SUZ12) based on ChIP-seq analysis and peak calling⁶. To ensure the origin of this classification is clear to the reader we now draw attention to Fursova et al., by citing this in the figure legend for Fig. 6a and the materials and methods section of the revised manuscript. Furthermore, we

have edited the text of the manuscript such that we only refer to ‘Polycomb genes’ to ensure this terminology is used uniformly throughout. We thank the reviewer for suggesting this clarification and have also drawn attention to what defines a Polycomb gene on lines 49-53 and cited Fursova et al as follows:

‘In vertebrates, both PRC1 and PRC2 are targeted to promoters of genes that have CpG island elements. Here they can deposit histone modifications and through feedback mechanisms create Polycomb chromatin domains that have high levels of H2AK119ub1, H3K27me3, and occupancy of PRC1/2 complexes, and we refer to these target genes where Polycomb domains form as Polycomb genes^{6,8}’

4. Lines 118-119, please mention that RING1B is a subunit of PRC1 that is targeted by the degron system.

We thank this reviewer for pointing out this oversight. We have now edited lines 120-122 of the text to ensure it is clear that RING1B is the targeted subunit as follows:

‘To test this, the MS2 reporter system was integrated into a degron cell line where addition of the small molecule auxin (IAA) leads to rapid depletion of RING1B, the catalytic subunit of PRC1, and turnover of H2AK119ub1 (Fig. 2c)^{21,26}’

5. Fig 3D. Its interesting that only a small percentage of alleles become active compared upon PRC1 degradation to the control. Why aren’t there more alleles active? Do all alleles become active eventually?

In Fig. 3d we are illustrating the fraction of time that each allele spends in the permissive-period as opposed to the percentage of alleles that become active. However, in response to the reviewer’s query, in Fig. 3e when we compare transcription imaging movies for *E2f6* and *Zic2* in the wild type (UNT) and PRC1-depleted (IAA) contexts it is evident that a large proportion of cells in the wild type state have no active alleles, whereas PRC1 depletion causes a large increase in the number of cells with active alleles, consistent with their derepression. In contrast, for the control gene, *Hspg2*, most cells have active alleles in the wild type context and this does not change after PRC1 depletion (Fig. 3e).

6. Figure 3E result is fantastic. Very clear that more alleles are active. However the fraction active doesn’t seem to be similar to that quantified in Fig 3D. How is Fig. 3D quantified?

We thank the reviewer for pointing out the importance of Fig. 3e, we also found this result very striking and immensely informative. As indicated in our response to point 5 above, Fig. 3d is not a quantification of Fig. 3e, but instead measures the fraction of time each allele spends in the permissive-period in the wild type and PRC1-depleted state. In agreement with more alleles entering into the permissive state and being transcribed after PRC1 depletion, the summed total fraction of time spent in the permissive state increases significantly for Polycomb genes. A detailed description how permissive periods were quantified is provided in the materials and methods section “Measurements of the fraction of time a promoter spends in the Permissive state” in lines 631-652.

7. It would be good to show the rates of the model in 3F.

Our very simple gene expression model in Fig. 3f does not rely on rates, but instead draws on experimentally measured parameters of transcription from live-cell imaging for both *E2f6* and *Zic2* as illustrated in Extended Data Fig. 4 and detailed in the materials and methods. In brief, in the model the number of mRNAs produced during ON-periods that occur within permissive-periods is obtained

by randomly sampling experimentally measured distributions of transcription parameters, which includes the number of ON-periods per permissive state, the ON-period-amplitude, and time interval between ON-periods (as outlined in the Ext. Data. Fig. 4f). We then use this simple model to explore how different probabilities of transitioning from the deep OFF-state into the permissive-state ($P_{O \rightarrow P}$) can shape the distribution of mRNAs produced, and compare this to experimentally obtained mRNA distributions in a cell population before and after PRC1 depletion (Fig. 3f). What is evident from this analysis is that simply increasing $P_{O \rightarrow P}$ by ~ 2 can account for the observed increases in gene expression, consistent with PRC1 controlling this feature of transcription.

8. What is the fold change for 5b? Presumably this is statistically significant.

To illustrate these effects more clearly to the reader, we have now also represented the Log2 fold changes in boxplots for each gene group in Fig. 6b. This demonstrates that the Polycomb genes on average show a larger increase in TAF1 binding (mean Log2FC=0.23) compared to the non-Polycomb target genes group (mean Log2FC=0.14), and that at non-CpG island genes (which lack Polycomb binding) there is no effect on TAF1 binding (Log2FC=0.03). These changes are statistically significant (p value $< 2.2 \times 10^{-16}$) and we now also indicate this in the figure legend.

9. Extended Figure 1a, the PCR screening experiment reveals multiple bands for the MS2x128 condition for the 3 labeled genes. Please, explain the presence of multiple amplicons. Moreover, there is no amplification corresponding to a "WT"/endogenous band, does it mean that the cell lines generated are homozygous?

The reviewer is correct that we observe multiple weaker bands for the MS2 amplicon (the strongest band is the expected size for the array) when we amplify it from engineered genomic DNA. We also observe this pattern when we PCR-amplify the MS2 array from the targeting plasmid containing the intact array. We believe the additional bands are an artefact of imprecise hybridisation and elongation events originating from portions of the repetitive amplified template DNA during the PCR cycling process. We also observe similar laddering of PCR products when we amplify other repetitive constructs containing, for example, bacterial Tet operator DNA elements. As suggested by the reviewer, the absence of a WT band indicates that the lines are homozygous.

10. Extended Figure 1b, in the MS2-GFP channel, why are there multiple spots corresponding to Hspg2 transcription? Are there random integrants of MS2-containing construct? In that same experiment, please clarify the location of the intronic probes (not overlapping the LHA/RHA sequences)?

As highlighted by the reviewer we often observe two active alleles for *Hspg2* in single cells during live-cell imaging which is also consistent with analysis based on intronic smRNA-FISH probes in fixed cells. However, we now realise that in the figure depicting transcription imaging for *Hspg2* (Extended Data Fig. 1b, lower panel MCP-GFP) we inadvertently left a spurious arrow on the image during curation that does not correspond to MCP-GFP signal (see a comparison of the submitted and revised figure in Reviewer Fig. 3 below). We apologise for this oversight and believe this may have led to confusion regarding the number of active alleles observed by the reviewer. This inadvertent oversight has now been corrected (Extended Data Fig. 1b). Based on our comparison of MCP-GFP signal and intronic smRNA-FISH we have no evidence to suggest that there are random integrations of the MS2-construct in any of our transcription imaging cell lines. Intronic RNA-FISH probes correspond exclusively to sequences in the endogenous gene sequence, not the MS2 array, and we only observed MCP-GFP signal when there is also a corresponding signal by intronic smRNA-FISH. In the revised manuscript we have now included the sequences of all RNA-FISH probe sequences used in our study in Supplementary Table 4.

Reviewer Figure 3. Panel b in Extended Data. Fig. 1 has been amended: the spurious arrow (indicated with yellow dashed circle) has been removed and outlines have been added to facilitate distinguishing individual cells.

11. Extended Figure 5a, please, discuss the HALO-tagging of endogenous proteins. Are all the proteins homozygously tagged? Moreover, why does not MED14 WB contain a band for the endogenous proteins (contrarily to the other tagged protein-associated WB)?

All HALO-tagged proteins are endogenously tagged and homozygous and we have updated the text, lines 185-187, to reflect that as follows:

‘To enable SPT we used CRISPR-Cas9 genome engineering to homozygously HALO-tag the endogenous genes corresponding to a series of core transcription regulators that represent distinct steps in transcription^{27,28} (Fig. 4c and Extended Data Fig. 5a,b).’

Regarding the MED14 western blot we have only been able to probe this line with an antibody against the tagged protein (anti-T7) which is why no signal is evident in the WT extract lane. Unfortunately, despite trying multiple commercial antibodies, none of these yielded a specific signal for the endogenous mouse MED14 protein. However, PCR genotyping of the MED14 Halo-tag lines also indicated this line is homozygote.

Reviewer #3:

Remarks to the Author:

Summary of the key results: The polycomb repressive system consists of two highly conserved complexes, PRC1 and PRC2, which are essential for mammalian development. They repress genes by depositing chromatin marks at promoters containing CpG islands. However, how these marks affect downstream transcription kinetics is unknown. Previous work has focused on ensemble measurements, making it difficult to infer the precise steps PRC1/2 act on, because genes transcribe out of sync across cells.

The paper focuses on the action of PRC1, which mono-ubiquitylates H2A. The authors measure the changes in transcription upon acute depletion of the catalytic subunit of PRC1, RING1B using smFISH and knock-in MS2 repeats into endogenous PRC1-responsive genes, enabling quantification of transcription bursting kinetics. These experiments show that PRC depletion does not affect bursting

dynamics while the genes are in a permissive state, but rather increases the time spent in the transcriptional permissive states. A simple stochastic model recapitulates the difference in total transcripts produced, predicting that PRC1 represses the entry into the permissive state. The authors support their findings using live-cell single-molecule tracking of a suite of general transcription factors in the presence or absence of PRC1, and find that PRC1 primarily influences the “early” PIC components by decreasing the bound fraction and stable binding time of these proteins. This result agrees well with the findings from MS2 experiments in which PRC1 influences the time these genes exist in a transcriptionally permissive state.

Originality and significance: The experiments are well-designed to specifically address the gap in our understanding of Polycomb repression, clearly outlining which step of the transcription cycle is impacted by PRC1 (de-repression, via inhibiting PIC assembly). These results nicely lay out what PRC1 can, and cannot do quantitatively to transcription regulation, which will inform models of expression dynamics during development. This work also identifies an inhibition of the early PIC, which will likely inspire follow up work to articulate the precise molecular intermediates of this competition.

Data & methodology: I appreciate the authors using endogenous genes to record transcriptional activity, and acute depletions that rapidly change the transcriptional state of the cell, because it provides access to direct effects of PRC depletion on relevant target genes. The suite of PIC factors measured by SMT is impressive.

We thank the reviewer for kind words. Indeed, our intention was to utilise live-cell approaches to dissect Polycomb-mediated repression and transcription in a way that previously was not possible.

Appropriate use of statistics and treatment of uncertainties: the data is rigorously acquired and analyzed. One note is that the authors chose to only analyze bursts of >2.5 transcripts. Small bursts (under the 2.5 threshold transcript) seem equally frequent, if not more frequent than larger bursts in the time traces displayed. How sensitive are the conclusions of the modeling to the choice of that threshold? E.g. what are the conclusions when a threshold of 1 transcript or more is chosen?

We thank the reviewer for raising this point. We would like to clarify that our analysis interrogates all bursts (ON-periods) of 1 or more transcripts and that no threshold has been applied. However, to measure RNA Pol II re-initiation rate within ON-periods at least two transcription events must have occurred. Therefore, we applied a >2.5 transcript threshold to identify the ON-periods where the time between RNA Pol II initiation events could be robustly calculated. This is described in detail on lines 627-629 of the materials and methods. Importantly, this does not influence the conclusions of our modelling as mRNA production in this context is derived from the amplitude and duration of the measured ON-periods.

Conclusions: Conclusions are well supported, based on a gold standard model system, the analysis of two PRC1-target and one control gene, and the analysis of several PIC components.

Clarity and context/references: The premise of the paper is well supported by the literature, and the manuscript is well written and easy to understand for a broad audience.

We thank the reviewer for the very careful consideration of our manuscript and for highlighting the importance of our new discoveries and the rigor of our approaches. We found the reviewer comments very helpful in further clarifying and supporting our findings as described in our point-to-point response to the reviewer’s suggested improvements.

Suggested improvements:

There is an apparent discrepancy in the effect sizes upon PRC1 depletion of the RNA copy number compared to that of the changes in bursting dynamics. For instance, Zic2 jumps ~2.5 fold (from 15 to 38 RNAs per cell, Fig. 2d), which should in principle be explained by a doubling of the bursting output. Yet the sole bursting parameter that changes upon PRC depletion (fraction of time in permissive period) increases by 2 fold only. More so for E2f6, which RNA levels increase by 2 fold, but for which the only significant change, the increase in permissive fraction is only ~30%. At equilibrium, the transcription output is expected to be proportional to the fraction of time spent in the active state (see eg. formula 2.9 in <https://www.ncbi.nlm.nih.gov/pmc/articles/PMC10073913/>), so this discrepancy is surprising. The first factor that comes to mind is the non-equilibrium nature of the experiment: an acute depletion followed by recording of dynamics over a time window shorter than some lifetimes in the system (the repressive state lasts several cell cycles). This is not a critique of this experimental design which is absolutely necessary to avoid secondary effects. But since there are conflicting factors at play in the dynamics, could the authors explain the reason for the apparent effect size mismatch, and provide an intuitive reason to the reader? For instance, RNA levels will lag at a high level due to RNA lifetimes in the hr range; on the other hand, incomplete re-entry into the permissive state during the experimental window might lead to lower RNA counts than expected at equilibrium.

We thank the reviewer for bringing up this important analytical point. We agree that the effects on mRNA levels (smRNA-FISH) and alterations in time spent in the permissive state (live-cell imaging) after PRC1 depletion do not necessarily appear to precisely converge and that in theory a “doubling in gene expression levels should be explained by doubling the bursting output”. We believe the explanation for this, as suggested by the reviewer, is that after rapid depletion of PRC1 we are likely creating a non-equilibrium state with respect to mRNA accumulation. Acute depletion will lead to sudden and synchronised (within the timescales examined) increase in the probability of transitioning into the permissive-state leading to transcription and ultimately transcript accumulation. However, transcript accumulation is dependent on new transcript production and mRNA half-life. As suggested by the reviewer, we cannot be certain under our experimental time-frames that we have reached the end-point equilibrium transcript level and extending our analysis to later time points would be convoluted by secondary effects on cell state caused by PRC1 depletion. As such it is perhaps not surprising that transcript accumulation and ‘bursting output’ as measured directly from transcription-imaging do not precisely converge. As advised by the reviewer we now highlight this important consideration in the revised version of the main text as follows on lines 148-152:

‘Although the relative increase in the fraction of time spent in permissive-periods and the expression changes after PRC1 depletion do not precisely converge (Fig. 3d, Fig. 2d), this is likely due to the non-equilibrium nature of transcript accumulation in our rapid degen systems which relies on the interplay between new transcript production and mRNA half-life’

Related to the previous point. Assuming the bursting metrics reflect the ‘true’ effect size of PRC1 repression in the unperturbed state (rather than the RNA levels which reflect the dynamic response of the system upon acute perturbation), the impact of PRC1 on expression seems modest (30% to 2-fold), which came a bit as a surprise, because PRC repression is often presented in the literature as an on/off switch, but these results would suggest that it is a much more subtle regulatory lever. How do the genes studied here compare to the rest of PCR-regulated genes? Are they “representative examples” of PRC repression, or weak responders?

We thank the reviewer for bringing up these important points and have discussed each in turn below.

Considering the second of these points first, we selected *E2f6* and *Zic2* as ‘representative examples’ of Polycomb target genes based on these genes having a typical Polycomb chromatin domain associated with their promoters and also being subject to derepression following PRC1 depletion based on RNA-seq and smRNA-FISH expression measurements. To illustrate this point more clearly, we have now shown genomic snapshots of the *E2f6* and *Zic2* Polycomb chromatin domains and effects on expression in new Extended Data Fig. 1c and compared these features to other classic Polycomb genes (*HoxA7* and *Meis1*). Another key parameter in our selection of *E2f6* and *Zic2* was that they must also have a level of pre-existing activation signal sufficient to drive at least some low-level transcription which is constrained by PRC1 in the unperturbed state, so that we could quantitate and compare features of transcription before and after PRC1 depletion. Lastly, *E2f6* and *Zic2* are not weak responders based on our smRNA-FISH based analysis. If you examine the fold change in transcript levels after PRC1 depletion of *E2f6* and *Zic2* and compare these to *HoxA7* and *Meis1* they are comparable (see Reviewer Fig. 4 below). The main difference is that *E2f6* and *Zic2* have higher pre-existing transcript levels and following PRC1 depletion their transcript levels are also higher. We believe this is because there is more pre-existing activation signal pushing against the repressive capacity of the Polycomb chromatin domains for *E2f6* and *Zic2*.

Reviewer Figure 4 – A comparison of transcript number with fold changes in transcript numbers after PRC1 depletion.

(a) Bar plots indicating the average transcripts per cell as measured by smRNA-FISH for the indicated genes in untreated (UNT) and 4 hours after PRC1 depletion (IAA). The measurements represent the average of 3 biological replicates with standard error.

(b) As per (a) but illustrating the fold change in average transcripts per cell.

Considering the first point, we agree with the reviewer that the representation in the literature of how the Polycomb system influences gene expression is often as that of a potent on/off repressor. We believe this stems from the fact that effects on gene expression after removing key components of the Polycomb system have historically been measured by reverse transcriptase PCR (RT-PCR). The

issue with this approach is that RT-PCR is inherently blind to transcript numbers and suffers from issues with non-linearity and detection sensitivity, especially at very low transcript levels. As such, in some contexts qRT-PCR analysis has led to expression changes being reported after depletion of key Polycomb system components in the range of dozens to hundreds of fold, in line with the view often espoused by the literature that the Polycomb system can counteract even strong activation signals. This is why our highly precise single-cell and single-molecule measurements have been so important. By directly counting individual transcripts we have demonstrated that Polycomb genes expression is very low in wild type cells and that depletion of PRC1 causes Polycomb genes to become derepressed, but that this does not correspond to the gene becoming 'activated' (see the next point below where we now explore this idea experimentally). Instead, what we observe is a central role for PRC1 in maintaining gene repression and that PRC1 depletion leads to a modest 'derepression' of Polycomb genes and now discover that this appears to manifest through rendering the gene promoter more susceptible to TFIID engagement and PIC formation. As such, we agree with the reviewer that one could consider Polycomb repression as '*a much more subtle regulatory lever*'. Nevertheless, the role of the Polycomb system in maintaining this inactive state is of fundamental importance as PRC1 depletion leads to the derepression of thousands of genes¹⁰ which causes unscheduled differentiation (or cell death) which is incompatible with even the earliest stages of embryonic development.

If they are representative, the present findings would suggest that PRC repression alone is unlikely to generate the kind of on/off switches observed during development, and maybe just provides a robustness against weak/off target activation, as the authors discuss. It would be interesting for the authors to comment on the relevance of PRC1 'strength' for its developmental role.

As suggested by the reviewer we believe that PRC1 provides robustness against 'weak/off target' activation. We have previously quantitated the effects of PRC1 depletion on expression (smRNA-FISH) of the *Meis1* Polycomb gene and compared this to expression levels in the activated state (Dobrinic et al ¹⁰ and Reviewer Fig. 5) and have now also generated a cell line in which MS2 repeats have been engineered into the first intron of *Meis1* in order to carry out new live cell imaging experiments that allow us to explore these transcriptional behaviours more thoroughly (new Extended Data Fig. 8 and Reviewer Fig. 5 below). smRNA-FISH measurements revealed that *Meis1* transcript levels are very low in the wild type context (UNT, Reviewer Fig. 5a) and that after PRC1 (4h IAA) depletion there is a roughly 2-fold increase in *Meis1* transcripts consistent with the Polycomb systems providing '*robustness against weak/off target activation*'. In contrast when cells are treated with retinoic acid which leads to activation of *Meis1* there is a roughly 30-fold increase in *Meis1* gene expression (72h RA, Reviewer Fig. 5a). This demonstrates that PRC1 can limit low level activation signals to quell inappropriate expression, whereas strong activation signals lead to accumulation of much higher transcript numbers. When we now examine the effects on transcription of *Meis1* in live cells we observe low and infrequent permissive periods in the wild type (UNT) cells. The frequency of these increases after PRC1 depletion are in line with increases in expression and consistent with our findings for *Zic2* and *E2f6* (Fig. 3e). However, in contrast, treatment with RA causes a dramatic increase in *Meis1* transcription, consistent with strong activation signal allowing the gene to become transcriptionally 'activated' where cells can accumulate high transcript levels. To ensure that the point about the Polycomb systems limiting low level activation signals is clear to the reader and to highlight the potential implications for development as suggested by the reviewer we have included the new live cell imaging experiments for *Meis1* in Extended Data Fig. 8 and added following sentences on lines 366-372 of the discussion in the revised manuscript:

'In the context of developmental transitions when Polycomb genes become activated, we envisage that limiting the frequency of entering into permissive-periods could also ensure

low-level activation signals are quelled, yet the gene promoter would remain receptive to strong and persistent activation signals necessary to initiate gene expression, as we show is the case of the Polycomb gene *Meis1* (Extended Data Fig. 8). Counteracting weak or inappropriate activation signals may be particularly important during development for suppressing noise and maintaining cell identity, as has been proposed previously as a key role for the Polycomb system⁶⁷

Reviewer Figure 5. Gene expression and transcription analysis of *Meis1* upon PRC1 depletion or activation.

(a) mRNA-FISH analysis of *Meis1*. Average transcript per cell numbers for *Meis1* (UNT), after PRC1 depletion (4h IAA), and after activation using 72h retinoic acid (RA) in absence of leukemia inhibitory factor (72h RA). Individual dots represent biological replicates (n=3).

(b) Live-cell transcription imaging of *Meis1*. In untreated (UNT), PRC1 depleted (IAA) and activated state (after 72 RA treatment). 141 cells per each condition are presented in the heat-map.

The value of $P_{O>P}$ for Zic2 IAA is 0.54 in Fig 3f, which does not seem to coincide with the position of the minimum in Extended data Fig. 4g (bottom left panel). Can the authors clarify or correct?

We thank the reviewer for pointing out this oversight. During figure preparation the $p_{O>P}$ values on the X-axis were inadvertently offset by 0.1. We apologise for this oversight and have now corrected the axis label in revised Extended Data Fig. 4g.

“When we examined the dynamics of other TFIID components, TAF11 showed an increased bound fraction whereas TAF1 was unaffected but both factors displayed increases in stable binding time.” This is curious, how might this be possible?

We thank the reviewer for highlighting this point. We were also curious why the bound fraction of TAF11 increased after PRC1 depletion, whereas the TAF1 was unaffected, given that they are both components of TFIID and both showed increased stable binding time after PRC1 depletion. We don’t yet have a full explanation for this difference, however we think it might be due to distinct dynamics and possibly how different structural lobes of TFIID interact with and assemble in the nucleus²⁸. Based on cryo-EM structures of the TFIID holocomplex, TAF11 resides in structural Lobe A and TAF1 is a key structural component of Lobe B/C (Reviewer Fig. 6a). While it is thought that binding of the TFIID complex to promoters occurs as an intact entity, this has never been interrogated using live-cell measurements that capture the dynamics of its individual subunits. While not a focus of our current

study, we have started to explore the inherent requirements of distinct TFIID lobes for the chromatin binding dynamics of the complex using SPT. As an inroad to this problem, we first dTAG-tagged TAF1 in the HALO-Tag-TAF11 cell line. We then depleted TAF1 (lobe B/C) and asked whether this influences TAF11 (Lobe A) dynamics. Unexpectedly, this revealed that the bound fraction of TAF11 was unaffected after the depletion of TAF1, and if anything increased slightly. In contrast, the long stable binding times of TAF11 were greatly reduced. This suggests that the bound fraction of TAF11 captured by SPT may correspond to a form that dynamically associates with chromatin, but does not exclusively sit within a stable TFIID holocomplex. In contrast, it would appear that long stable binding events correspond to inclusion of TAF11 in a TFIID holocomplex that includes Lobe B/C and rely on the presence of TAF1. These observations are compatible with the knowledge that TFIID subcomplexes can form as distinct entities, at least in the cytoplasm before they are transported to the nucleus and recent publications of where it was also proposed that TFIID subcomplexes may engage with chromatin independently of holo-complex formation^{29,30}, mentioned in lines 212-214.

The reason we highlight these findings, is that they speak to the point raised by the reviewer in that the binding kinetics of individual TFIID subunits are not inextricably linked, consistent with the distinct effects on TAF11 and TAF1 binding we observe after PRC1 depletion. A speculative corollary from these preliminary interpretations could be that more dynamically associating TFIID subcomplexes are disproportionately influenced by PRC1 and that this has knock-on effect that ultimately shapes the stable binding of the TFIID holocomplex as is evident from the concordant effects on TAF11 and TAF1 stable binding times. While this is a seductive hypothesis, at this point we are reticent to make such inferences without first carrying out a detailed systematic dissection of how: (1) each TFIID subcomplex component binds chromatin using SPT, (2) depletion of each component within the TFIID affects the dynamics of other factors in the complex, and (3) we have a genome-wide understanding of where these binding events occur and how the above depletions affect occupancy at specific sites across the genome (ChIP). This work is distinct from our current study, but will be an important focus of future work aimed at dissecting the mechanisms of TFIID assembly and promoter binding.

Reviewer Figure 6. TAF1 (lobe B/C) does not influence the bound fraction of TAF11 (lobe A) but is required for its stable binding.

(a) A schematic of TFIID complex structure. Lobes A and B/C core are indicated. (b) Western blot analysis of TAF11 after introducing a HaloTag to enable Single Particle Tracking experiments (left). Arrows indicate wild type (WT) and HaloTagged (HT) protein bands. Western blot analysis of T7-dTAG-TAF1 (right) before and after inducing degradation (UNT and 2h dTAG-13, respectively). (c) Single Particle Tracking analysis of bound fraction (left) and stable binding time of HT-TAF11 (TFIID lobe A) after dTAG-TAF1 (lobe B/C) depletion.

If the stable binding time is increased by a factor of 2, surely that would influence the bound fraction of TAF1? Unless the TAF1 on-rate onto chromatin is reduced by the same factor?

Intuitively, one would assume if the stable binding times for TAF1 increase this should also cause increases in the bound fraction. However, for such an effect to manifest, the number of stable binding events (measured with 500ms exposure imaging) needs to make up a large enough proportion of total binding events (measured with 15 ms exposure imaging) to have an effect on the total bound fraction measurements. The fact that elevated stable binding time of TAF1 after PRC1 depletion does not equate to a discernible effect on the total bound fraction, suggests the fraction of stably bound TAF1 molecules is low and that PRC1 primarily affects the stable binding time of TAF1. Importantly, it has

been proposed that stable binding of TFIID to the promoter is indicative of functional PICs, in agreement with the effect on expression we observe being related to the elevated stable binding times of TFIID (TAF1, TAF11 and TBP).

“We then depleted either PRC1 or PRC1 and TAF1 simultaneously and examined the expression of the *Zic2* and *E2f6* Polycomb target genes using smRNA-FISH (Fig. 5E, F). Importantly, this revealed that neither Polycomb target gene was derepressed in the absence of TAF1, suggesting that TFIID binding enables elevated expression in the absence of PRC1/H2AK119ub1.” Depletion of TAF1 does not completely abolish transcription, and does not seem to have any substantial effect on *E2f6* and *Zic2* when PRC1 is present (Fig 5F). This seems strange as it is a basal transcription factor. Could the authors comment on why they think this is the case?

In Fig. 5f we are using smRNA-FISH to count and quantify mRNA transcripts after acute TAF1 or PRC1/TAF1 depletion. This assay was primarily designed to capture derepression of Polycomb genes and ask if new transcript production required TFIID as we hypothesised based on elevated TAF1 binding in SPT and ChIP-seq analysis. However, importantly, any transcripts that were produced before our rapid depletion of TAF1 would remain until they were turned over by the RNA-decay machinery¹³. Therefore, while our approach is well suited to capturing Polycomb gene derepression and the effects that TAF1 has on this, decreases in expression are not as easily captured under the timescales of our experiments due to the requirement for pre-existing RNAs to be turned over. Therefore, in the context of the experiments in Fig. 5f the dTAG-13 treatment and acute removal of TAF1 alone only causes modest, but discernible, reductions in transcript numbers as most pre-existing mRNAs have not yet had time to be turned-over in agreement with the measured half-life of *E2f6* and *Zic2* transcripts being several hours (Extended Data Fig. 4e)¹³.

“Interestingly, we also observed a modest yet significant increase in TAF1 binding across non-Polycomb enriched transcription start sites, indicating that PRC1 may constrain the binding of TFIID more broadly (Fig. 5B and Extended Data Fig. 6B). Consistent with this possibility, low levels of PRC1 are detected at non-Polycomb gene promoters, and when we analysed gene expression across these genes we also observed a modest increase in expression after PRC1 depletion (Extended Data Fig. 6A)”. Is there a positive correlation between the increase of TAF1 at promoters and an increase in logFC of expression for these genes? The Extended Data Fig. 6 shows the distribution of these parameters in bulk, but did the authors consider plotting these against each other in a scatter plot?

As suggested by the reviewer we have now created scatter plots comparing the log₂FC in TAF1 ChIP-seq signal and the log₂FC in RNA-seq across Polycomb genes, non-Polycomb genes, and non-CGI genes (Extended Data Fig. 7e and have also illustrated this in Reviewer Fig. 7 below for convenience). There is a modest positive correlation between the increase in TAF1 binding and the increase in transcription upon removal of PRC1 at Polycomb genes as expected, with a more nuanced correlation at non-Poycomb genes, and no correlation at non-CGI genes. We thank the reviewer for suggesting this way of illustrating the data, and have now drawn attention to this important observation on lines 287-296 of the revised manuscript as follows:

‘Interestingly, in cChIP-seq analysis we also observed a modest yet significant increase in TAF1 binding across non-Polycomb gene transcription start sites, indicating that PRC1 may also constrain the binding of TFIID more broadly (Fig. 6b and Extended Data Fig. 7b). Consistent with this possibility, low levels of PRC1 are detected at non-Polycomb gene promoters, and when we analysed gene expression across these genes we also observed a modest increase in expression after PRC1 depletion (Fig. 6a, Extended Data Fig. 7c). These findings are in line with previous observations that PRC1 and H2AK119ub1 may also have more subtle yet pervasive effects on gene expression^{8,21}. Nevertheless, we find the effects on expression and increases in TAF1 binding correlated best at Polycomb genes (Extended Data Fig. 7e), in agreement with the Polycomb system playing a prominent role maintaining these genes in a lowly transcribed or inactive state.’

Reviewer figure 7- Correlation between changes in TAF1 binding (ChIP-seq, Log2 fold change) and Expression (RNA-seq, Log2 fold change) after PRC1 depletion at Polycomb, non-Polycomb, and non-CpG Island (non-CGI) genes.

“Based on these detailed kinetic measurements, we find that PRC1/H2AK119ub1 limits the binding of factors involved in the earliest stages of PIC formation (Fig. 4E).” The interpretation is consistent with the data, but this statement seems too strong without any corresponding *in vitro* data to support this claim, though such studies are beyond the scope of this study. The ChIP data is certainly helpful towards making this conclusion, but secondary effects of the removal of PRC1 perturbation cannot be conclusively ruled out in the context of a live cell. Perhaps the authors could slightly soften some of the language surrounding this claim, and make clear these potential caveats?

We thank the reviewer for highlighting this important point. While our live-cell imaging approaches provide a major new conceptual advance in our understanding of how the Polycomb system controls transcription, we agree with the reviewer that it would be helpful to soften the language used in this specific sentence by changing ‘we find’ to ‘we propose’ as follows in lines 235-237 of the revised manuscript.

‘Based on these detailed kinetic measurements, we propose’ that PRC1/H2AK119ub1 limits the binding of factors involved in the earliest stages of PIC formation (Fig. 4e).’

As highlighted in the discussion section (lines 354-356) our new discoveries set the stage for ‘*future in vitro biochemical and structural work ... to understand whether H2AK119ub1 influences how the core transcriptional machinery interacts with promoter chromatin to enable gene repression*’. This will require fully reconstituted *in vitro* transcription assays on chromatin templates containing H2AK119ub1 and single molecule imaging approaches to simultaneously examine PIC formation and

transcription. As such, we agree that our discoveries now set the stage for this challenging new endeavour and explicitly state that this is beyond the scope of our current study as highlighted by the reviewer.

References:

- 1 Farcas, A. M. *et al.* KDM2B links the Polycomb Repressive Complex 1 (PRC1) to recognition of CpG islands. *Elife* **1**, e00205 (2012). <https://doi.org:10.7554/eLife.00205>
- 2 Blackledge, N. P. & Klose, R. J. The molecular principles of gene regulation by Polycomb repressive complexes. *Nat Rev Mol Cell Biol* **22**, 815-833 (2021). <https://doi.org:10.1038/s41580-021-00398-y>
- 3 Rose, N. R. *et al.* RYBP stimulates PRC1 to shape chromatin-based communication between Polycomb repressive complexes. *Elife* **5** (2016). <https://doi.org:10.7554/eLife.18591>
- 4 Blackledge, N. P. *et al.* Variant PRC1 complex-dependent H2A ubiquitylation drives PRC2 recruitment and polycomb domain formation. *Cell* **157**, 1445-1459 (2014). <https://doi.org:10.1016/j.cell.2014.05.004>
- 5 Blackledge, N. P. *et al.* PRC1 Catalytic Activity Is Central to Polycomb System Function. *Mol Cell* **77**, 857-874 e859 (2020). <https://doi.org:10.1016/j.molcel.2019.12.001>
- 6 Fursova, N. A. *et al.* Synergy between Variant PRC1 Complexes Defines Polycomb-Mediated Gene Repression. *Mol Cell* **74**, 1020-1036 e1028 (2019). <https://doi.org:10.1016/j.molcel.2019.03.024>
- 7 Schmitges, F. W. *et al.* Histone methylation by PRC2 is inhibited by active chromatin marks. *Mol Cell* **42**, 330-341 (2011). <https://doi.org:10.1016/j.molcel.2011.03.025>
- 8 Scelfo, A. *et al.* Functional Landscape of PCGF Proteins Reveals Both RING1A/B-Dependent- and RING1A/B-Independent-Specific Activities. *Mol Cell* **74**, 1037-1052 e1037 (2019). <https://doi.org:10.1016/j.molcel.2019.04.002>
- 9 Tamburri, S. *et al.* Histone H2AK119 Mono-Ubiquitination Is Essential for Polycomb-Mediated Transcriptional Repression. *Mol Cell* **77**, 840-856 e845 (2020). <https://doi.org:10.1016/j.molcel.2019.11.021>
- 10 Dobrinic, P., Szczurek, A. T. & Klose, R. J. PRC1 drives Polycomb-mediated gene repression by controlling transcription initiation and burst frequency. *Nat Struct Mol Biol* **28**, 811-824 (2021). <https://doi.org:10.1038/s41594-021-00661-y>
- 11 Long, H. K. *et al.* Epigenetic conservation at gene regulatory elements revealed by non-methylated DNA profiling in seven vertebrates. *Elife* **2**, e00348 (2013). <https://doi.org:10.7554/eLife.00348>
- 12 Endoh, M. *et al.* PCGF6-PRC1 suppresses premature differentiation of mouse embryonic stem cells by regulating germ cell-related genes. *Elife* **6** (2017). <https://doi.org:10.7554/eLife.21064>
- 13 Herzog, V. A. *et al.* Thiol-linked alkylation of RNA to assess expression dynamics. *Nat Methods* **14**, 1198-1204 (2017). <https://doi.org:10.1038/nmeth.4435>
- 14 Lehmann, L. *et al.* Polycomb repressive complex 1 (PRC1) disassembles RNA polymerase II preinitiation complexes. *J Biol Chem* **287**, 35784-35794 (2012). <https://doi.org:10.1074/jbc.M112.397430>
- 15 Stock, J. K. *et al.* Ring1-mediated ubiquitination of H2A restrains poised RNA polymerase II at bivalent genes in mouse ES cells. *Nat Cell Biol* **9**, 1428-1435 (2007). <https://doi.org:10.1038/ncb1663>
- 16 Donczew, R., Warfield, L., Pacheco, D., Erijman, A. & Hahn, S. Two roles for the yeast transcription coactivator SAGA and a set of genes redundantly regulated by TFIID and SAGA. *Elife* **9** (2020). <https://doi.org:10.7554/eLife.50109>

- 17 Sun, F. *et al.* The Pol II preinitiation complex (PIC) influences Mediator binding but not promoter-enhancer looping. *Genes Dev* **35**, 1175-1189 (2021). <https://doi.org:10.1101/gad.348471.121>
- 18 Petrenko, N., Jin, Y., Dong, L., Wong, K. H. & Struhl, K. Requirements for RNA polymerase II preinitiation complex formation in vivo. *Elife* **8** (2019). <https://doi.org:10.7554/eLife.43654>
- 19 Warfield, L. *et al.* Transcription of Nearly All Yeast RNA Polymerase II-Transcribed Genes Is Dependent on Transcription Factor TFIID. *Mol Cell* **68**, 118-129 e115 (2017). <https://doi.org:10.1016/j.molcel.2017.08.014>
- 20 Santana, J. F., Collins, G. S., Parida, M., Luse, D. S. & Price, D. H. Differential dependencies of human RNA polymerase II promoters on TBP, TAF1, TFIIB and XPB. *Nucleic Acids Res* **50**, 9127-9148 (2022). <https://doi.org:10.1093/nar/gkac678>
- 21 Serebreni, L. *et al.* Functionally distinct promoter classes initiate transcription via different mechanisms reflected in focused versus dispersed initiation patterns. *EMBO J* **42**, e113519 (2023). <https://doi.org:10.15252/emboj.2023113519>
- 22 Matsui, T., Segall, J., Weil, P. A. & Roeder, R. G. Multiple factors required for accurate initiation of transcription by purified RNA polymerase II. *J Biol Chem* **255**, 11992-11996 (1980).
- 23 He, Y., Fang, J., Taatjes, D. J. & Nogales, E. Structural visualization of key steps in human transcription initiation. *Nature* **495**, 481-486 (2013). <https://doi.org:10.1038/nature11991>
- 24 Kwan, J. Z. J. *et al.* RNA Polymerase II transcription independent of TBP in murine embryonic stem cells. *Elife* **12** (2023). <https://doi.org:10.7554/eLife.83810>
- 25 Teves, S. S. *et al.* A dynamic mode of mitotic bookmarking by transcription factors. *Elife* **5** (2016). <https://doi.org:10.7554/eLife.22280>
- 26 Baranello, L., Kouzine, F., Sanford, S. & Levens, D. ChIP bias as a function of cross-linking time. *Chromosome Res* **24**, 175-181 (2016). <https://doi.org:10.1007/s10577-015-9509-1>
- 27 Schmiedeberg, L., Skene, P., Deaton, A. & Bird, A. A temporal threshold for formaldehyde crosslinking and fixation. *PLoS One* **4**, e4636 (2009). <https://doi.org:10.1371/journal.pone.0004636>
- 28 Patel, A. B. *et al.* Structure of human TFIID and mechanism of TBP loading onto promoter DNA. *Science* **362** (2018). <https://doi.org:10.1126/science.aau8872>
- 29 Bernardini, A. *et al.* Hierarchical TAF1-dependent co-translational assembly of the basal transcription factor TFIID. *Nat Struct Mol Biol* **30**, 1141-1152 (2023). <https://doi.org:10.1038/s41594-023-01026-3>
- 30 Hisler, V. *et al.* RNA polymerase II transcription with partially assembled TFIID complexes. *bioRxiv* (2023). <https://doi.org:10.1101/2023.11.27.567046>

Decision Letter, first revision:

Our ref: NCB-A51828A

25th April 2024

Dear Dr. Klose,

Thank you for submitting your revised manuscript "Polycomb sustains promoters in a deep OFF-state by limiting PIC formation to counteract transcription" (NCB-A51828A). It has now been seen by the original referees and their comments are below. The reviewers find that the paper has improved in revision, and therefore we'll be happy in principle to publish it in Nature Cell Biology, pending minor revisions to satisfy the referees' final requests and to comply with our editorial and formatting guidelines.

Thank you again for your interest in Nature Cell Biology Please do not hesitate to contact me if you have any questions.

Sincerely,

Sabrya Carim, PhD
(she/her/hers)
Associate Editor, Nature Cell Biology
Nature Portfolio

Springer Nature
The Campus, 4 Crinan Street, London N1 9XW, UK
sabrya.carim@springernature.com
<https://orcid.org/0000-0001-9485-1938>

Reviewer #1 (Remarks to the Author):

The authors have thoroughly addressed all of my comments. However, there are significant text changes required to now adapt the paper to the new data, as outlined below.

The new Figure 5 changes the paper's message very significantly. This is not yet reflected in the

revised manuscript and would be a very big issue if not addressed. The new findings have have significance to the title, abstract and whole narrative of the paper, and in particular their choice to refer to ncPRC1 and cPRC1 as "PRC1". In the new Figure 5, they now show that it is ncPRC1, and not cPRC1, that is the primary determinant in regulating stable TFIID binding and transcriptional repression. Importantly, since the term "PRC1" is an umbrella term for both ncPRC1 and cPRC1, and since their new data adds to what we already know - that these respective different forms of PRC1 have clearly different, and sometimes overlapping functions - then they must specify ncPRC1 unambiguously in the title, abstract, and in the introduction last paragraph and throughout the text thereafter. To be clear, "PRC1" should be replaced with one of "ncPRC1", "cPRC1" or "ncPRC1 and cPRC1" or "all forms of PRC1" throughout the title, abstract, introduction, results, legends and discussion, as appropriate.

Furthermore, based on the findings presented in the new Extended Figure 6A, it is recommended to designate the genes as "ncPRC1 only bound" (instead of "non-Polycomb") and "ncPRC1 and cPRC1 bound" (instead of "Polycomb") throughout the manuscript. This would add further clarity.

Reviewer #2 (Remarks to the Author):

Comments and concerns have been satisfactorily addressed by Szczurek et al.

Reviewer #3 (Remarks to the Author):

all comments have been addressed. The Meis1 live imaging data is a nice addition!

Decision Letter, final checks:

Our ref: NCB-A51828A

13th May 2024

Dear Dr. Klose,

Thank you for your patience as we've prepared the guidelines for final submission of your Nature Cell Biology manuscript, "Polycomb sustains promoters in a deep OFF-state by limiting PIC formation to counteract transcription" (NCB-A51828A). Please carefully follow the step-by-step instructions provided in the attached file, and add a response in each row of the table to indicate the changes that you have made. Ensuring that each point is addressed will help to ensure that your revised manuscript can be swiftly handed over to our production team.

In recognition of the time and expertise our reviewers provide to Nature Cell Biology's editorial process, we would like to formally acknowledge their contribution to the external peer review of your manuscript entitled "Polycomb sustains promoters in a deep OFF-state by limiting PIC formation to counteract transcription". For those reviewers who give their assent, we will be publishing their names alongside the published article.

Nature Cell Biology offers a Transparent Peer Review option for new original research manuscripts submitted after December 1st, 2019. As part of this initiative, we encourage our authors to support increased transparency into the peer review process by agreeing to have the reviewer comments, author rebuttal letters, and editorial decision letters published as a Supplementary item. When you submit your final files please clearly state in your cover letter whether or not you would like to participate in this initiative. Please note that failure to state your preference will result in delays in accepting your manuscript for publication.

Cover suggestions

COVER ARTWORK: We welcome submissions of artwork for consideration for our cover. For more information, please see our guide for cover artwork.

Nature Cell Biology has now transitioned to a unified Rights Collection system which will allow our Author Services team to quickly and easily collect the rights and permissions required to publish your work. Approximately 10 days after your paper is formally accepted, you will receive an email in providing you with a link to complete the grant of rights. If your paper is eligible for Open Access, our Author Services team will also be in touch regarding any additional information that may be required to arrange payment for your article.

Please note that *Nature Cell Biology* is a Transformative Journal (TJ). Authors may publish their research with us through the traditional subscription access route or make their paper immediately open access through payment of an article-processing charge (APC). Authors will not be required to make a final decision about access to their article until it has been accepted. Find out more about Transformative Journals

Please note that you will not receive your proofs until the publishing agreement has been received

through our system.

[Redacted]

Best regards,

Kendra Donahue
Staff
Nature Cell Biology

On behalf of

Sabrya Carim, PhD
(she/her/hers)
Associate Editor, Nature Cell Biology
Nature Portfolio

Springer Nature
The Campus, 4 Crinan Street, London N1 9XW, UK
sabrya.carim@springernature.com
<https://orcid.org/0000-0001-9485-1938>

Reviewer #1:

Remarks to the Author:

The authors have thoroughly addressed all of my comments. However, there are significant text changes required to now adapt the paper to the new data, as outlined below.

The new Figure 5 changes the paper's message very significantly. This is not yet reflected in the revised manuscript and would be a very big issue if not addressed. The new findings have have significance to the title, abstract and whole narrative of the paper, and in particular their choice to refer to ncPRC1 and cPRC1 as "PRC1". In the new Figure 5, they now show that it is ncPRC1, and not cPRC1, that is the primary determinant in regulating stable TFIID binding and transcriptional repression. Importantly, since the term "PRC1" is an umbrella term for both ncPRC1 and cPRC1, and since their new data adds to what we already know - that these respective different forms of PRC1 have clearly different, and sometimes overlapping functions - then they must specify ncPRC1

unambiguously in the title, abstract, and in the introduction last paragraph and throughout the text thereafter. To be clear, "PRC1" should be replaced with one of "ncPRC1", "cPRC1" or "ncPRC1 and cPRC1" or "all forms of PRC1" throughout the title, abstract, introduction, results, legends and discussion, as appropriate.

Furthermore, based on the findings presented in the new Extended Figure 6A, it is recommended to designate the genes as "ncPRC1 only bound" (instead of "non-Polycomb") and "ncPRC1 and cPRC1 bound" (instead of "Polycomb") throughout the manuscript. This would add further clarity.

Reviewer #2:

Remarks to the Author:

Comments and concerns have been satisfactorily addressed by Szczurek et al.

Reviewer #3:

Remarks to the Author:

all comments have been addressed. The *Meis1* live imaging data is a nice addition!

Author Rebuttal, first revision:

We would like to thank the reviewers for taking the time to consider our revisions in light of their original comments.

We were very pleased that all three reviewers concluded that we had thoroughly addressed all of their comments in the revised manuscript and appreciated the additional supportive comment that the new *Meis1* live-imaging experiments were a nice addition.

Reviewer 1 had one last request that ensure the contribution of canonical (cPRC1) and non-canonical (ncPRC1) PRC1 is made clear throughout the manuscript. We have achieved this by drawing attention to this important distinction in the last paragraph of the introduction (lines 76-77), in the text section associated with Figure 5 (lines 231-258) where we describe our findings that cPRC1 is not responsible for repression, and also reiterate/contextualize this point explicitly in the discussion (lines 305-306, 324-327, 366-368). However, in order to ensure that our work is accessible to the broad readership of Nature Cell Biology, and to meet the word limit restrictions, we have refrained from going into this level of detail in the Title and Abstract. Furthermore, we do not stipulate in the results section of the manuscript that the effects we observe are due to either cPRC1 or ncPRC1 until we introduce this distinction and carry out experiments to test their contribution in Figure 5. Until that point in the manuscript we refer to having depleted PRC1 (both cPRC1 and ncPRC1) as this is what is achieved by removing its core structural components

(RING1A/B). We believe that the contribution of cPRC1 and ncPRC1 complexes to PIC formation and Polycomb mediated repression are now clear in the finalized manuscript and we thank the reviewer for suggesting these final refinements.

Final Decision Letter:

Dear Dr Klose,

I am pleased to inform you that your manuscript, "The Polycomb system sustains promoters in a deep OFF-state by limiting pre-initiation complex formation to counteract transcription", has now been accepted for publication in Nature Cell Biology. Congratulations!

Please note that *Nature Cell Biology* is a Transformative Journal (TJ). Authors may publish their research with us through the traditional subscription access route or make their paper immediately open access through payment of an article-processing charge (APC). Authors will not be required to make a final decision about access to their article until it has been accepted. Find out more about Transformative Journals

If you have not already done so, we strongly recommend that you upload the step-by-step protocols used in this manuscript to protocols.io (<https://protocols.io>), an open online resource that allows researchers to share their detailed experimental know-how. All uploaded protocols are made freely available and are assigned DOIs for ease of citation. Protocols and Nature Portfolio journal papers in which they are used can be linked to one another, and this link is clearly and prominently visible in the online versions of both. Authors who performed the specific experiments can act as primary authors for the Protocol as they will be best placed to share the methodology details, but the Corresponding Author of the present research paper should be included as one of the authors. By uploading your Protocols onto protocols.io, you are enabling researchers to more readily reproduce or adapt the methodology you use, as well as increasing the visibility of your protocols and papers. You can also establish a dedicated workspace to collect your lab Protocols. Further information can be found at <https://www.protocols.io/help/publish-articles>.

You can use a single sign-on for all your accounts, view the status of all your manuscript submissions and reviews, access usage statistics for your published articles and download a record of your

refereeing activity for the Nature Portfolio.

With kind regards,

Sabrya Carim, PhD
(she/her/hers)
Associate Editor, Nature Cell Biology
Nature Portfolio

Springer Nature
The Campus, 4 Crinan Street, London N1 9XW, UK
sabrya.carim@springernature.com
<https://orcid.org/0000-0001-9485-1938>

** Visit the Springer Nature Editorial and Publishing website at www.springernature.com/editorial-and-publishing-jobs for more information about our career opportunities. If you have any questions please click here.**